

# The role of elevated atmospheric $CO_2$ and increased fire in Arctic amplification of temperature during the Early to mid-Pliocene

Tamara Fletcher[1*], Lisa Warden[2*], Jaap S. Sinninghe Damsté[2,3], Kendrick J. Brown[4,5], Natalia Rybczynski[6,7], John Gosse[8], and Ashley P Ballantyne[1]

[1] College of Forestry and Conservation, University of Montana, Missoula, 59812, USA
[2] Department of Marine Microbiology and Biogeochemistry, NIOZ Royal Netherlands Institute for Sea Research, Den Berg, 1790, Netherlands
[3] Department of Earth Sciences, University of Utrecht, Utrecht, 3508, Netherlands
[4] Natural Resources Canada, Canadian Forest Service, Victoria, V8Z 1M, Canada
[5] Department of Earth, Environmental and Geographic Science, University of British Columbia Okanagan, Kelowna, V1V 1V7, Canada
[6] Department of Palaeobiology, Canadian Museum of Nature, Ottawa, K1P 6P4, Canada
[7] Department of Biology & Department of Earth Sciences, Carleton University, Ottawa, K1S 5B6, Canada
[8] Department of Earth Sciences, Dalhousie University, Halifax, B3H 4R2, Canada
*Authors contributed equally to this work

*Correspondence to*: Tamara Fletcher (tamara.fletcher@umontana.edu)

**Abstract.** The mid-Pliocene is a valuable time interval for understanding the mechanisms that determine equilibrium climate at current $CO_2$ concentrations. One intriguing, but not fully understood, feature of the early to mid-Pliocene climate is the amplified arctic temperature response. Current models underestimate the degree of warming in the Pliocene Arctic and validation of proposed feedbacks is limited by scarce terrestrial records of climate and environment, as well as discrepancies in current $CO_2$ proxy reconstructions. Here we reconstruct the $CO_2$ and summer temperature from a re-dated 3.9 +1.5/-0.5 Ma sub-fossil fen-peat deposit on west-central Ellesmere Island, Canada, and investigate fire as a potential feedback to Arctic amplification of warming during the mid-Pliocene.

Average $CO_2$ was determined using isotope ratios of mosses to be 440 ± 50 ppm. The estimate for average mean summer temperature is 15.4±0.8°C using specific bacterial membrane lipids, i.e. branched glycerol dialkyl glycerol tetraethers. Macro-charcoal was present in all samples from this Pliocene section with notably higher charcoal concentration in the upper part of the sequence. This change in charcoal was synchronous with a change in vegetation that saw fire promoting taxa increase in abundance. Paleovegetation reconstructions are consistent with warm summer temperatures, relatively low summer precipitation and an incidence of fire comparable to fire adapted boreal forests of North America, or potentially central Siberia. To our knowledge, this study represents the furthest northern evidence of fire during the Pliocene and highlights the important role of forest fire in the ecology and climatic processes of the Pliocene High Arctic. The results provide evidence that terrestrial fossil localities in the Pliocene High Arctic were probably formed during warm intervals that coincided with relatively high $CO_2$ concentrations that supported productive biotic communities. This study indicates that interactions between



paleovegetation and paleoclimate were mediated by fire in the High Arctic during the Pliocene, even though $CO_2$ concentrations were only ~30 ppm higher than modern.

# 1 Introduction

Current rates of warming in the Arctic are almost double the rate of global warming. Since 1850, global land surface
temperatures have increased by approximately 1.0°C, whereas arctic surface land temperatures have increased by 2.0°C (Jones and Moberg, 2003; Pagani et al., 2010). Such arctic amplification of temperatures has also occurred during other warm climate anomalies in Earth's past. Paleoclimate records from the Arctic indicate that the change in arctic summer temperatures during past global warm periods was 3–4 times larger than global temperature change (Miller et al., 2010). While the latest ensemble of earth system models (ESMs) provide fairly accurate predictions of the modern amplification of arctic temperatures hitherto
observed (Marshall et al., 2014), they often under-predict the amplification of arctic temperatures during past warm intervals in Earth's history, including the Eocene (33.9–56 Ma; Huber, 2008; Shellito et al., 2009), and the Pliocene (2.6–5.3 Ma; Dowsett et al., 2012; Salzmann et al., 2013) epochs. These differences suggest that either the models are not simulating the full array of feedback mechanisms properly for past climates, or that the full array of fast and slow feedback mechanisms have not fully engaged for the modern Arctic. If the later, the Arctic region has yet to reach the full amplification potential
demonstrated in the past.

The Pliocene is an intriguing climatic interval that may offer important insights into climate feedbacks. It captures the transition from the preceding Miocene to the Pleistocene when orbital regulators of climate transitioned from the 41,000 year obliquity cycle to the 100,000 year eccentricity cycle, respectively. Atmospheric $CO_2$ values likewise varied (Royer et al., 2007) decreasing from values comparable to modern (Haywood et al., 2016; Pagani et al., 2010; Stap et al., 2016), to lower
levels (Raymo et al., 2006); a state transition that may revert in the future under high $CO_2$. Of additional importance, continental configurations were similar to present (Dowsett et al., 2016). While global mean annual temperatures (MATs) during the Pliocene were only ~ 3°C warmer than present day (Fig. 1), arctic land surface MATs may have been as much as 15 to 20°C warmer (Ballantyne et al., 2010; Csank et al., 2011a; Csank et al., 2011b; Fletcher et al., 2017). Further, arctic sea surface temperatures may have been as much as 10 to 15°C warmer than modern (Robinson, 2009), and sea-levels were approximately
25m higher than present (Dowsett et al., 2016). As such, the terrestrial environment of the Arctic was significantly different, with tree line ecosystems at much higher latitudes nearly eliminating the tundra biome (Salzmann et al., 2008).

Several mechanisms have been proposed as drivers of arctic amplification, including vastly reduce sea-ice extent (Ballantyne et al., 2013), cloud and atmospheric water vapor effects (e.g. Feng et al., 2016; Swann et al., 2010), vegetation controls on albedo (Otto-Bliesner and Upchurch Jr, 1997), and increased meridional heat transport by the oceans (Dowsett et al., 1992)
though it is now considered to be of lesser influence (Hwang et al., 2011). We propose that fire in arctic ecosystems may also be an important proximal mechanism for amplifying arctic surface temperatures during the Pliocene.





Deposition of modern black carbon from industrial emissions caused a decrease in albedo of ice that may have accelerated ice melt over the 20th century (McConnell et al., 2007). However, in natural systems fire has complex impacts on local radiative budgets. It influences thermal regimes directly through changes in albedo, both due to the impact of vegetation change on albedo (Chapin III et al., 2005) and production of black carbon. The net radiative impact of forest fires in the modern boreal
forest is thought to be a slight cooling due to enhanced albedo when canopy cover is lost, compensating for black carbon effects (Randerson et al., 2006). Fire also influences thermal regimes indirectly through altered vegetation interactions with the cryosphere (Brown et al., 2015; Fisher et al., 2016), and atmosphere (Bonan, 2008). Feng et al. (2016) found that the interaction between aerosols produced by forest fire and clouds may contribute to the reduced seasonality of the Pliocene High Arctic under lower aerosol, preindustrial boundary conditions. Given the sensitivity of arctic surface temperatures to sea-ice
extent during the Pliocene (Ballantyne et al., 2013), the effect of industrial black carbon on ice melting (McConnell et al., 2007), and the disagreement between model simulations and observations of surface temperatures at high latitudes during the Pliocene (Dowsett et al., 2012), the role of fire in amplifying temperature in past warm intervals deserves investigation.

Although it is generally thought that atmospheric $CO_2$ concentrations of ~ 400 ppm provided the dominant global radiative forcing during the mid-Pliocene, $CO_2$ proxies over the Pliocene do not necessarily agree (Fig. 1). Overall, reconstructions of
Pliocene $CO_2$ range between 190 and 440 ppm (Martinez-Boti et al., 2015; Seki et al., 2010). For example, Boron-based Pliocene $CO_2$ reconstructions tend to be lower than (258 ± 35 ppm; mean ± s.d.; Tripati et al., 2009), the alkenone-based isotopic proxy average 357 ± 45 ppm (Pagani et al., 2010; Seki et al., 2010), whereas other estimates suggest values between 330–400 ppm (Seki et al., 2010). While $CO_2$ estimates from stomata and paleosols tend to be less precise, they are within the range of boron and alkenone derived estimates (Royer, 2006). Thus there is no clear consensus on $CO_2$ concentrations in
Earth's atmosphere during the Pliocene. Dating uncertainties are an additional confounding factor, further complicating site to site comparisons. This suggests an additional hypothesis to explain why modern Arctic amplification and models agree, but the Pliocene amplification estimates and models disagree; that $CO_2$ levels were higher than currently used for model boundary conditions during the deposition of the proxies used for past temperature estimation at sites with high dating uncertainties. Although direct effects may be small (Feng et al., 2017), reconstructing the $CO_2$ from the same deposits from which
paleoclimate and paleoecological proxies are derived, may help reconcile previous estimates and contribute to constraining climate sensitivities during the Pliocene.

To advance understanding of arctic amplification during past warm intervals in Earth's history such as the Pliocene this investigation targets an exceptionally well-preserved arctic sedimentary sequence to simultaneously reconstruct atmospheric $CO_2$, summer temperature, vegetation and fire disturbance history from a single site over its deposition.



## 2 Methods

### 2.1 Site description

To investigate the environment and climate of the Pliocene Arctic we focused on a fossil site, Beaver pond (BP), located at 78° 33′ N (Fig. 2) on Ellesmere Island. The stratigraphic section located at ~380 meters above sea level (MASL) today includes
unconsolidated bedded sands and gravels, and rich organic layers including a thick fossil rich peat layer, up to 2.4 m thick, with sticks gnawed by an extinct beaver (*Dipoides spp.*). The assemblage of fossil plants and animals at BP has been studied extensively to gain insight into the past climate and ecology of the Canadian High Arctic (Ballantyne et al., 2006; Csank et al., 2011a; Csank et al., 2011b; Fletcher et al., 2017; Mitchell et al., 2016; Rybczynski et al., 2013; Tedford and Harington, 2003; Wang et al., 2017). Previous paleoenvironmental evidence suggests the main peat unit is a rich fen deposit with a neutral to
alkaline pH, associated with open water (Mitchell et al., 2016), likely a lake edge fen or shallow lake fen, within a larch-dominated forest-tundra environment (Matthews and Fyles, 2000), not a low pH peat-bog. While the larch species identified at the site, *Larix groenlandia*, is extinct (Matthews and Fyles, 2000), many other plant remains are Pliocene examples of taxa that are extant (Fletcher et al., 2017).

The fen-peat unit examined in this study was sampled in 2006 and 2010. The unit sampled spanned the 1 m remaining of
Unit III as per Mitchell et al. (2016). The main sequence examined across the methods used in this study includes material above (Unit IV) and below (Unit II) Unit III, with a total sampled profile of 1.65 m. Unit III has been estimated to represent ~20 000 years of deposition based on modern northern fen growth rates (Mitchell et al., 2016). The atmospheric $CO_2$ estimates from this locality were based on 22 sample layers from the 2006 field campaign, and the charcoal was based on 31, while the temperature estimates from specific bacterial membrane lipids were taken from 22 of the sample layers collected in 2006 and
an additional 12 samples collected in 2010. The same samples from the 2006 season were analyzed for each of $CO_2$, mean summer temperature and char count where contents of the sample allowed. Pollen was tabulated in 10 samples of these 2006, located at different stratigraphic depths.

### 2.2 Geochronology

While direct dating of the peat was not possible, we were able to establish a burial age for fluvial sediments deposited
approximately 4–5 m above and 30 m to the southwest of the peat. We used a method based on the ratio of isotopes produced in quartz by secondary cosmic rays. The cosmogenic nuclide burial dating approach measures the ratio of cosmogenic $^{26}Al$ ($t\frac{1}{2}$ = 0.71 Ma) and $^{10}Be$ ($t\frac{1}{2}$ = 1.38 Ma) in quartz sand grains that were exposed on hillslopes and alluvium prior to final deposition at BP. Once the quartz grains are completely shielded from cosmic rays, the ratio of the pair will predictably decrease because $^{26}Al$ has double the radiodecay rate of $^{10}Be$. In 2008, four of the medium to coarse grained quartz samples
were collected from a vertical profile of planar crossbedded fluvial sands between 8.7 and 10.4 m below the overlying till surface. The samples were 5 cm thick, separated by an average of 62 cm, and should closely date the peat (the sandy braided stream beds represent on the order of ~10⁴ years from the top of the peat to the highest sample). Quartz concentrates were



extracted from the arkosic sediment using Frantz magnetic separation, heavy liquids, and differential leaching with HF in ultrasonic baths. When sample aliquots reached aluminum concentrations <100 ppm (ICP-OES) as a proxy of feldspar abundance, the quartz concentrate was subjected to a series of HF digestion and rinsing steps to ensure that more than 30% of the quartz had been dissolved to remove meteoric $^{10}$Be. Approximately 200.00 mg of Be extracted from a Homestake Gold

Mine beryl-based carrier was added to 150.00 g of each quartz concentrate (no Al carrier was needed for these samples). Such large quartz masses were digested because of the uncertainty in the abundance of the faster decaying isotope. Following repeated perchloric-acid dry-downs to remove unreacted HF, pH-controlled precipitation, column chemistry ion chromatography to extract the Be and Al ions, precipitation in ultrapure ammonia gas, and calcination at temperatures above 1000°C in a Bunsen flame for three minutes, oxides were mixed with equal amounts of niobium and silver by volume. These

were packed into stainless steel targets for measurement at Lawrence Livermore National Laboratory's accelerator mass spectrometer (AMS). Uncertainty estimates for $^{26}$Al/$^{10}$Be were calculated as 1σ by combining AMS precision with geochemistry errors in quadrature. For a complete detailed description of TCN methods see Rybczynski et al. (2013). The ages provided here are updated from Rybczynski et al. (2013) by using more recent production rate information and considering the potential for increasing exposure to deeply penetrating muons during the natural post-burial exhumation at BP.

**2.3 Atmospheric CO₂ Reconstruction**

In order to reconstruct atmospheric CO₂ concentrations during the Pliocene, we derived a method based on the different sensitivity of isotopic discrimination of plant groups to their environment (Farquhar et al., 1989; Fletcher et al., 2008; White et al., 1994). Specifically, we used measurements of stable carbon isotopic discrimination in C3 vegetation to approximate the carbon isotopic signature of the atmosphere, and measurements of carbon isotopic discrimination in bryophytes to estimate the

partial pressure of atmospheric CO₂, which was then converted to atmospheric CO₂ concentration. According to theory (Farquhar et al., 1989), plants discriminate ($\Delta ^{13}$C) against the heavier isotope in atmospheric CO₂, such that:

$$\Delta^{13} C = a + (b - a) \frac{p_i}{P_a} \qquad (1)$$

Where $a$ is the fractionation of atmospheric CO₂ due to diffusion (~ -4.4 ‰) and $b$ is the fractionation of atmospheric CO₂ due to carboxylation by the enzyme rubisco (~ -27 ‰). This physical and chemical discrimination is then modulated by stomatal control of partial pressure of intercellular CO₂ – ($p_i$) with respect to the partial pressure of atmospheric CO₂ ($p_a$). Therefore isotopic discrimination in C3 plants ($\Delta ^{13}C_{C3}$) is largely a function of stomatal conductance and in bryophytes that lack stomata ($\Delta ^{13}C_{bryo}$) is largely a function of partial pressure in atmospheric CO₂ (i.e. $p_a$). While other environmental factors, such as

humidity, temperature, light availability, and microclimate also play important roles in isotopic discrimination in bryophytes (Fletcher et al., 2008; Ménot and Burns, 2001; Royles et al., 2014; Skrzypek et al., 2007; Waite and Sack, 2011; White et al., 1994), the first order control on discrimination is the partial pressure of atmospheric CO₂ (Fletcher et al., 2008; White et al.,





1994). Because atmospheric $CO_2$ is relatively well mixed in the troposphere its mean annual concentration does not significantly vary due to location. However, because total atmospheric pressure decreases with elevation the partial pressure of all atmospheric gases, including atmospheric $CO_2$, must also decrease with atmospheric height ($h$) according to the following exponential function:

$$p_{a(h)} = p_{a(i)}e^{-h/H} \tag{2}$$

Such that the partial pressure of atmospheric $CO_2$ at any given height in the atmosphere ($p_{a(h)}$) can be calculated based on the initial atmospheric partial pressure of atmospheric $CO_2$ ($p_{a(i)}$) and a reference height ($H = 7600$ m), where atmospheric pressure

goes to 0.37 Pa (Bonan, 2015). Therefore assuming that carbon isotopic discrimination in bryophytes varies in response to the atmospheric partial pressure of atmospheric $CO_2$ ($p_i / p_a \rightarrow p_a$), Eq. (2) can be substituted into Eq. (1), such that the natural logarithm of $\Delta^{13}C_{bryo}$ varies as a function of the partial pressure of atmospheric $CO_2$. Furthermore, if the assumptions of this empirical relationship are valid and time invariant, then this empirical relationship can in theory be used to predict the partial pressure of atmospheric $CO_2$ based on carbon isotopic measurements of bryophytes.

To test this prediction, we compiled data from four studies investigating carbon isotopic variability of different bryophytes, primarily mosses, along elevational transects at different locations. Based on the elevations, locations, and years in which these samples were collected, the atmospheric partial pressure of atmospheric $CO_2$ was estimated from ERA-interim reanalysis data of total atmospheric pressure (Dee et al., 2011) in conjunction with globally averaged values atmospheric $CO_2$ concentrations (Global View-$CO_2$, 2013). For our analysis we only included measurements of carbon isotopic variability in non-vascular

mosses and all isotopic values were normalized to cellulose based on the empirical relationship reported by Ménot and Burns (2001). Carbon isotopic discrimination values for all plant material was calculated as:

$$\Delta^{13}C = (\delta^{13}C_{atm} - \delta^{13}C_{plant})/(1 + \delta^{13}C_{plant}/1000) \tag{3}$$

where $\delta^{13}C_{plant}$ represents the C isotopic composition of plant cellulose and $\delta^{13}C_{atm}$ represents the mean annual carbon isotopic composition of atmospheric $CO_2$ of the year when samples were collected (Global View-$CO_2$, 2013), or in the case of sub-fossil mosses when the samples were growing.

In order to derive estimates of atmospheric $CO_2$ concentrations during the Pliocene, we first had to estimate the $\delta^{13}C$ of atmospheric $CO_2$ during the Pliocene to solve for $\Delta^{13}C$ of mosses (Eq.(2)). This was accomplished by simultaneous

measurements of $\delta^{13}C$ in the cellulose of sub-fossil plant buckbean (*Menyanthes trifoliata* L.) that was also found at the BP site. We also measured $\delta^{13}C$ of modern buckbean to constrain our estimates of $p_i / p_a$. For constraining our reconstructions of Pliocene $CO_2$, carbon isotopic measurements were made on sub-fossil mosses (*Scorpidium scorpoides* (Hedw.) Limpr.). All plant and moss material were rinsed in deionized water, and dried prior to cellulose extraction according Leavitt and Danzer (1993). All carbon isotopic measurements were performed at University of Arizona's environmental isotope laboratory.



## 2.4 Paleotemperature Reconstruction

Paleotemperature estimates were determined based on the distribution of fossilized, sedimentary membrane lipids known as branched glycerol dialkyl glycerol tetraethers (brGDGTs) that are well preserved in peat bogs, soils, and lakes (Powers et al., 2004; Weijers et al., 2007c). These unique lipids are thought to be synthesized by a wide array of Acidobacteria within the soil

(Sinninghe Damsté et al., 2014; Sinninghe Damsté et al., 2011). Previously, it has been established that the degree of methyl branching (expressed in the methylation index of branched tetraethers; MBT) is correlated with mean annual air temperature (MAT), and the relative amount of cyclopentane moieties (cyclization index of branched tetraethers; CBT) has been shown to correlate with both soil pH and mean annual air temperature (Weijers et al., 2007b). Because of the relationship of the distribution of these fossilized membrane lipids with these environmental parameters, it has been used for paleoclimate

applications in different environments including coastal marine sediments (Bendle et al., 2010; Weijers et al., 2007a), peats (Ballantyne et al., 2010; Naafs et al., 2017), paleosoils (Peterse et al., 2011; Zech et al., 2012), and lacustrine sediments (Loomis et al., 2012; Niemann et al., 2012; Pearson et al., 2011; Zink et al., 2010).

Improved separation methods (Hopmans et al., 2016) have recently led to the separation and quantification of the 5- and 6-methyl brGDGT isomers that used to be treated as one since the 6-methyl isomers were co-eluting with the 5-methyl isomers

(De Jonge et al., 2013). This has led to the definition of new indices and improved MAT calibrations based on the global soil (De Jonge et al., 2014), peat (Naafs et al., 2017), and African lake (Russell et al., 2018) datasets.

Sediment samples were freeze-dried and then ground and homogenized with a mortar and pestle. Next, using the Dionex™ accelerated solvent extraction (ASE), 0.5–1.0 g of sediment was extracted with the solvent mixture of dichloromethane (DCM):methanol (9:1, v/v) at a temperature of 100°C and a pressure of 1500 psi (5 min each) with 60% flush and purge 60 s.

The Caliper Turbovap®LV was utilized to concentrate the collected extract, which was then transferred using DCM and dried over anhydrous $Na_2SO_4$ before being concentrated again under a gentle stream of $N_2$ gas. To quantify the amount of GDGTs, 1 µg of an internal standard (C46 GDGT; Huguet et al., 2006) was added to the total lipid extract. Then, the total lipid extract was separated into three fractions using hexane:DCM (9:1, v:v) for the apolar fraction, hexane:DCM (1:1, v:v) for the ketone fraction and DCM:MeOH (1:1, v:v) for the polar fraction, using a column composed of $Al_2O_3$, which was activated for 2 h at

150°C. The polar fraction, which contained the GDGTs, was concentrated under a steady stream of $N_2$ gas before being then re-dissolved in hexane:isopropoanol (99:1, v:v) at a concentration of 10 mg ml$^{-1}$ and subsequently passed through a 0.45 µm PTFE filter. Finally, the polar fractions were analyzed for GDGTs on a high performance liquid chromatography – atmospheric pressure positive ion chemical ionization – mass spectrometry (UHPLC-APCI-MS) using the method described by (Hopmans et al., 2016). The polar fractions of some samples were re-run on the UHPLC-APCI-MS multiple times and in those cases the

fractional abundances of the brGDGTs were averaged and those values were used in the transfer functions. The overall estimate error of 2.03°C was determined by using the transfer function error (Ter = 2.0°C) and reproducibility error (Tre = 0.32°C), calculated as the average of the standard deviations from the duplicates.





For the calculation of brGDGT-based proxies, the brGDGTs are specified by the Roman numerals as indicated in Fig. S1. The 6-methyl brGDGTs are distinguished from the 5-methyl brGDGTs by a prime. The novel indices, including $MBT'_{5Me}$ based on just the 5-methyl brGDGTs and the CBT′ that was used to calculate the pH (De Jonge et al., 2014):

$$MBT'_{5Me} = (Ia + Ib + Ic) / (Ia + Ib + Ic + IIa + IIb + IIc + IIIa + IIIb + IIIc) \qquad (4)$$

$$CBT' = {}^{10}log[(Ic + IIa' + IIb' + IIc' + IIIa' + IIIb' + IIIc')/(Ia + IIa + IIIa)] \qquad (5)$$

Mean summer air temperature (MST) was determined using the distributions of aquatically produced brGDGTs in the calibration developed by Pearson et al. (2011). When this calibration is used the fractional abundances of IIa and IIa′ must be summed because these two isomers co-eluted under the chromatographic conditions used by Pearson et al. (2011):

$$MST\ (°C) = 20.9 + 98.1 \times [Ib] - 12 \times ([IIa] + [IIa']) - 20.5 \times [IIIa] \qquad (6)$$

The square brackets denote the fractional abundance of the brGDGT within the bracket relative to the total brGDGTs. Mean annual air temperature (MAT) and surface water pH were also calculated using a novel calibration created using sediments from East African lakes analysed with the novel chromatography method and based upon $MBT'_{5Me}$ (Russell et al., 2018).

$$MAAT = -1.2141 + 32.4223 * MBT'_{5Me} \qquad (7)$$

$$Surface\ water\ pH = 8.95 + 2.65 * CBT' \qquad (8)$$

**2.5 Vegetation and Fire Reconstruction**

For charcoal, a total of thirty 2 cm$^3$ samples were taken at 5 cm intervals from depths from 300 and 301.45 MASL at the BP site, with an additional 2cm$^{-3}$ sample collected at 301.65 MASL. All samples were deflocculated using sodium hexametaphosphate and passed through 500, 250 and 125 μm nested mesh sieves. The residual sample caught on each sieve was then collected in a gridded petri dish and examined using a stereomicroscope at 20-40X magnification to obtain charcoal concentration (fragments cm$^{-3}$). Charcoal area (mm$^2$ cm$^{-3}$) was measured for each sample using specialized imaging software from Scion Corporation. For a detailed description of methods see Brown and Power (2013).

Vegetation was reconstructed using pollen and spores (herein pollen) at selected elevations at an upper and lower elevation and that corresponded with changes in charcoal. Samples were processed using standard approaches (Moore et al., 1991), whereby 1cm$^3$ sediment subsamples were treated with 5% KOH to remove humic acids and break up the samples. Carbonates were dissolved using 10% HCl, whereas silicates and organics were removed by HF and acetolysis treatment, respectively. Pollen slides were made by homogenizing 35 μl of residue, measured using a single-channel pipette, with 15 μl of melted



glycerin jelly. Slides were counted using a Leica DM4000 B LED compound microscope at 400–630x magnification. A reference collection and published keys (McAndrews et al., 1973; Moore et al., 1991) aided identification.

In addition to tabulating pollen and charcoal, a list of taxa derived from Beaver Pond was previously compiled in Fletcher et al. (2017). Extant species from this list were selected and their modern observations extracted from The Global Biodiversity Information Facility (GBIF.org, 2017). Observation data was grouped by 5° latitude 5° longitude grids cells, and the shared species count calculated using R (R Core Team, 2016). Modern fire frequency was mapped using the MODIS 6 Active Fire Product. The fire pixel detection count per day, within the same 5° latitude 5° longitude grids cells was counted over the ten years 2006–2015, and standardized by area of the cell. The modern climate maps were generated using data from WorldClim 1.4 (Hijmans et al., 2005). The values for the bioclimatic variables mean temperature of the warmest quarter (equivalent to mean summer air temperature; MST) and precipitation of the warmest quarter (summer precipitation) were also averaged by grid cell. The shared species count, climate values, and fire day detections were mapped to the northern polar stereographic projection in ArcMap 10.1.

## 3 Results

### 3.1 Geochronology

The burial dating results with $^{26}Al/^{10}Be$ in quartz sand at 10 m below modern depth provides four individual ages. From shallowest to deepest, the burial ages are 3.6 +1.5/-0.5 Ma, 3.9 +3.7/-0.5 Ma, 4.1 +5.8/-0.4 Ma, and 4.0 +1.5/-0.4 Ma (Table S3), with an unweighted mean age of 3.9 Ma. The convoluted probability distribution function yields a maximum probability age of 4.5 Ma. The optimized ages for the top and bottom sample were 3.6 and 4.0 Ma, in agreement with the individual most probable ages. Optimized ages could not be computed for the two middle samples owing to the large uncertainties in $^{26}Al/^{27}Al$ measurement which caused the uncertainties to extend beyond the maximum saturation burial ages (ca. 8 Ma). Despite the apparent upward younging of the burial ages, the 1σ-uncertainties overlap rendering the samples indistinguishable. Given the asymmetry in the probability distribution functions, and the inability to convolve all samples, the unweighted mean age is 3.9 Ma, with an uncertainty of +1.5/-0.5 Ma as indicated by the two samples with unsaturated limits. The age of the Beaver Pond peat is stratigraphically younger, however considering time for lateral channel migration and aggradation on the contemporaneous braid plain, the peat age is likely older by 104 to 105 years, i.e. within the uncertainty of the mean burial date.

### 3.2 Atmospheric CO₂ Reconstruction

As expected carbon isotopic discrimination in mosses shows a positive relationship with partial pressure of atmospheric $CO_2$ (Fig. 3) and, as predicted from theory, the natural logarithm of carbon isotopic discrimination in mosses ($\Delta^{13}C_{moss}$) is responsive to $p_a$. Consistent with the exponential relationship between elevation and atmospheric pressure (Eq. 2) the relationship is non-linear with a greater change in $p_a$ and thus a greater change in $\Delta^{13}C_{moss}$ at lower elevations. Despite fitting our model to Ln($\Delta$



$^{13}C_{moss}$), we found a slightly better fit based on a second order polymomial (RMSE = 1.096 ‰) than a linear model (RMSE = 1.097 ‰), suggesting that other non-linear processes and not just $p_a$ may be affecting $\delta^{13}C_{moss}$ variability with elevation.

While there does appear to be a global relationship between pa and $\Delta^{13}C$ of mosses, there are notable differences among sites. Moss $\Delta^{13}C$ values tended to be generally lower in the Swiss Alps (mean = 17.4 ‰) and higher in Hawaii (mean = 20.6 ‰) and the slope of the relationship between $p_a$ and $\Delta^{13}C$ appears to vary across sites with the Andes having the smallest slope and Poland having a much greater slope. While these subtle differences are probably due to local climate affects, it appears that $p_a$ is the primary physical mechanism explaining the previously reported global relationship between $\Delta^{13}C$ of mosses and elevation globally across all sites (Ménot and Burns, 2001; Royles et al., 2014; Skrzypek et al., 2007; Waite and Sack, 2011). We also evaluated model performance using a global standard atmospheric sea level pressure of 101.325 kPa, or site specific atmospheric pressure estimates from ERA-interim reanalysis data. We found that the model using site specific atmospheric pressure estimates performed better at predicting $\Delta^{13}C_{moss}$ (RMSE = 1.096 ‰) than the model using global standard atmospheric sea level pressure (RMSE = 1.216 ‰). Thus the optimal model characterizing the observed modern relationship between $\Delta^{13}C_{moss}$ and the $p_a$ was the second order polynomial:

$$Ln\, \Delta^{13}C = 0.0021 \times pCO_2{}^2 - 0.0991 \times pCO2 + 4.0107 \tag{9}$$

This polynomial was solved numerically to derive estimates of $pCO_2$ during the Pliocene based on sub-fossil moss samples collected at the BP site.

Based on our analysis of cellulose extracted from four different *Menyanthes* L. (buckbean) plants growing at different locations in the modern boreal forest, we found $\Delta^{13}C$ of buckbean to be fairly constant 16 ± 0.4 ‰, yielding an estimate of $p_i / p_a$ in modern buckbean of 0.51. Applying this modern of $p_i / p_a$ to our $\delta^{13}C$ measurements from sub-fossil buckbean we obtained estimates of $\delta^{13}C_{atm}$ during the Pliocene of -6.23 ± 0.9 ‰. Using our empirical transfer function (Eq. 9) in combination with these estimates $\delta^{13}C_{atm}$, allowed us to approximate atmospheric $CO_2$ concentrations over the Pliocene interval captured at the BP site (Fig. 4). We estimated a mean atmospheric $CO_2$ concentration over this interval of 440 ± 50 ppm with considerable variability between a minimum atmospheric $CO_2$ concentration of 270 ppm and a maximum atmospheric $CO_2$ concentration of 470 ppm. Overall, this proxy approach had a prediction uncertainty of less than 10% of the estimate (1 σ = 35 ppm).

### 3.3 Paleotemperature Estimates

### 3.3.1 Provenance of branched GDGTs

Previously, brGDGT derived MAT estimates (-0.6 ± 5.0 °C) from BP sediments were developed using the older chromatography methods that did not separate the 5- and 6- methyl brGDGTs, and a soil calibration (Ballantyne et al., 2010). In marine and lacustrine sediments, bacterial brGDGTs were thought to originate predominantly from continental soil erosion arriving in the sediments through terrestrial runoff, however, a number of more recent studies have indicated aquatically




produced brGDGTs could be affecting the distribution of the sedimentary brGDGTs and thus the temperature estimates based upon them (Warden et al., 2016; Zell et al., 2013; Zhu et al., 2011). Since the discovery that sedimentary brGDGTs can have varying sources, different calibrations have been developed depending on the origin of the brGDGTs, i.e. soil calibration (De Jonge et al., 2014), peat calibration (Naafs et al., 2017) and aquatic calibrations (i.e. Foster et al., 2016; Pearson et al., 2011;

Russell et al., 2018). Therefore, several studies have recommended that the potential sources of the sedimentary brGDGTs should be investigated before attempting to use brGDGTs for paleoclimate applications (De Jonge et al., 2015; Warden et al., 2016; Yang et al., 2013; Zell et al., 2013). In this study, we examine the distribution of brGDGTs in an attempt to determine their origin and consequently the most appropriate calibration to utilize in order to reconstruct temperatures from the BP sediments.

Branched GDGTs IIIa and IIIa′ on average had the highest fractional abundance of the brGDGTs detected in the BP sediments (see Fig. S1 for structures; Table S1). A previous study established that when plotted in a ternary diagram the fractional abundances of the tetra-, penta- and hexamethylated brGDGTs, soils lie within a distinct area (Sinninghe Damsté, 2016). To assess whether the brGDGTs in the BP deposit were predominantly derived from soils, we compared the fractional abundances of the tetra-, penta- and hexamethylated brGDGTs in the BP sediments to those from modern datasets in a ternary

diagram (Fig. 6). Since the contribution of brGDGTs from either peat or aquatic production could affect the use of brGDGTs for paleoclimate application, in addition to comparing the samples to the global soil dataset (De Jonge et al., 2014), peat and lacustrine sediment samples were added into the ternary plot to help elucidate the provenance of brGDGTs in the BP sediments. According to Sinninghe Damsté (2016), it is imperative to only compare samples in a ternary diagram like this where all of the datasets were analyzed with the novel methods that separate the 5- and 6-methyl brGDGTs since the improved separation

can result in an increased abundance of hexamethylated brGDGTs. Recently, samples from East African lake sediments were analyzed using these new methods (Russell et al., 2018) and so these samples were included in the ternary plot for comparison (Fig. 6). Although the lakes from the East African dataset are all from a tropical area, they vary widely in altitude and, thus, in MAT. We separated them into three categories by MAT (lakes >20°C, lakes between 10-20°C and lakes<10°C). By comparing all the samples in the ternary plot, it was evident that the BP samples plotted closest to the lacustrine sediment samples from

regions in East Africa with a MAT <10°C, suggesting that the provenance of the majority of the brGDGTs from the BP sediments was not soil or peat but lacustrine aquatic production.

The average estimated surface water pH for the BP sediments (8.6±0.2) calculated using eq. (8), is within the 6−9 range typical of lakes and rivers (Mattson, 1999). This value is near the upper limit of rich fens characterized by the presence of *S. scorpioides* (Kooijman and Westhoff, 1995; Kooijman and Paulissen, 2006) and is higher than what would be expected for

peat-bog sediments that are acidic (pH 3–6; Clymo, 1964) and which constitute most of the peats studied by Naafs et al. (2017). A predominant origin from lake aquatic production is in keeping with previous interpretation of the paleoenvironment of the BP site, which was at least at times covered by water as evidenced by fresh water diatoms, fish remains and gnawed beaver sticks in the sediment (Mitchell et al., 2016).



### 3.3.2 Aquatic Temperature Transfer Function

Since there is evidence that the majority of the brGDGTs in the BP sediments are aquatically produced, an aquatic transfer function was used for reconstructing temperature. When we apply the African lake calibration (Eq. 7), the resulting estimated MAT for BP is 7.1 ± 1.0 °C. This value is high compared to other previously published estimates from varying proxies, which

have estimated MAT in this region to be in the range of -5.5 to 0.8°C, (Ballantyne et al., 2010; Ballantyne et al., 2006; Csank et al., 2011a; Csank et al., 2011b; Fletcher et al., 2017). A concern when applying this calibration is that it is based on lakes from an equatorial region that does not experience substantial seasonality, whereas, the Pliocene Arctic BP site did experience substantial seasonality (Fletcher et al., 2017). Biological production (including brGDGT production) in BP was likely skewed towards summer and, therefore, summer temperature has a larger influence on the reconstructed MAT. Unfortunately, no

global lake calibration set using individually quantified 5- and 6-methyl brGDGTs is yet available. Therefore, to calculate mean summer air temperatures (MST, Eq. 6) we applied the aquatic transfer function developed by Pearson et al. (2011) by combining the individual fractional abundances of the 5- and 6-methyl brGDGTs. The Pearson et al. (2011) calibration was based on a global suite of lake sediments including samples from the Arctic, thus covering a greater range of seasonal variability. The resulting average estimated mean summer temperature was 15.4 ± 0.8 °C, with temperatures ranging between

14.1 and 17.4 °C (Fig. 4). This is in good agreement with recent estimates based on Climate Reconstruction Analysis using Coexistence Likelihood Estimation (CRACLE; Fletcher et al., 2017) that concluded that MSTs at BP during the Pliocene were approximately 13 to 15°C.

### 3.4 Vegetation and Fire Reconstruction

All sediment samples from BP contained charcoal (Fig. 4), indicating the consistent prevalence of biomass burning in the High

Arctic during this time period. However, counts were variable throughout the section, with the middle and lower sections (18 fragments cm$^{-3}$) containing less charcoal compared to the upper section upper section (710 fragments cm$^{-3}$). Overall, samples from BP contained on average 100.0 ± 165 fragments cm$^{-3}$ (mean ± 1 σ), with charcoal area averaging 12.3 ± 20.2 mm$^2$ cm$^{-3}$. The variability of charcoal within any given sample was relatively low with a 1 σ among charcoal area of approximately 2 mm$^2$ cm$^{-3}$.

The three sections analysed for pollen reveal variations in vegetation (Figs. 4 and 5). Near the bottom of the section (300.3-300.4 MASL), *Larix* (26%) and *Betula* (17%) were the dominant trees. *Alnus* (6%) and *Salix* (6%) together with ericaceous pollen (4%) were relatively high. In contrast, low numbers of *Picea* (3%), *Pinus* (3%) and fern spores were recorded. Additional wetland taxa like *Myrica* (5%) and Cyperaceae (6%) were also noted. Overall, the non-arboreal (23%) signal was well developed. Crumpled and/or ruptured inaperturate grains with surface sculpturing that varied from scabrate to verricate

were noted in the assemblage (12%), but could not be definitely identified. It is possible that these grains represent *Populus*, Cupressaceae or additional Cyperaceae pollen. Between 301.15-301.25 MASL, *Larix* (38%) and *Betula* (21%) increased in abundance, followed by ferns (7%). Cyperaceae remained at similar levels (6%) whereas *Picea* and *Pinus* decreased to 2%

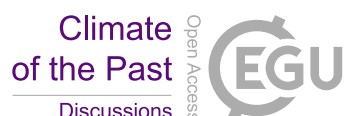

and 1%, respectively. Unidentified inaperturate types collectively averaged 14%. *Larix* pollen (23%) remained abundant near the top of the section (301.35-301.45 MASL), whereas *Betula* (2%) decreased. *Picea* (16%) *Pinus* (6%) and ferns (23%) increased in abundance. Of the ferns, trilete spores and cf. *Botrychium* were most abundant, followed by cf. *Dryopteris*. Inaperturate unknowns (10%) were also observed. Other notables included Ericaceae (2%) and Cyperaceae (2%). While rare, Onagraceae grains were also observed (Fig. 5).

According to GBIF-based mapping exercise, the paleofloral assemblage at BP most closely resembles modern day vegetation found in northern North America, particularly on the eastern margin (e.g. New Hampshire, New Brunswick and Nova Scotia) and the western margin (Alaska, Washington, British Columbia, and Alberta; Fig. 7a), and central Fennoscandia. Of these areas, the western coast of northern North America and eastern coast of southern Sweden has the most similarity to the reconstructed BP climate in terms of MST (Fig. 7b) and summer precipitation (Fig. 7c).

While high counts of active fire days are common in the western part of the North American boreal forest, it is not as common in the eastern part of the North American boreal forest (Fig. 7d), likely due to the differences in the precipitation regime. There was also low fire counts in Fennoscandia likely due to historical severe fire suppression (Brown and Giesecke, 2014; Niklasson and Granström, 2004). Therefore, based on our reconstruction of the climate and ecology of the BP site, our results suggest that BP most closely resembled a boreal-type forest ecosystem shaped by fire, similar to those of Washington, British Columbia, Northwest Territories, Yukon and Alaska (but see Sect. 4.3).

## 4 DISCUSSION

### 4.1 Geochronology

The plant and animal fossil assemblages observed at BP suggest a depositional age between 3 and 5 Ma (Matthews Jr and Ovenden, 1990; Tedford and Harington, 2003). This biostratigraphic age was corroborated with an amino-acid racemization age ($>2.4 \pm 0.5$ Ma) and Sr-correlation age (2.8–5.1 Ma) on shells (Brigham-Grette and Carter, 1992) in biostratigraphically correlated sediments on Meighen Island, situated 375 km to the west-north-west. The previously calculated burial age of 3.4 Ma for the BP site isa minimum age because no post-depositional production of $^{26}$Al or $^{10}$Be by muons was assumed. If the samples are considered to have been buried at only the current depth (ca. 10 m, see supplemental data) then the ages plot to the left and outside of the burial field, indicating that the burial depth was significantly deeper for most of the post-depositional history. The revised cosmogenic nuclide burial age is 3.9 +1.5/-0.5 Ma. It is the best interpretation of burial age data based on improved production rate systematics (e.g. Lifton et al., 2014), and more reasonable estimates of erosion rate and ice cover since the mid-Pliocene (see Table S4). As the stratigraphic position of the cosmogenic samples is very close to the BP peat layers, we interpret the age to represent the approximate time that the peat was deposited.



## 4.2 Pliocene atmospheric $CO_2$ levels

We have derived a transfer function that allows us to predict the partial pressure of atmospheric $CO_2$ in Earths' past based on carbon isotopic measurements in byrophytes. However, many of the studies included in our transfer function identify other mechanisms that may also influence carbon isotopic discrimination in bryophytes. Because these other mechanisms may
violate the assumptions of applying this transfer function to the past or contribute error to our reconstructions of atmospheric $CO_2$ concentrations during the Pliocene, we discuss these mechanisms below.

It has been suggested that in the absence of stomatal regulation, that surface water may control the gradient in partial pressure (i.e. $p_i / p_a$ ) in bryophytes (White et al., 1994), due to the greater resistance to diffusion of $CO_2$ in water than in the atmosphere. For instance, Ménot and Burns (2001) found that most mosses growing along an elevational transect in Switzerland
experienced discrimination with elevation in response to decreased partial pressure, except one species *Sphagnum cuspidatum* Ehrh. ex Hoffm., which grows almost exclusively in wet hollows. In a study of Hawaiian bryophytes Waite and Sack (2011) found consistent slopes of less isotopic discrimination with elevation in all species, however, species growing on young substrate showed significantly less isotopic discrimination. The most likely explanation is that lack of canopy cover on the older substrates lead to greater photosynthetic rates, which lead to reduced $p_i$. Lastly, decreased discrimination of mosses
growing along an elevational transect in Poland (Skrzypek et al., 2007), was found to be highly correlated with temperature. Although temperature is the primary factor driving most metabolic reactions, it does not provide a physical mechanism explaining the relationship between elevation and isotopic discrimination in mosses. Skrzypek et al. (2007) found slightly different relationships between elevation and carbon isotopic discrimination in mosses growing on the windward versus leeward side of their elevational transects suggesting that changes in lapse rate may also play a factor. Collectively, these
studies suggest that microclimatic factors may explain differences in isotopic discrimination of mosses within and among different sites possibly contributing to different intercepts for sites reported in Fig. 3, and that dry vs. moist lapse rates may also play a role in regulating the different slopes among sites. In fact, the greatest elevational range reported among sites was for the elevational transect in the Andes (320 to 3100 m), but this site did not experience the widest range in $\Delta^{13}C_{moss}$. This tropical transect had a very moist lapse rate resulting in the least change in atmospheric temperature and humidity with
elevation. Nonetheless, by projecting these data as a function of partial pressure we provide a physical mechanism to explain variations in moss carbon isotopic values globally and we help reconcile the previously reported empirical relationships, such as elevation, temperature, and over-story, all of which tend to be covariates of decreasing partial pressure with elevation. While differences in microclimate and lapse rate are clearly important factors in regulating $\Delta^{13}C_{moss}$, these factors contribute to the global error in our model for predicting $p_a$ and ultimately to uncertainties in our estimates of atmospheric $CO_2$ concentrations
during the Pliocene.

Our reconstructions of $CO_2$ concentration for this mid-Pliocene interval are within the range of previously reported $CO_2$ estimates, tending to agree with alkenone estimates from Pagani et al. (2010). This suggests that $CO_2$ concentrations during this warm Pliocene interval were above 400 ppm. In fact our mean Pliocene value (440 ± 50 ppm) is not statistically different





from the alkenone based estimates (357 ± 47 ppm) previously reported by Pagani et al. (2010). Generally, our estimates showed sustained atmospheric $CO_2$ estimates of approximately 450 ppm with only four anomalously low values (Fig. 4). These estimates could represent an actual reduction in atmospheric $CO_2$, or they might be artefacts of sampling or analysis. It should be noted that poor preservation and a possible shift in dominant moss species to *Drepanocladius spp.* was evident in samples

corresponding to these two anomalously low $CO_2$ estimates. While one of these samples contained only 0.17 mg/C and a $\delta^{13}$C value of -20.9 ‰, the other contained 0.88 mg/C and a $\delta^{13}$C value of -25.0 ‰. Thus it is conceivable that the sample corresponding to the atmospheric $CO_2$ estimate of 270 ppm, might be approaching our minimum detection limit and should be verified in subsequent studies. Overall, these $CO_2$ estimates are consistent with recent estimates derived from both alkenones and from boron isotopes (Martinez-Boti et al., 2015; Seki et al., 2010).

It should also be noted that changes in growth rate due to phosphorus availability and biases in shell size are known to contribute uncertainty to alkenone-derived $CO_2$ concentration estimates (Seki et al., 2010). Similar assumptions may affect boron-derived estimates of $CO_2$ concentrations. For instance, a recent update on the global boron cycle estimates the mean residence time of boron to be ~ 1.5 Ma and suggests that boron isotopes may not be sensitive to ocean pH on timescales less than 1 Ma (Schlesinger and Vengosh, 2016). This may help explain the apparent lack of variability in boron isotope and

boron/calcium based $CO_2$ estimates during the Pliocene (Hönisch et al., 2009; Tripati et al., 2009); however, boron isotopes do seem to reproduce the $CO_2$ variability measured in ice cores over the Pleistocene (Hönisch et al., 2009). Our atmospheric $CO_2$ estimates are in the range of previous estimates, although the values seem slightly high and are variable, suggesting that these estimates probably represent atmospheric conditions during a Pliocene warm interval and not necessarily an integrated average over the entire Pliocene.

There are numerous assumptions based on known uncertainties in our $CO_2$ reconstruction approach. First of all, our empirically based approach requires some estimate of the isotopic ratio of atmospheric $CO_2$ during this time, which we derive from C3 vegetation (Fletcher et al., 2008; White et al., 1994). Here we estimate the isotopic composition of the atmosphere over the Pliocene to be $\delta^{13}$C = -6.23 ± 0.9 ‰, which is within the range of values recorded over glacial-interglacial intervals in ice cores $\delta^{13}$C = -6.2 to -7.0 ‰ (Bauska et al., 2016) and consistent with estimates derived from carbon isotope measurements

of foraminifera (Ravelo et al., 2004). If we assume that the isotopic composition of atmospheric $CO_2$ was -8.2 ‰ during the Pliocene and similar to today due to greater transfer of lighter carbon from the terrestrial reservoir to the atmospheric reservoir, that would result in reduced $\Delta^{13}C_{moss}$ and decreases in our mean estimate of atmospheric $CO_2$ to approximately 420 ppm. This adjustment to our original estimate of $\delta 13$C of atmospheric $CO_2$ would bring our atmospheric $CO_2$ estimate more in line with previous reconstructions, but is still within the range of error of our original estimate.

Another critical assumption of our approach is that the total pressure of the atmosphere has not changed at the BP site since the Pliocene either through increased partial pressure of constituent gases or more likely through changes in elevation due to dynamic isostacy. The current elevation of the site is approximately 380 MASL with a summertime total atmospheric pressure of approximate 88.5 kPa. If we assume that the site was at 0 m during the Pliocene that would increase the total summertime atmospheric pressure to 93.9 kPa and would decrease our Pliocene $CO_2$ estimates to about 420 ppm. However, estimates of



dynamic eustacy since the Pliocene from paleoshorelines at lower latitudes are between 5 and 20 m (Rovere et al., 2014), suggesting that our assumptions regarding elevation at the site probably have a negligible impact on our estimates of Pliocene atmospheric $CO_2$ concentrations, especially given the uncertainty of the proxy approach. Therefore the assumptions to our approach in estimating past $CO_2$ may be leading to estimates that are biased slightly high relative to previous estimates. When these assumptions are considered, our estimates still suggest atmospheric $CO_2$ concentrations exceeding 400 ppm during this Pliocene warm interval.

## 4.3 Fire, vegetation, climate

The vegetation reconstruction indicates that open *Larix-Betula* parkland persisted in the basal (300.3-300.4 MASL) parts of the sequence. Groundcover was additionally dominated by shrub birch, ericaceous heath and ferns. While the regional climate may have been somewhat dry, the record suggests that, locally, a moist fen environment dominated by Cyperaceae, existed near the sampling location. Shrubs including *Alnus* and *Salix* likely occupied the wetland margins.

The corresponding relatively low concentration of charcoal may reflect lower severity fires or higher sedimentation rates. If the former, it is posited that a surface fire regime existed. This premise is supported by the fire ecology characteristics of the dominant vegetation. *Larix* does not support crown fires due to leaf moisture content (de Groot et al., 2013) and self-pruning (Kobayashi et al., 2007). The persistence and success of larch in modern-day Siberia appears to be driven by its high growth rate (Jacquelyn et al., 2017) tolerance of frequent surface fire due to thick lower bark (Kobayashi et al., 2007) and tolerance of spring drought due to its deciduous habit (Berg and Chapin III, 1994). Arboreal *Betula* are very intolerant of fire and easily girdled. However, they are quick to resprout and are often found in areas with short fire return intervals. Like *Larix*, arboreal *Betula* have high moisture content of their foliage and are not prone to crown fires. *Betula* nana L., an extant dwarf birch, is a fire endurer that resprouts from underground rhizomes or roots (Racine et al., 1987) thus regenerating quickly following lower severity fires (de Groot et al., 1997). The vegetation and fire regime characteristics are similar further up the sequence at 301.15-301.25 MASL, with the exception that ferns increased in abundance while heath decreased.

In the upper part of the sequence (301.35-301.45 MASL), where charcoal was abundant, the *Larix-Betula* parkland was replaced by a mixed boreal forest assemblage with a fern understory. Canopy cover was more closed compared to the preceding intervals. The forest was dominated by *Larix* and *Picea*, with lesser amounts of *Pinus*. While *Betula* remained part of the forest, it decreased in abundance possibly due to increased competition with the conifers. Based on exploratory CRACLE analyses of climate preferences using GBIF occurrence data (GBIF.org, 2018a, b, c, d) of the dominant taxa (*Larix-Betula* vs. *Larix-Picea-Pinus*), the expansion of conifers could indicate slightly warmer summers (MST ~15.8 °C vs. 17.1 °C). This result is in contrast to the stable MST estimated by bacterial tetraethers, although within reported error, and the small change is certainly within the climate distributions of both communities. The analyses also suggests that slightly drier conditions may have prevailed during the three wettest months (249-285mm vs. 192-219mm). While the interaction between climate, vegetation and fire is complex, the aforementioned changes in climate could have directly altered both the vegetation and fire regime, which in turn further promoted fire adapted taxa. In addition to regional climatic factors, community change at the site



may have been further influenced by local hydrological conditions, such as channel migration, pond infilling and ecosystem engineering by beaver (*Cantor spp.*).

The high charcoal content suggests that fire was an important disturbance mechanism, although it could also reflect a slow sedimentation rate. If the former, it is likely that frequent, mixed severity fires persisted. While *Larix* is associated with surface

fire, *Picea* and *Pinus* are adapted to higher intensity crown fires. A crown fire regime may have established as conifers expanded, altering fuel loads and flammability. For example, black spruce sheds highly flammable needles and its lower branches can act as fuel ladders facilitating crown fires (Kasischke et al., 2008), and was previously tentatively identified at BP (Fletcher et al., 2017). While it has thin bark and shallow roots maladapted to survive fire (Auclair, 1985; Brown, 2008; Kasischke et al., 2008), it releases large numbers of seeds from semi-serotinous cones, leading to rapid re-establishment (Côté

et al., 2003). The documentation of Onagraceae pollen at the top of the sequence could potentially reflect post-fire succession. For example, the species *Epilobium angustifolium* L. is an early-seral colonizer of disturbed (i.e. burnt) sites, pollinated by insects.

It is possible that the *Larix-Betula* parkland dominated intervals correspond to the peat- and sand-stratigraphic Units II and III described by Mitchell et al. (2016), whereas the mixed boreal forest in the upper part of the sequence is contemporaneous

with Unit IV, described as peat and peaty sand, coarsening upwards. While it is clear that the vegetation and fire regimes changed through time at this Arctic site, $CO_2$ and temperatures appear more stable, or at least to have no apparent trend. Thus, it is suggested that the fire regime at BP was primarily regulated by regional climate and vegetation, and perhaps additionally by changing local hydrological conditions. Regarding climate, MST remained high enough throughout the sequence to allow for fire disturbance and the pollen suggests that temperatures may have marginally increased in the upper part of the sequence.

Alternatively, other climate variables, such as the precipitation regime, or local hydrological change may have initiated the change in community. Up-sequence changes in vegetation undoubtedly influenced fine fuel loads and flammability. Indeed, the fire ecological characteristics of the vegetation are consistent with a regional surface fire regime yielding to a crown fire regime.

*Betula* and *Alnus*, which occurred earlier in the depositional sequence, are favored by beaver in foraging (Busher, 1996;

Haarberg and Rosell, 2006; Jenkins, 1979). Moreover, the presence of sticks cut by beaver in Unit III reveals that beavers were indeed at the site, moistening the local land surface. The lack of beaver cut sticks and changes in sediment in Unit IV may indicate that the beavers abandoned the site, possibly in response to changes in vegetation (i.e. increased conifers and decreased *Betula*) limiting preferred forage or due to lateral channel migration, as evidenced by the coarsening upward sequence described by Mitchell et al. (2016). As a result, the local land surface may have become somewhat drier, contemporaneous

with the change towards *Larix-Picea-Pinus* forest and a mixed severity fire regime.

Matthews and Fyles (2000) similarly indicated that the Pliocene BP environment was characterized by an open larch dominated forest-tundra environment, sharing most species in common with those now found in three regions, including central Alaska to Washington in western North America, the region centered around the Canadian/US border in eastern North America, as well as Fennoscandia in Europe (Fig 7a). Wildfire is a key driver of ecological processes in modern boreal forests



(Flannigan et al., 2009; Ryan, 2002), and although historically rare, is becoming more frequent in the tundra in recent years (Mack et al., 2011). The modern increase in fire frequency is likely as a consequence of atmospheric $CO_2$ driven climate warming and feedbacks such as reduced sea ice extent (Hu et al., 2010), because the probability of fire is highest where temperature and moisture are conducive to growth and drying of fuels followed by conditions that favor ignition (Whitman et al., 2015). Young et al. (2017) confirmed the importance of summer warmth and moisture availability patterns in predicting fire across Alaska, highlighting a July temperature of ~13.5 °C as a key threshold for fire across Alaska.

The abundance of charcoal at BP demonstrates that climatic conditions were conducive for ignition and that sufficient biomass available for combustion existed across the landscape. Mean summer temperatures at BP likely exceeded the ~13.5 °C threshold (Young et al., 2017) that drastically increases the chance of wildfire as demonstrated here from brGDGT derived temperatures and corroborated by previous studies with a seasonal component (Csank et al., 2011b; Fletcher et al., 2017). An increase in atmospheric convection has been simulated in response to diminished sea-ice during warmer intervals (Abbot and Tziperman, 2008), but this study did not confirm if this increase in atmospheric convection was sufficient to cause lightning ignitions. An alternative ignition source for combustion of biomass on Ellesmere Island during the Pliocene is coal seam fires, which have been documented to be burning at this time (Estrada et al., 2009). However, given the interaction of summer warmth and ignition by lightning within the same climate range as posited for BP, we consider lightning the most likely source of ignition for Pliocene fires in the High Arctic.

Fire return intervals cannot be calculated from the BP charcoal counts due to the absence of a satisfactory age-depth model and discontinuous sampling. As strong interactions are observed between fire regime and ecosystem assemblage in the boreal forest (Brown and Giesecke, 2014; Kasischke and Turetsky, 2006), and in response to climate, comparison with modern fire regimes for areas with shared species compositions and climates may inform a potential range of mean fire return interval (MFRI).

The modern area with the most species in common with BP is central northern Alaska (Fig. 7A). The area over which shared species were calculated is largely tundra, but includes the ecotone between tundra and boreal forest. Other zones that share many species with BP are continuous with Alaska down the western coast of North America to the region around the border of Canada and the United States, the eastern coast of North America in the region around the border of Canada and the United States (~50°N), and central Fennoscandia. Of these zones, the MST of Alaskan tundra sites (6–9°C) are less similar to BP (15.4°C) than ~50°N on both western and eastern coastal North American sites and central Fennoscandia (12–18°C, Fig. 7B). The eastern coast of North America has higher rainfall during the summer (>270 mm), than the west coast and Alaska (Fig. 7C), which correlates to the timing of western fires. The low summer precipitation for much of the west (<200 mm), is consistent with previously published summer precipitation estimates for BP (~190 mm). As a result, the fire regime of the west coast ~50°N may be a better analogue for BP than the east coast of North America. In central Fennoscandia there is also a west vs. east coastal variation in summer precipitation with the western, Nordic part of the region experiencing higher summer precipitation (252– >288 mm), than the more similar eastern, Swedish part of the region (~198 mm).



Comparison to modern fire detection data (Fig. 7D) suggests that the two regions most climatically similar to BP, ~50°N western North America and central Sweden, have radically different fire regimes. This is likely caused by historical fire suppression in Sweden that limits utility of very modern data for comparison in this study (Brown and Giesecke, 2014; Niklasson and Granström, 2004). To understand the fire regimes as shaped by climate and species composition rather than human impacts, we considered both the modern and recent Holocene reconstructions for these regions (Table 1). This shows that, a) within any region variation arises from the complex spatial patterning of fire across landscapes, and b) that the regions most similar to BP (~50°N western North American and eastern Fennoscandian reconstructions for the recent Holocene) have shorter fire return intervals than the cooler Alaskan tundra or wetter summer ~50°N eastern North American coast.

While the shared species for Siberia appears low, the number of observations in the modern biodiversity database used is likewise low – perhaps causatively so. Given the similar climate to BP on the Central Siberian Plateau and some key aspects of the floras in Siberia such as the dominance of larch, we considered the fire regime of the larch forests of Siberia. Kharuk et al. (2016; 2011) studied MFRIs across Siberia, from 64°N to 71°N, the northern limit of larch stands. They found an average MFRI across that range of 110 years, with MFRI increasing from 80 years in the southern latitudes to ~300 in the north (Table 1). Based on similarity of the climate variables, the more southerly MFRIs (~80 years) may be a better analogue. Key differences between boreal fires in the North America compared to Russia are a higher fire frequency with more burned area in Russia, but a much lower crown fire and a difference in timing of disturbance, with spring fires prevailing in Russia compared to mid-summer fires in western Canada (de Groot et al., 2013; Rogers et al., 2015).

Critically, the charcoal record reveals that biomass burning could have been a potential feedback mechanism amplifying or dampening warming during the Pliocene due to its prevalence through time, and the complex direct impacts on the surface radiative budget and direct and indirect effects on the top of the atmosphere radiative budget (Feng et al., 2016). Further investigation is warranted to better characterize the fire regime to improve accuracy of fire simulations in earth system models of Pliocene climate.

## 5. CONCLUSION

The record of high $CO_2$ supports the hypothesis that Pliocene Arctic terrestrial fossil localities probably represent periods of higher warmth that supported higher productivity. The novel temperature estimates presented here suggest that summer temperatures were considerably warmer during the Pliocene (~15.4°C) compared to modern day Eureka, Canada (~4.1°C; Fig. 2). This highlights the increasing influence of arctic amplification of temperatures as $CO_2$ exceeds modern levels. Our reconstruction of the paleovegetation and ecology of this unique site on Ellesmere Island suggests an assemblage similar to forests of the western margins of North America and eastern Fennoscandia. The evidence of recurrent fire and concurrent changes in taxonomic composition suggests that fire played an active role in Pliocene forests, shaping the environment and influencing the climate of the Arctic during the Pliocene. The importance of fire in the modern boreal forest suggests that fire may have had direct and indirect impacts on Earth's radiative budget at high latitudes during the Pliocene, although the net



impact of the component process remains unknown. Alterations to Earth's surface and atmospheric radiative budget as a result of fire may help reconcile the gap between high latitude temperature estimates observed from proxies and simulated from models as the impact of these processes are better characterized. Collectively, these reconstructions provide new insights into the paleoclimatology and paleoecology of the Canadian High Arctic, ~3.9 Ma. Our results further support that near future

climate forcing from $CO_2$ concentrations not experienced for over 3 million years will likely cause a dramatic shift in the climate and ecosystems of the Arctic.

*Data Availability.* The data generated and used in this analysis are available in the supplemental information associated with this article.

*Sample Availability.* Samples used in this analysis are curated by the Canadian Museum of Nature. Sample numbers used for each analysis are given in the supplemental information (Table S1 and S2).

*Supplemental Link.* To be provided by Copernicus Publishing

*Author Contribution.* Conceptualization: A.P.B. with modification by other authors; Methodology: A.P.B., J.G., J.S.S.D., K.J.B., T.F.; Formal analysis: All authors; Investigation: A.P.B., J.G., K.J.B., L.W., T.F.; Resources: A.P.B., J.G., J.S.S.D., K.J.B.; Data curation: A.P.B., J.G., K.J.B., L.W., T.F.; Writing—Original draft: All authors; Writing—Review and editing: All authors; Supervision: A.P.B., J.S.S.D., K.J.B., N.R.; Project administration: A.P.B., N.R., T.F.; Funding acquisition:

A.P.B., J.G., J.S.S.D., K.J.B., N.R., T.F. (Definitions as per the CRediT Taxonomy)

*Competing interests.* The authors declare that they have no conflict of interest

*Acknowledgements.* This work was funded by NSF Polar Programs to A.P.B.; National Geographic Committee for Research

and Exploration Grant (9912-16) and Endeavour Research Fellowship (5928-2017) to T.F.; National Geographic Explorer Grant (7902-05) and The W. Garfield Weston Foundation Grant to N.R.; student travel (NR supervised) was supported by the Northern Scientific Training Program (NSTP) from the government of Canada; an NSERC Discovery Grant (239961) with Northern Supplement (362148) to J.C.G; Natural Resources Canada (SO-03 PA 3.1 Forest Disturbances Wildland Fire) to K.J.B.; the European Research Council under the European Union′s Seventh Framework Programme (FP7/2007-2013) / ERC

grant agreement n ° [226600], and funding from the Netherlands Earth System Science Center (NESSC) through a gravitation grant (NWO 024.002.001) from the Dutch Ministry for Education, Culture and Science to J.S.S.D.

Alice Telka (Paleotec Services) identified and prepared macrofossil plants for the $CO_2$ analysis. We are also grateful to Nicholas Conder who assisted with sample preparation for the vegetation/fire reconstruction. We also acknowledge the 2006, 2010 and 2012 field teams including D. Finney (Environment Canada), H. Lasson (McGill), M. Vavrek (Royal Ontario



Museum), A. Dececchi (Queens University), W.T. Mitchell (Carleton University), R. Smith (University of Saskatchewan), and C. Schröder-Adams (Carleton University). The field research was supported by a paleontology permit from the Government of Nunavut, CLEY (D.R. Stenton, J. Ross) and with the permission of Qikiqtani Inuit Association, especially Grise Fiord (Nunavut). Logistic support was provided by the Polar Continental Shelf Program (M. Bergmann, B. Hyrcyk, B.

Hough, M. Kristjanson, T. McConaghy, J. MacGregor and the PCSP team).

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



**Table 1. Modern and recent Holocene fire return interval reconstructions for the candidate analogous regions considered in this study.**

| Region | Modern | | Reference | Recent Holocene | | Reference |
|---|---|---|---|---|---|---|
| Alaskan Tundra | Seward Peninsula | 273* | Kasischke et al. (2002) | Up-Valley | 263 | Higuera et al. (2011) |
| | Nulato Hills | 306* | | Down-valley | 142 | |
| Alaskan Boreal | Porcupine/ Upper Yukon (Central) | ~100 | Yarie (1981) | | | |
| | Sites near Fairbanks, and Delta Junction (Central) | 70130 | Johnstone et al. (2010a); Johnstone et al. (2010b); Johnstone and Kasischke (2005) | | | |
| | Kenai Peninsula | | Lynch et al. (2002) | Interior Alaska and Kenai Peninsula | 198 ± 90 | Lynch et al. (2002) |
| | Yukon river Lowlands | 120 | Kasischke et al. (2002) | Brooks Range | 145 | Higuera et al. (2009) |
| | Kuskokwim Mountains | 218 | | | | |
| | Yukon-Tanama Uplands | 330 | | | | |
| | Tanana-Kuskokwim Lowlands | 178 | | | | |
| | Kobuk Ridges and Valleys | 175 | | | | |
| | Davidson Mountains | 403 | | | | |
| | North Ogilive Mountains | 112 | | | | |
| | Ray Mountains | 109 | | | | |

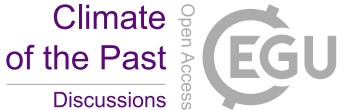



| | Yukon-Old Crow Basin | 81 | | | | |
|---|---|---|---|---|---|---|
| Western North America | Darkwoods, British Columbia | ~69 | Greene and Daniels (2017) | | | |
| | Cascade Mountains, Washington | ~27 | Wright and Agee (2004) | | | |
| | Desolation Peak, Washington Coastal type | 108-137 | | | | |
| | Desolation Peak, Washington Interior type | ~52 | | | | |
| Eastern North America | Quebec – west | ~270* | Bouchard et al. (2008) | Maine | ≥ 800 | Lorimer (1977) |
| | Quebec – east | >500* | | | | |
| | | | | Quebec – "Spruce zone" | 570 | de Lafontaine and Payette (2011) |
| | | | | Quebec – "Fir zone" | >1000 | |
| | Quebec – Abitibi northwest | 418* | Bergeron et al. (2006 post-1940)^ | Quebec – Abitibi northwest | 189 | Bergeron et al. (2006 post-1940)^ |
| | Quebec – Abitibi southwest | 388* | | Quebec – Abitibi southwest | 165 | |
| | Quebec – Abitibi east | 418* | | Quebec – Abitibi east | 141 | |
| | Quebec – Abitibi southeast | 2083* | | Quebec – Abitibi southeast | 257 | |





| | | | | | | |
|---|---|---|---|---|---|---|
| | Quebec – Temiscamingue north | 2083* | | Quebec – Temiscamingue north | 220 | |
| | Quebec – Temiscamingue south | 2777* | | Quebec – Temiscamingue south | 313 | |
| | Quebec – Waswanipi | 418* | | Quebec – Waswanipi | 128 | |
| | Quebec – Central Quebec | 388* | | Quebec – Central Quebec | 150 | |
| | Quebec – North Shore | 645* | | Quebec – North Shore | 281 | |
| | Quebec – Gaspésia | 488* | | Quebec – Gaspésia | 161 | |
| | Quebec – northwestern - lakeshore | 99' | Bergeron (1991) | Quebec – northwestern - lakeshore | 63' | Bergeron (1991) |
| | Quebec – northwestern – lake island | 112' | | Quebec – northwestern – lake island | 74' | |
| Fennoscandia | Sweden | * | Niklasson and Drakenberg (2001); Niklasson and Granström (2004) | North Sweden | 50-150 | Niklasson and Granström (2004); Niklasson and Granström (2000) |
| | | | | Southern Sweden | 20 | Niklasson and Drakenberg (2001) |
| | Central Sweden | * | Brown and Giesecke (2014) | Central Sweden - Klotjärnen | 180 | Brown and Giesecke (2014) |
| | | | | Central Sweden - Holtjärnen | 240 | |
| Siberian Plateau | Northern | 300 | | | | |
| | Southern | 80 | | | | |



| | Mean (64-71°N) | 110 | Kharuk et al. (2016); Kharuk et al. (2011) | | | |
|---|---|---|---|---|---|---|

^ = The reciprocal converted from burn rate (%) (see Van Wagner et al., 2006)

* = Estimates likely effected in some areas by human activity. In such instances Recent Holocene is preferred.

' = Fire cycle

†='Recent' here refers to records that (or have distinct sections that) begin after the end of the Holocene Climate Optima and

5    end near present





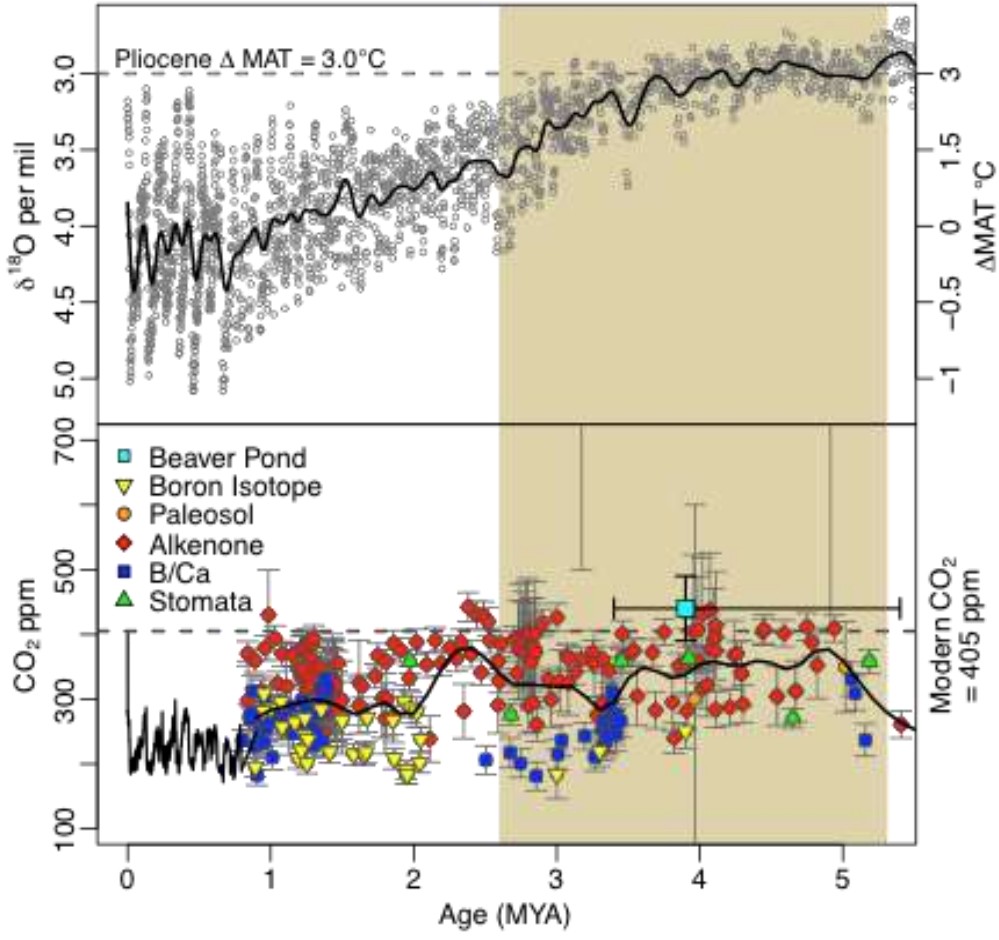

**Figure 1. Atmospheric CO₂ concentration and global temperatures and spanning the last 5 million years of Earth's history. Modern atmospheric CO₂ measurements (NOAA/ESRL), and ice core observations from EPICA (Luthi et al., 2008) are compared with proxy estimates (top panel) for the Pliocene Epoch highlighted in red (Hönisch et al., 2009; Pagani et al., 2010; Royer, 2006; Tripati et al., 2009). Mean annual temperatures (MAT) are inferred from compiled δ¹⁸O foraminifera data (Lisiecki and Raymo, 2005) and plotted as anomalies from present (bottom panel). Smoothed curves have been fit to highlight trends in *p*CO₂ and temperature during the Pliocene. The results from this paper (BP) are included with both age and *p*CO₂ error.**

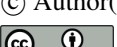



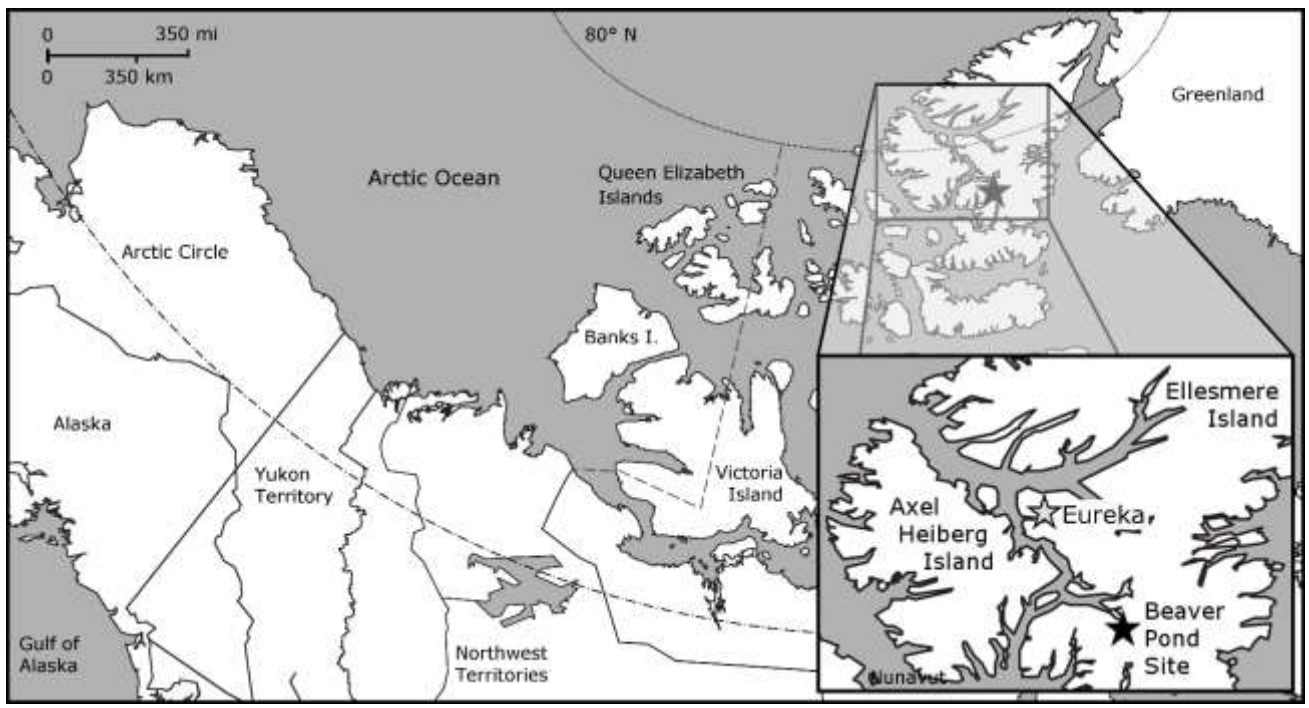

**Figure 2. Map of the Canadian Arctic Archipelago, highlighting the location of the Beaver Pond Site (Black Star; 78° 33′ N; 82° 25′ W) and Eureka Climate Station (Grey Star; 80° 13' N, 86° 11' W ¬ used for modern climate comparison) on west-central Ellesmere**

5 **Island.**





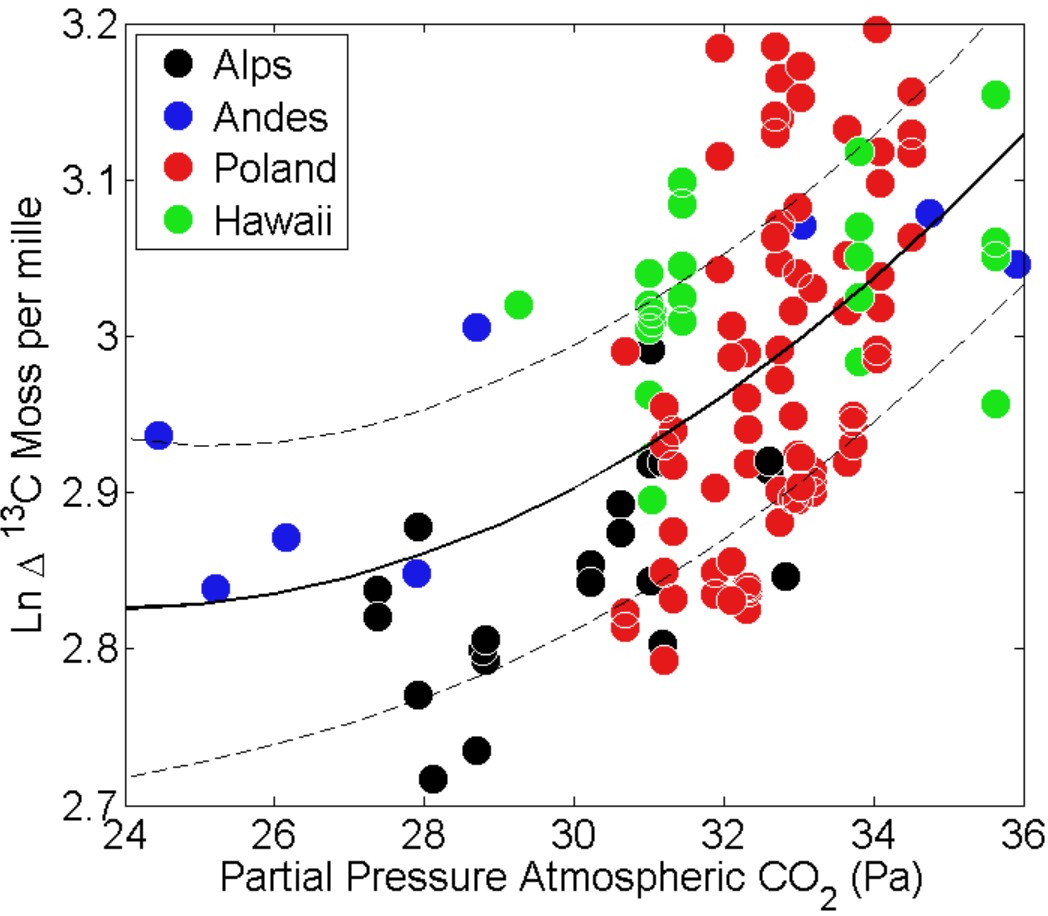

**Figure 3. Sensitivity of carbon isotopic discrimination to the partial pressure of atmospheric CO₂ in mosses from two different elevational transects. Moss carbon isotope data collected from an elevational transect in the Swiss Alps (Ménot and Burns, 2001) and from an elevational transect in the Peruvian Andes (Royles et al., 2014). Partial pressure of atmospheric CO₂ calculated from atmospheric surface pressure reanalysis data (Dee et al., 2011) combined with atmospheric CO₂ observations from year moss samples were collected. Carbon isotopic measurements of mosses reported as the natural logarithm of isotopic discrimination (Δ) from atmospheric δ¹³CO₂ (GlobalGlobal View-CO₂, 2013) from the year mosses were collected in units of per mille (i.e. ‰). Optimal model fit (black line) with ± 1σ confidence limits (black dashed lines) from Eq. (9).**

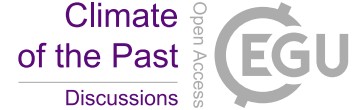



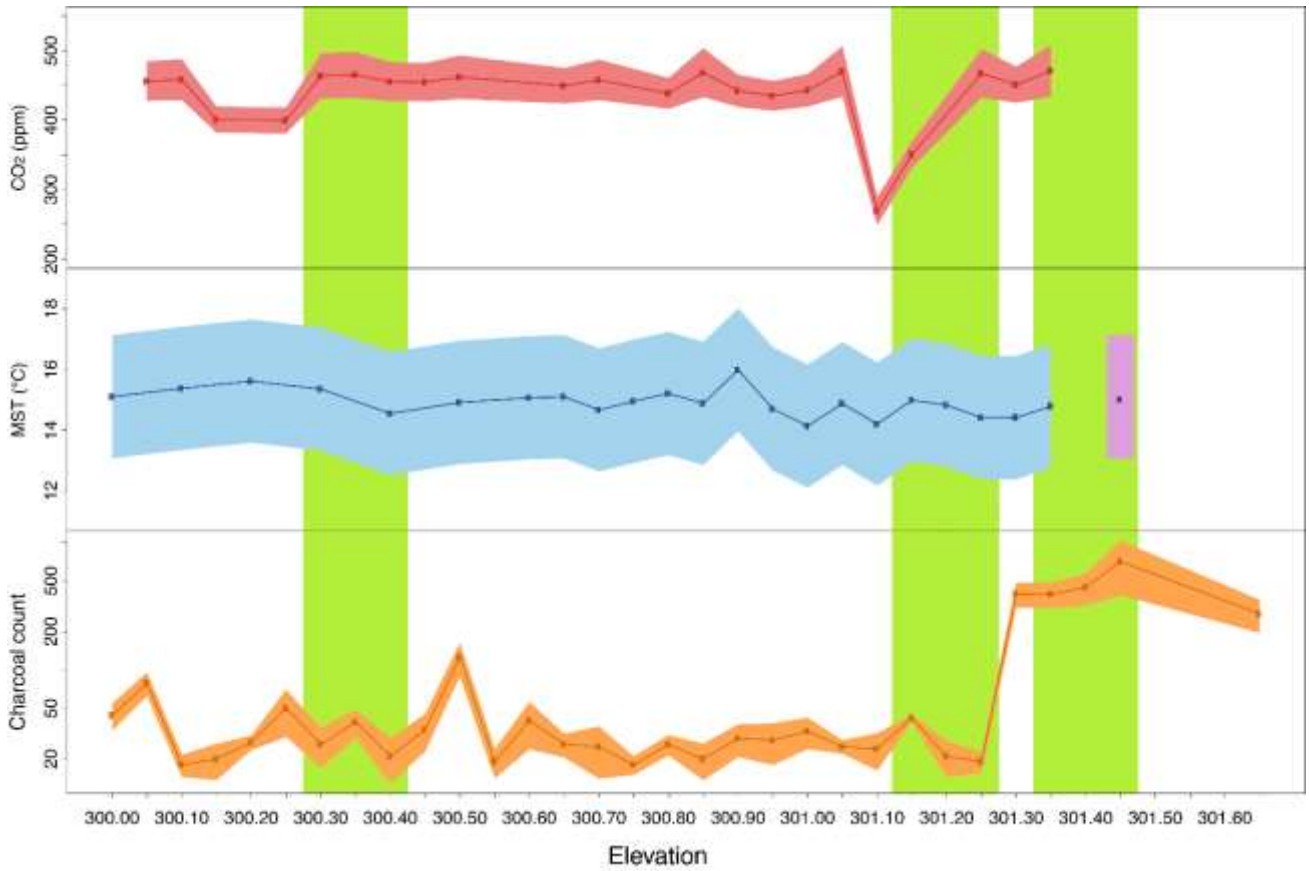

**Figure 4. Reconstruction of atmospheric CO₂, mean summer temperature, and fire for the Canadian High Arctic during the Pliocene from the 2006 series unless noted. Atmospheric CO₂ concentrations estimated from carbon isotopic measurements of mosses and plants (red; ± 2 σ). Mean summer temperature reconstructed from a brGDGT based proxy (blue; ± 2 σ) and relative 2010 data point in approximate relative position (purple; ± 2 σ). Charcoal counts reported as the number of fragments per volume of peat (Orange ± 2 σ). Green boxes indicate relative depths of pollen sampling. Elevation of the deposit is reported as meters above sea level. (Data: Table S2)**





**(A)**

**(B)**

Figure 5. (A) Bar charts showing the relative pollen abundance in each portion of the section (error bars = 95% confidence intervals; MASL- Meters Above Sea Level). (B). Pollen plate of select grains encountered in the BP section: (a) *Pinus*, (b) half a *Picea* grain, (c) *Larix*, (d) *Betula*, (e) *Alnus*, (f) *Salix*, (g) *Myrica*, (h) ericaceous grain, (i) *Epilobium*, and (j) Cyperaceae. 50um scale = (a–c), 75um scale = (d–j).





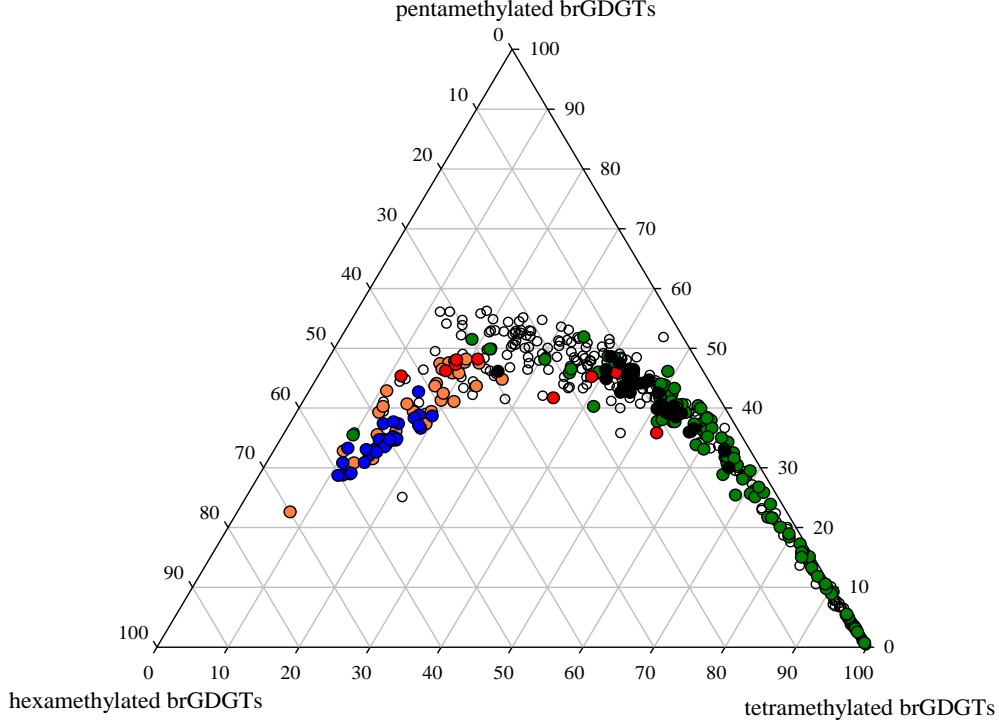

**Figure 6. A ternary plot illustrating the fractional abundances of the tetra- (Ia-c), penta (IIa-c and II′a-c), and hexamethylated (IIIa-c and III′a-c) brGDGTs. The global soil dataset (open circles; De Jonge et al., 2014), the global peat samples (green circles; Naafs et al., 2017), and lake sediments from East Africa (black circles indicate samples from lakes >20°C, red circles indicate samples from lakes between 10–20°C and orange circles designate samples from lakes <10°C; Russell et al., 2018) are included for comparison with the Beaver Pond sediments (blue circles; this study).**



**Figure 7. (a) Modern geographic distribution of observed occurrences of species common to the Beaver Pond species list, (b) Mean temperature of the warmest quarter (summer average) derived from WorldClim, (c) Mean precipitation of the warmest quarter (summer rain) derived from WorldClim, (d) Count of unique fire pixels detected per day, over 10 years from MODIS 6 Fire Product, normalized by area of the latitude by longitude grid.**