# Peer review of "Evidence for fire in the Pliocene Arctic in response to amplified temperature"

_Climate of the Past, 2018_

## Referee Comment (RC1) · R.H. Smittenberg (Referee) · 11 Jul 2018

The paper presents a detailed investigation of a sedimentary deposit from the Pliocene period, located in the Canadian Arctic. The results give insight in the past environment and climate of this high arctic region from a geologic era with atmospheric CO2 levels comparable to that of today (as indicated by earlier studies), and is thus relevant for our understanding how the long-term climate may develop in a high CO2 world, with focus on the high Arctic region. The majority of the paper is well written, however some parts are not. My main concern lies in the section about the atmospheric CO2 reconstruction. Although I do agree that the basic concept that higher CO2 availability

for plants could, in principle, lead to a stronger fractionation against 13C, and thus that the 13C content of fossil plants could possibly be used to reconstruct past levels of atm. CO2 (in essence using the same approach Pagani et al took in using the 13C content of specific algal lipids, long-chain alkenones), I find that there are some major flaws in their execution.

Below is a more detailed list of comments.

Abstract: there are many issues with the English style and exact and careful phrasing. For instance, one needs to assume that CO2 concentrations are in the atmosphere. In line 24: isotope ratios of 440 ppm? Line 30: 'furthest northern evidence' (northern-most?) .

p2line6. No newer references? P2l16-18. Revise / make clear and expand what the relevance is of the 100k vs 41K orbital cycles, give references. Or leave out. P2l30. In the rest of the paper it does not become very clear how fire has a large impact as climate amplifier. What is a 'proximal mechanism'? P3: Generally written in a very sloppy manner. P3l1-12. Confusing piece mixing up sea ice conditions, industrial black carbon and natural (counteracting) effects. How could one have observed temperatures in the Pliocene? In other words: revise. P3l16. Check writing P3l20-25: Particularly badly written. To what does 'This' exactly refer to (l21)? Dating uncertainties suggest an additional hypothesis? Can proxies be 'deposited'? P4l10/11. Entirely unclear: 'spanned the 1 m remaining of Unit II as per Mitchell et al' P4l17. 'samples of these 2006'-? P4l32. 'Approximately' 200.00 mg Be (and thus not 200.01)?? s(ame for 150.00 g quartz?)

Section 2.3 and 3.2 (P5l12. Carbon isotopic discrimination) The authors start using their equation 1 (taken from Farquhar 1989), derived for C3 plants with stomata, to describe fractionation against 13C by bryophytes. This is fine, they start in the same way as Fletcher et al (2008), as astomatous plants like bryophytes do (isotopically) behave fairly similarly. However, instead of taking the well developed model used by

Fletcher et al (the basic concept and many tests described in GCA vol 70, p5676; but also see Fletcher 2005 in GlobBiogeochemCycles) they try to re-invent the wheel – however a very crooked one. The substitution of eq 2 into eq 1 is fine as long as one wants to back-correct for (paleo)height once a p(a) has been estimated from any transfer function of Delta13C to pCO2. However one cannot simply substitute p(i)/p(a) (a ratio between 0-1) simply by p(a), it totally changes the equation/model and even units. Moreover, I really do not see any reason or advantage of using the natural log of Delta13C instead of Delta13 - unless one wants to focus on the height term of eq. 2. Set their overly simplified and actually wrong theoretical exercise aside, the authors then compare Delta13C with pCO2 from a range of altitudes (not taking into account also lower pO2 levels) to arrive at some empirical relation between these two. As they write in their discussion, there are many confounding factors that could have influenced the observed C isotope fractionation - indeed resulting at different slopes for each site. Choosing a simple polynomial fit through this data has no theoretical basis at all, and is highly biased by the few Andean results and the Swiss sites. The majority of their plot comes from a Polish site – however in that original article the primary cause for the 13C discrimination was thought to be temperature, not altitude, although these two factors do co-vary. In the discussion (section 4.2) the authors are reasonably cautious about their model, however in my opinion the their framework is in any case ready for the trash bin and should get removed from the paper. I really wonder why the authors have not taken the model and results of Fletcher (2005, 2006, 2008), which does have a solid theoretical base ground in isotope systematics but also plant physiology. The first thing the authors need to do is discuss their results within the framework and transfer functions from Fletcher. Once they do that , I am skeptical if their data is not too compromised many environmental factors like temperature or humidity, but this remains to be seen. Also note that the framework of Fletcher only appears to work with reasonable (un)certainty on a larger amplitude of pCO2 between 300-2500 ppm. I don't think that the confounding factors give 'subtle differences' (p9l31). In the end, interpreting the 13C values from bryophytes from one single location appears to be a

very uncertain enterprise. The authors need to provide a solid error assessment, at the moment this is highly under-developed.

About the assessment of the Pliocene 13CO2 value using buckbean 13C values: this appears to be a reasonable approach, although it would be good to get this estimate confirmed by measuring some more plants. Note Stuiver&Braziunas (1989, Nature328, p58 Tree cellulose 13C/12C isotope ratios and climatic change) who observe a relation between latitude and fractionation (likely cause by changes in Temp and humidity).

About the measurements of the plant 13C values: why was chosen to measure cellulose instead of bulk tissue? What is the expected difference in 13C, knowing that sugars typically have less depleted 13C values than the bulk (lipids being more depleted)? The description of the preparation and isotope measurement methods is very limited and should get expanded.

Section 2.4 The sections about the brGDGTs as well as the one about fire, vegetation and climate are well written, and extensive and critical enough, and I have no real comments here. However, to make the jump from the local fire frequency at one location to the notion that fire could have been a global climate feedback mechanism during the Pliocene is a very large jump to conclusions (p19l12-16). It is fine to mention this possibility, but I would not use the word 'reveal' (line 12) but use a more careful wording (e.g. indicates, suggests).

P6l32. How 'well' are the brGDGTs really preserved? P7l1 That the brGDGTs are 'thought to be sourced by a wide array of acidobacteria within the soil' is still under investigation and there is still only scant evidence. For one, brGDGTs are also produced aquatically. Rephrase. P7l23. How was a concentration of 10 mg ml-1 (of brGDGTs, if one reads the text) made? Concentration of Total lipid extract or polar fraction? P7l25 mass spectrometry. UHPLC or HPLC?. P7l28. From where does the transfer function error come? P8l1 minus term missing

3. Results P9l11. Not clear what the maximum probability of age of 4.5 Ma means,

when earlier the most likely age is estimated at 3.9 Ma? P9l19. Is an error of 104 to 105 years relevant on the geologic timescale of millions of years?

Conclusions: Depending on any revision of the paper, alter or remove mention to past $CO_2$ levels. Also be more careful in the conclusions with respect to the role of fire on climate. The paper showcases well that fire was part of the arctic climate – however this is not so different from the present day boreal realm, and the paper does not at all investigate, model or discuss this aspect. The same is true for the last sentence about present day arctic climate change, the paper does not focus at all on the present day arctic.

Figure 3: No references given for the Polish and Hawaiian sites. Fig. 4 I would also plot the originally measured (estimated) Delta13C values,not only reconstructed $pCO_2$ (but as stated above, I find that this aspect of the paper needs an overhaul in any case).

---

## Referee Comment (RC2) · R.H. Smittenberg (Referee) · 11 Jul 2018

Additional comment besides the just uploader referee comment: I find the title rather misleading. The paper does not really discuss either the role of elevated pCO2 nor that of fire as arctic amplifiers. It merely provides paleoenvironmental and -climatic proxy data of this high Arctic site (and is in this sense very useful). But not more than that.

---

## Referee Comment (RC3) · D. Royer (Referee) · 15 Jul 2018

Fletcher and colleagues present a Pliocene high arctic record of CO2, temperature, plant species composition, and inferred fire frequency. They then explore how these components may be interconnected. The study summarizes an impressive amount of data. My expertise lies with paleo-CO2 reconstruction and so my review will focus on there.

1. CO2 reconstruction.

-Ben Fletcher developed a process-based model for paleo-CO2 reconstruction based

on the $\delta$13C of liverworts. I'm surprised that the authors have not tried to incorporate/modify this method for their own study. It's not even mentioned! Instead, the authors rely on a present-day empirical-based model, which is likely to be inferior to a process-based model. To give just one example, the authors note the problem of growth rates with other paleo-CO2 methods (p. 15, line 10). But growth rate is a key uncertainty with the authors' method, and something that is acknowledged and (partially) addressed in the Fletcher model. This is a key deficiency with the current manuscript.

Fletcher, B. J., Beerling, D. J., Royer, D. L., and Brentnall, S. J., 2005, Fossil brophytes as recorders of ancient CO2 levels: Experimental evidence and a Cretaceous case study: Global Biogeochemical Cycles, v. 19, p. GB3012, doi:3010.1029/2005GB002495.

Fletcher, B. J., Brentnall, S. J., Quick, W. P., and Beerling, D. J., 2006, BRYOCARB: A process-based model of thallose liverwort carbon isotope fractionation in response to CO2, O2, light and temperature: Geochimica et Cosmochimica Acta, v. 70, p. 5676-5691.

-Using leaf $\delta$13C to reconstruct air $\delta$13C is problematic because many factors—for example water stress—can affect leaf $\delta$13C. The authors are assuming no change in water stress (and other factors that could affect leaf $\delta$13C) between the present-day and Pliocene. Given what is said in section 4.3, this assumption is tenuous.

Diefendorf, A. F., Mueller, K. E., Wing, S. L., Koch, P. L., and Freeman, K. H., 2010, Global patterns in leaf 13C discrimination and implications for studies of past and future climate: Proceedings of the National Academy of Sciences, USA, v. 107, p. 5738-5743.

Kohn, M. J., 2010, Carbon isotope compositions of terrestrial C3 plants as indicators of (paleo)ecology and (paleo)climate: Proceedings of the National Academy of Sciences, USA, v. 107, p. 19691-19695.

-The empirical transfer function (Figure 3) maxes out at 360 ppm. The authors use the function to reconstruct ∼450 ppm. This is a problem with extrapolation.

-The empirical transfer function is based on a mix of species. Some of the scatter is likely due to "vital effects". This needs to be acknowledged. The best transfer function would be one based on the same species (or genus) as the fossil material.

-The authors have underestimated the uncertainty associated with their paleo-CO2 reconstructions. As best as I can tell, their stated uncertainty (1ïАş = 35 ppm) is the confidence interval from Figure 3 (dashed lines). A confidence interval says how confident one is in the regression. But if one wishes to infer the y-axis value from a new single data point (as is being done here), the prediction interval is appropriate. And the prediction interval is wider than the confidence interval. In addition, the authors have not propagated uncertainty associated with the measurement(s) of leaf $\delta$13C at each level; the authors are assuming no error. Beerling et al. (2009) lays out a solid strategy for propagating uncertainty with these kind of empirical functions.

Beerling, D. J., Fox, A., and Anderson, C. W., 2009, Quantitative uncertainty analyses of ancient atmospheric CO2 estimates from fossil leaves: American Journal of Science, v. 309, p. 775-787.

-The authors deal with the confounding factor of water stress in the Discussion, but this section should move to the Introduction. Otherwise, the informed reader will be wondering why the authors haven't dealt with the issue while they are reading the Intro, Methods, and Results.

2. CO2 compilation.

-The B/Ca estimates should be excluded as they are not reliable.

Allen, K. A., and Hönisch, B., 2012, The planktic foraminiferal B/Ca proxy for seawater carbonate chemistry: a critical evaluation: Earth and Planetary Science Letters, v. 345–348, p. 203-211.

-It looks like some Pliocene estimates have been missed. See Foster et al. (2017) for compilation and citations.

Badger, M. P. S., Schmidt, D. N., Mackensen, A., and Pancost, R. D., 2013b, High-resolution alkenone palaeobarometry indicates relatively stable pCO2 during the Pliocene (3.3–2.8'Ma): Philosophical Transactions of the Royal Society A, v. 371, 20130094.

Bartoli, G., Hönisch, B., and Zeebe, R.E., 2011, Atmospheric CO2 decline during the Pliocene intensification of Northern Hemisphere glaciations: Paleoceanography, v. 26, PA4213, doi:10.1029/2010PA002055.

Foster, G. L., Royer, D. L., and Lunt, D. J., 2017, Future climate forcing potentially without precedent in the last 420 million years: Nature Communications, v. 8, p. 14845, doi:14810.11038/ncomms14845.

Martínez-Botí, M. A., Foster, G. L., Chalk, T. B., Rohling, E. J., Sexton, P. F., Lunt, D. J., Pancost, R. D., Badger, M. P. S., and Schmidt, D. N., 2015, Plio-Pleistocene climate sensitivity evaluated using high-resolution CO2 records: Nature, v. 518, p. 49-54.

Seki, O., Foster, G.L., Schmidt, D.N., Mackensen, A., Kawamura, K., and Pancost, R.D., 2010, Alkenone and boron-based Pliocene pCO2 records: Earth and Planetary Science Letters, v. 292, p. 201-211.

Stap, L. B., de Boer, B., Ziegler, M., Bintanja, R., Lourens, L. J., and van de Wal, R. S. W., 2016, CO2 over the past 5 million years: continuous simulation and new $\delta$11B-based proxy data: Earth and Planetary Science Letters, v. 439, p. 1-10.

Zhang, Y. G., Pagani, M., Liu, Z., Bohaty, S. M., and DeConto, R., 2013, A 40-million-year history of atmospheric CO2: Philosophical Transactions of the Royal Society A, v. 371, 20130096.

3. Temperature component.

-The temperature record feels like a "third wheel" to the CO2 and fire records. In the discussion, other temperature records (often from the same site) are emphasized more than the record generated here.

4. Link between fire and climate.

-Quite a lot of space in the Discussion is devoted to how fire and climate are interconnected. And the bulk of this discussion centers on the literature. But, the record generated here shows no obvious link between fire and CO2 or temperature (Figure 4). As a result, there is a logical disconnect. For example, from the Introduction (p. 2, lines 30-31): "We propose that fire in arctic ecosystems may also be an important proximal mechanism for amplifying arctic surface temperatures during the Pliocene."

Minor comments:

p. 3, line 15: B/Ca (not "Boron")

p. 3, line 19: Foster et al. (2017, Nature Communications) is a more current reference

p. 3, line 24: what do you mean by "Although direct effects may be small"?

p. 4, lines 14-16: "The unit sampled spanned the 1 m remaining of Unit III as per Mitchell et al. (2016). The main sequence examined across the methods used in this study includes material above (Unit IV) and below (Unit II) Unit III, with a total sampled profile of 1.65 m." Parts of these sentences are confusing: 'unit' appears to have a different meaning as 'Unit'; what does '1 m remaining of Unit III' mean? (is most of the originally sampled material gone?); the first sentence implies that all of the data come from Unit III, but the second sentence says that some of Units II and IV are included too.

p. 6, line 31: "We also measured $\delta$13C of modern buckbean to constrain our estimates of pi / pa." More context is needed so that the reader can understand this statement. Why do the pi/pa estimates need to be 'constrained', and why do present-day measurements allow you to do this?

p. 6, line 32: say that cellulose was measured (ditto in line 31).

p. 7, line 11: "paleosols"

p. 7, lines 30-32: Provide a citation for the transfer function. Is the combined error a one-sigma error? Two-sigma? Was quadrature used to calculate the combined error?

p. 8, line 19: "MAT"?

p. 9, lines 17-18: What is the difference between a maximum probability age and an optimized age?

p. 9, line 22: Unweighted mean age already stated.

p. 9, line 23: How was this uncertainty computed?

p. 9, lines 28-30: The first and second parts of this sentence are saying the same thing.

p. 9, lines 30-31: I don't understand why nonlinearity is expected because Figure 3 plots the log of carbon isotope discrimination (also, see next sentence in the manuscript).

p. 10, line 2: These "other processes" don't need to be nonlinear; the combined additive effect is nonlinear.

p. 10, lines 9-12: It is inappropriate to use the site-specific regression because the associated site-specific information is not available for the Pliocene samples.

p. 10, line 11: Is this the same model as in line 1?

p. 15, line 23: "over the Pliocene". Surely you don't mean the entire Pliocene?

p. 19, lines 30-31: "fire played an active role in…influencing the climate of the Arctic during the Pliocene." That's not what your data suggest (Figure 4).

Figure 3: Please add the theoretical regression (from equations 2 & 3). Add linear

y-axis tick marks too. In the caption, say that the isotopic discrimination is based on cellulose (or an inferred cellulose value).

Figure 4: unit needed for x-axis.

Figure S2: Plots need axis labels

Figure S3: Axis and tick mark labels are too pixelated to read

---

## Referee Comment (RC4) · C. Schweger (Referee) · 14 Aug 2018

1) The manuscript presents new information, methods and the concept (hypothesis) that fire may amplify temperature increases. Excellent These are important additions to our knowledge base and new ideas.

2) The new biogeochemical methods are quite exciting but as the authors admit, the methods and results are based on many assumptions. My knowledge of some of the methods is clearly limited but it is obvious that there will no doubt be refinements in the future. That is to be expected. As I have pointed out in my review the key hypothesis involving fire amplification is not really explored very well. Therefore, the paper is good.

3) Field work at key sites requires detailed and clear descriptions so readers who can't visit the site understand what took place. If the site is to be revisited, new workers should know exactly what and where research took place during past field seasons. I don't believe the field descriptions are clearly presented. There is need for a figure diagraming the site, the units and where sampling was done for geochronology, chemistry, pollen, etc. Figures 5 and 6 are reversed. Of course there are some typos and closer editing is required. I'd give a fair rating.

I believe this manuscript should be accepted subject to recommended minor revisions. If the authors take my comments to heart as being constructive and complete them, I see no reason for me to review the manuscript again.

Respectfully your,

Charles Schweger, Prof. Emeritus

Please also note the supplement to this comment:
https://www.clim-past-discuss.net/cp-2018-60/cp-2018-60-RC4-supplement.pdf

**Supplement:**

Peer Review: Tamara Fletcher et al. "The role of elevated atmospheric CO2 and increased fire in Arctic amplification of temperature during the Early to mid-Pliocene"

My goodness, how important a simple beaver pond has become. But this pond is 4 million years old, at 77 degrees North latitude and may hold important data that can explain discrepancies in paleoclimate reconstructions for a critical period in recent earth history. This is an important paper and show cases the impact of biogeochemistry methods on paleoecology. Unfortunately, I can offer little insight with this part of the paper. My lack of knowledge and inability to read the equations which were in Chinese or special characters accounts for this. This paper is not, however, without it's faults at other levels.

While much is made of the role of fire in amplifying the arctic temperature response to elevated CO2 there is surprisingly little discussion on the topic. Page 19, lines 18- 22, mention Feng et al. 2016 and the direct and indirect effects but these are in the most general terms. Line 32 in the Conclusions is similarly uninformative and therefore unconvincing. We go from great methodological detail to the most general statements in the main focus of the paper.

Page 4, Site Description leaves much to be desired. Luckily, I had a copy of Mitchell et al. 2016 to provide the details. I certainly couldn't follow what Fletcher et al. were describing. Perhaps there is need for a site diagram to illustrate the stratigraphy, the Units and where and when specific sampling was done. That the BP site has been collected off and on for more than a decade and perhaps into the future and the data assembled is so important means that reproducibility is very important. Therefore, to me, detailed site description and sample locations are critical.

I'm assuming the fen (Beaver Pond) peat is autochthonous even through the over all site is in a fluvial environment. I'm assuming that a till is the surface deposit and that the surface represents the stratigraphic datum and all Unit and sample measurements are from the surface. I'm assuming that Unit III (Mitchell et al.) is the fen peat.

Let me look at page 8, Vegetation and Fire Reconstruction. Page 8, Line 28, sampled at an upper and lower elevation . . . that correspond with changes in charcoal.
Explain "upper and lower elevation" or provide depths. Does this mean that charcoal samples were processed first and the pollen samples selected on those results?

P. 9, L., 3, plant taxa. L. 4, is there a better word for "observation" L. 23. What is meant by "The age of the Beaver Pond peat is stratigraphically younger . . ." Younger than what? Need diagram. L.25, where does the 104 t0 105 years come from? And how can you be this precise?

P. 10, l. 2 and 3, check pa and p sunscript a Is this a typo or a different measure

P. 11, Figures 6 and 5 are reversed. 6 is referred to before 5.

P.12, L.20, 21 and 25.  Do the authors mean "samples" or sections?  If "sections" then I'm not sure what they are talking about as one samples sections in the course of field work and samples are processed in subsequent lab work.  Line 29, describes potential Populus pollen.  Since this is such an important component of the pollen assemblage, I am surprised that more effort wasn't made to identify the unknown.  Possibly SEM analysis, opinion of other experts, even DNA.  Populus is afterall capable of inhabiting high latitudes, is an important species for beavers and almost an expected component in a boreal forest environment.  Were wood fragments identified?

Was a comparison made between macrofossils and pollen taxa?  Wouldn't that be interesting and useful?   What are the NAP taxa discovered?  Why are no pollen sums presented?

Page 13, L. 2, Does 6% Pinus pollen indicate that pine was a component locally?  Would the work of Jocelyne Bourgeois, GSC, have any bearing on or aid these interpretations?  She analyzed high latitude pollen and demonstrated how pollen and charcoal could be transported long distances.

Page 13, L.28, Such a lack of precision.  What does "very close to the BP peat" mean?

Page 17, L. 13.  Here is another example of field work and site description problems.  "It is possible that the Larix-Betula Parkland dominated . . . correspond to the . . . Units II and III."  Why aren't they sure?  Was there a continuity in field workers over the seasons of field work or was N. R. the only participant carryover?   Has the site eroded from 2008 to 2010 to 2012?

Page 16 to 20. Discussion and Conclusions
What type of data would the presence of fire provide climate modelers to improve their simulations?  Albedo?  Canopy transpiration-evaporation?  Surface texture,  snow capture and melt?  Fire reoccurrence is an important measure but what about regrowth?  Northern B.C. fires can decimate a landscape and within five years it is lush with deciduous regrowth and conifer seedlings.

If this paper is to be significant and meet the authors claims they must be able to more fully discuss the importance of fire amplification and model conditions.  Nearly every pollen sample in the world will contain some charcoal of varying size classes.  What then makes this work unique and important?

The text will need some careful editing.  There are abbreviations that I don't recognize and I enjoy the author's recognition of that "There are numerous assumptions . . ." p. 15, L. 20.  No doubt the measured ages, temperatures and precipitations will be refined in the future as methods improve and assumptions become more sound.

I believe this is a useful, perhaps significant paper but it needs further work in certain key areas that will help to make the case that fire is, was, an important factor in amplification of arctic temperatures.

Respectfully yours,
Charles Schweger, Professor Emeritus

---

## Author Comment (AC1) · 15 Oct 2018

In response to this comment we have edited the title, and hope this better reflects the content of the article. The title is now "Elevated atmospheric CO2 and fire linked to arctic amplification of temperature during the Early to mid-Pliocene".

---

## Author Comment (AC2) · 15 Oct 2018

The paper presents a detailed investigation of a sedimentary deposit from the Pliocene period, located in the Canadian Arctic. The results give insight in the past environment and climate of this high arctic region from a geologic era with atmospheric CO2 levels comparable to that of today (as indicated by earlier studies), and is thus relevant for our understanding how the long-term climate may develop in a high CO2 world, with focus on the high Arctic region. The majority of the paper is well written, however some parts are not. My main concern lies in the section about the atmospheric CO2 reconstruction. Although I do agree that the basic concept that higher CO2 availability for plants could, in principle, lead to a stronger fractionation against 13C, and thus that the 13C content of fossil plants could possibly be used to reconstruct past levels of atm. CO2 (in essence using the same approach Pagani et al took in using the 13C content of specific algal lipids, long-chain alkenones), I find that there are some major flaws in their execution.

RE: We have addressed Dr. Smittenberg's concerns that have lead us to revisit our analysis and a major revision of our manuscript.

Below is a more detailed list of comments.

Abstract: there are many issues with the English style and exact and careful phrasing. For instance, one needs to assume that CO2 concentrations are in the atmosphere. In line 24: isotope ratios of 440 ppm? Line 30: 'furthest northern evidence' (northern- most?) .

RE: Changes have been made to the text to clarify these ambiguities.

p2line6. No newer references?

RE: A more recent paper, Francis and Skific 2015, has been added.

P2l16-18. Revise / make clear and expand what the relevance is of the 100k vs 41K orbital cycles, give references. Or leave out.

RE: This sentence has been removed.

P2l30. In the rest of the paper it does not become very clear how fire has a large impact as climate amplifier. What is a 'proximal mechanism'?

RE The term 'proximal' has been removed, and it has been clarified that this hypothesis was a motivator for the study rather than an outcome.

P3: Generally written in a very sloppy manner.

P3l1-12. Confusing piece mixing up sea ice conditions, industrial black carbon and natural (counteracting) effects. How could one have observed temperatures in the Pliocene? In other words: revise.

RE: Although this article was motivated by the question of fire's role in Arctic amplification, it does not directly address it within this paper, and so this paragraph discussing the interactions of fire on climate has been removed.

P3l16. Check writing P3l20-25: Particularly badly written. To what does 'This' exactly refer to (l21)? Dating uncertainties suggest an additional hypothesis? Can proxies be 'deposited'?

RE: This paragraph has been edited to improve clarity

P4l10/11. Entirely unclear: 'spanned the 1 m remaining of Unit II as per Mitchell et al' P4l17. 'samples of these 2006'-?

RE: These sections have been edited to improve clarity

P4l32. 'Approximately' 200.00 mg Be (and thus not 200.01)?? same for 150.00 g quartz?)

RE: This has now been changed to approximately 200 and approximately 150.

Section 2.3 and 3.2 (P5l12. Carbon isotopic discrimination) The authors start using their equation 1 (taken from Farquhar 1989), derived for C3 plants with stomata, to describe fractionation against 13C by bryophytes. This is fine, they start in the same way as Fletcher et al (2008), as astomatous plants like bryophytes do (isotopically) behave fairly similarly. However, instead of taking the well developed model used by Fletcher et al (the basic concept and many tests described in GCA vol 70, p5676; but also see Fletcher 2005 in GlobBiogeochemCycles) they try to re-invent the wheel – however a very crooked one. The substitution of eq 2 into eq 1 is fine as long as one wants to back-correct for (paleo)height once a p(a) has been estimated from any transfer function of Delta13C to pCO2. However one cannot simply substitute p(i)/p(a) (a ratio between 0-1) simply by p(a), it totally changes the equation/model and even units. Moreover, I really do not see any reason or advantage of using the natural log of Delta13C instead of Delta13 - unless one wants to focus on the height term of eq. 2. Set their overly simplified and actually wrong theoretical exercise aside, the authors then compare Delta13C with pCO2 from a range of altitudes (not taking into account also lower pO2 levels) to arrive at some empirical relation between these two. As they write in their discussion, there are many confounding factors that could have influenced the observed C isotope fractionation - indeed resulting at different slopes for each site. Choosing a simple polynomial fit through this data has no theoretical basis at all, and is highly biased by the few Andean results and the Swiss sites. The majority of their plot comes from a Polish site – however in that original article the primary cause for the 13C discrimination was thought to be temperature, not altitude, although these two factors do co-vary. In the discussion (section 4.2) the authors are reasonably cautious about their model, however in my opinion the their framework is in any case ready for the trash bin and should get removed from the paper. I really wonder why the authors have not taken the model and results of Fletcher (2005, 2006, 2008), which does have a solid theoretical base ground in isotope systematics but also plant physiology. The first thing the authors need to do is discuss their results within the framework and transfer functions from Fletcher. Once they do that , I am skeptical if their data is not too compromised many environmental factors like temperature or humidity, but this remains to be seen. Also note that the framework of Fletcher only appears to work with reasonable (un)certainty on a larger amplitude of pCO2 between 300-2500 ppm. I don't think that the

confounding factors give 'subtle differences' (p9l31). In the end, interpreting the 13C values from bryophytes from one single location appears to be a very uncertain enterprise. The authors need to provide a solid error assessment, at the moment this is highly under-developed.

RE: We agree that reconstructing past $CO_2$ levels, considering all the underlying assumptions and sources of error, is no trivial task. In fact, our first attempts to reconstruct past $CO_2$ levels using $\Delta^{13}C$ of bryophytes, were based on the original approach pioneered by White et al. (1994). However, we found that many of these previously proposed theoretical models that have been since refined in the BRYOCARB model of Fletcher et al. (Fletcher et al. 2008) had far too many physiological variables that were poorly constrained when dealing with paleoenvironments and thus had too many degrees of freedom to provide reliable estimates of past $CO_2$. Dr. Fletcher actually provided us with his BRYOCARB model and we also found it difficult to produce reasonable estimates of Pliocene $CO_2$ concentrations based on the number of tunable parameters, which is why we opted to derive our own independent empirical transfer function based on the increase in $\Delta^{13}C$ of bryophytes as a function of changes in the partial pressure of $CO_2$ with elevation. Our empirical approach is based on the same physical principles and underlying assumptions as the BRYOCARB model: 1.) plants that lack stomates have no mechanism to actively regulate their $CO_2$ gradient and thus are sensitive to the partial pressure of $CO_2$ in their environment. 2.) as the partial pressure of $CO_2$ increases, more $CO_2$ is driven into the moss through diffusion and a greater pool of $CO_2$ is available for discrimination by rubisco (Farquhar, Ehleringer and Hubick 1989). Based on the comments from Reviewer #1 and Reviewer #2, we have revisited the BRYOCARB model to better evaluate our results.

One test is to see how well the BRYOCARB model performs at predicting the observed $\Delta^{13}C$ of modern mosses across the gradient in partial pressure of atmospheric $CO_2$. We iteratively optimized tunable parameters in the BRYOCARB model to predict the overall mean value of observed $\Delta^{13}C$ values 19.6 ± 2.1sd. It is clear that both BRYOCARB and our empirical model show the expected increase in $\Delta^{13}C$ with $pCO_2$; however, the slopes of these relationships are markedly different (*Fig. C1*). There are clearly processes affecting the Moss $\Delta^{13}C$ values that are not necessarily captured by the BRYOCARB model and are contributing to the large error envelope in our empirical model. The BRYOCARB model does not appear to be sensitive enough over this range of atmospheric $CO_2$ partial pressures to simulate the response in the observed $\Delta^{13}C$ moss values. Furthermore the RMSE from our empirical model is 1.8 $\Delta^{13}C$, whereas the RMSE for the BRYOCARB model is 2.1 $\Delta^{13}C$.

[Figure]

*Figure C1. Predictions from BRYOCARB model (blue dashed line) and our empirical transfer function (black solid line with error envelope) compared with observed moss isotopic discrimination (points). Moss $\Delta^{13}C$ observations are from elevational transects in the Swiss Alps, Peruvian Andes, Poland, and Hawaii.*

As mentioned by Dr. Smittenberg it has been previously noted that temperature may explain the variability in $\Delta^{13}C$ with elevation (Skrzypek et al. 2007). To test this explanation, we evaluated to what extent temperature could affect the isotopic discrimination across the elevation transects of observed moss $\Delta^{13}C$ and ultimately the impact on our Pliocene $CO_2$ estimates. Using the BRYOCARB model, with all other variables held constant, we determined that some of the variance in $\Delta^{13}C$ that is not described by partial pressure in atmospheric $CO_2$ (Fig. C1), may be explained by temperature (Fig. C2). In fact, assuming a moist adiabatic lapse rate 5 deg C/ km for the published data from the Peruvian Andes, we see that the data span ~2700m in elevation correspond to a 13.5 °C range in temperature. According to BRYOCARB this can result in a 1.0 ‰ enrichment in moss $\Delta^{13}C$ as temperature decreases with elevation (Fig. C2). Unfortunately, this temperature effect is in the ***opposite direction*** as the observed depletion in moss $\Delta^{13}C$ with elevation (Fig. C1), indicating that decreased temperatures at higher elevations should lead to increased discrimination. According to our mean summer temperature reconstruction from tetraethers (14.1 to 17.5 °C), temperature variations over the Pliocene could result in a 0.2 ‰ response in $\Delta^{13}C$ and thus a fairly negligible effect on our empirical estimates of Pliocene $CO_2$ concentrations.

[Figure]

*Figure C2. Changes in moss $\Delta^{13}C$ as a function of temperature. Where the entire range of temperatures represents the lapse rate of moss samples from the Peruvian Andes and the red box represents the mean summer temperature range for the Arctic derived from our tetraether measurements.*

It was also suggested by Dr. Smittenberg that moss $\Delta^{13}C$ may vary as a function of decreasing partial pressure of $O_2$ in the atmosphere with elevation. We also tested this explanation, by revising the original BRYOCARB model so that it was formulated in terms of $pO_2$ instead of $O_2$ concentration, while holding all other variables constant. We found that $\Delta^{13}C$ does indeed change as a function of elevation; however, the relationship is ***negative*** (Fig. C3), in contrast to the positive relationship in response to $pCO_2$ (Fig C1). Furthermore, there is no evidence that the relatively high concentration of $O_2$ 21% was significantly different during the Pliocene. Therefore we do not suspect that changes in $pO_2$ have an appreciable effect on our $pCO_2$ reconstructions because they do not explain the observed increase in moss $\Delta^{13}C$ with elevation and probably did not change significantly at our site during the Pliocene.

[Figure]

*Figure C3. Changes in moss $\Delta^{13}C$ only as a function of $pO_2$ showing an increase in $\Delta^{13}C$ with elevation (i.e. reduced $pO_2$).*

We have included equations 1 and 2 simply to demonstrate the physical relationship between isotopic discrimination and the partial pressure of atmospheric $CO_2$. Dr. Smittenberg is correct in that we cannot simply replace pi/pa with pa and substitute equation 2 into equation 1, we have revised the text to reflect that these equations simply provide the physical basis used to derive an empirical transfer function to predict atmospheric $CO_2$ (P6 L13-15).

We have simplified our empirical transfer function, where we now use a linear function to predict $\Delta^{13}C$ moss from the partial pressure of atmospheric $CO_2$ across an elevational gradient (Fig. C1). We also compare our empirical transfer function with the theoretical predictions from the BRYOCARB model.

Lastly, we do not understand what the reviewer is implying by stating that our 'data is too compromised' by environmental factors to provide reasonable estimates of past $CO_2$ concentrations. It is not clear whether they are referring to our measurements, or the data that we have compiled from the literature. Our measurements are no more compromised than the previous studies that have used $^{13}C$ of mosses to estimate concentrations of atmospheric $CO_2$ in the past (White et al. 1994, Fletcher et al. 2008). We agree with Smittenberg that the assumptions made in estimating past $CO_2$ levels based on moss $^{13}C$ measurements contribute considerable error; however, the assumptions regarding how the diffusion of $CO_2$ into the organism is regulated by the partial pressure of $CO_2$ in its environment is the basis for many commonly used proxies of past $CO_2$, including isotopic measurements of alkenones (Pagani et al. 2010).

About the assessment of the Pliocene 13CO2 value using buckbean 13C values: this appears to be a reasonable approach, although it would be good to get this estimate confirmed by measuring some more plants. Note Stuiver&Braziunas (1989, Nature328, p58 Tree cellulose 13C/12C isotope ratios and climatic change) who observe a relation between latitude and fractionation (likely cause by changes in Temp and humidity).

RE: In order to estimate past $CO_2$ concentrations from isotopic discrimination in mosses, we must know the isotopic composition of source $CO_2$ against which mosses are discriminating. In order to do this we used the original approach of White et al. (1994), such that identifiable C3 paleo-vegetation was analyzed for its isotopic composition and compared to the isotopic composition of modern plants from the same taxonomic group. Although, we agree that we only used 4 modern buck bean (*Menyanthes trifoliata*) samples from different locations across the boreal forest to assess the stomatal sensitivity (i.e. pi/pa) to different environments, we found very little variability across these values of only 0.4 ‰. While Stuiver et al. analyze carbon isotopes in coniferous trees with a global distribution, the distribution of modern buckbean is mainly restricted to the boreal forest, so we don't suspect that the large climatic gradients of temperature and humidity identified as factors affecting conifer $\Delta^{13}C$ would have the same impact on buckbean. Furthermore, our estimates of atmospheric $\delta^{13}CO_2$ during the Pliocene are statistically indistinguishable from estimates derived from planktonic foraminifera (Ravelo et al. 2004).

About the measurements of the plant 13C values: why was chosen to measure cellulose instead of bulk tissue? What is the expected difference in 13C, knowing that sugars typically have less depleted 13C values than the bulk (lipids being more de- pleted)? The description of the preparation and isotope measurement methods is very limited and should get expanded.

RE: Essentially, because these samples were embedded in peat, they were quite dirty and we wanted to ensure that they were clean and that we were performing our isotopic measurements on the same

molecules across all samples.  More details have been added regarding the rational for sample preparation and hollocellulose extraction, as well as the reference providing details on the method (P5 L2).

Section 2.4 The sections about the brGDGTs as well as the one about fire, vegetation and climate are well written, and extensive and critical enough, and I have no real comments here. However, to make the jump from the local fire frequency at one location to the notion that fire could have been a global climate feedback mechanism during the Pliocene is a very large jump to conclusions (p19l12-16). It is fine to mention this possibility, but I would not use the word 'reveal' (line 12) but use a more careful wording (e.g. indicates, suggests).

RE: The wording here has been changed as suggested

P6l32. How 'well' are the brGDGTs really preserved?

RE: GDGTs are well preserved in anoxic sediments that have not experienced burial in the sub-surface (see Schouten et al., 2013, Org. Geochem. 54, 19-61 for a review). This applies for the Pliocene Arctic deposits investigated here.

P7l1 That the brGDGTs are 'thought to be sourced by a wide array of acidobacteria within the soil' is still under in- vestigation and there is still only scant evidence. For one, brGDGTs are also produced aquatically. Rephrase.

RE: The sentence has been rephrased.

P7l23. How was a concentration of 10 mg ml-1 (of brGDGTs, if one reads the text) made? Concentration of Total lipid extract or polar fraction?

RE: Based on the weight of the polar fraction. This is now clear from the modified sentence.

P7l25 mass spectrometry. UHPLC or HPLC?.

RE: This is now explained.

P7l28. From where does the transfer function error come?

RE: This sentence has been eliminated because the transfer functions are only introduced later. In the remainder of the text we do not discuss the errors in the transfer functions; we only provide the standard deviations of the estimated temperature from our sample set

P8l1 minus term missing

RE: This has been corrected.

3. Results P9l11. Not clear what the maximum probability of age of 4.5 Ma means, when earlier the most likely age is estimated at 3.9 Ma?

RE: The 3.9 MA is simply the unweighted mean of the 4 samples. The most probably age based on the convolution of all of their individual probability distribution functions is 4.5. In the next couple of sentences we explain that, because the PDF are not complex owing to reaching the radiodecay-based saturation of the isotopes, we recommend using the simple mean. This section has been reworded for clarity.

P9l19. Is an error of 104 to 105 years relevant on the geologic timescale of millions of years?

RE:104 and 105 years should have been $10^4$ and $10^5$ years. The superscripting is corrected at the earlier mention (Page 4), and this section has been rewritten for clarity.

Conclusions: Depending on any revision of the paper, alter or remove mention to past CO2 levels. Also be more careful in the conclusions with respect to the role of fire on climate. The paper showcases well that fire was part of the arctic climate – however this is not so different from the present day boreal realm, and the paper does not at all investigate, model or discuss this aspect. The same is true for the last sentence about present day arctic climate change, the paper does not focus at all on the present day arctic.

*RE:* In the revised manuscript we have provided two independent reconstructions of past $CO_2$ levels- one from the theoretical BRYOCARB model and the other from our empirical relationship. Both of these approaches have their assumptions and biases, so including both estimates provides a full range of potential $CO_2$ concentrations during the Pliocene.

The ramifications of fire for the radiative budget, and potential for inclusion in future models has been reduced in the conclusion. Extension to present-day climate-change has been removed.

Figure 3: No references given for the Polish and Hawaiian sites. Fig. 4 I would also plot the originally measured (estimated) Delta13C values,not only reconstructed pCO2 (but as stated above, I find that this aspect of the paper needs an overhaul in any case).

RE: Figure 3 has been revised and the references have been added.

**Elevated atmospheric $CO_2$ and fire linked to arctic amplification of temperature during the Early to mid-Pliocene**

Tamara Fletcher[1*], Lisa Warden[2*], Jaap S. Sinninghe Damsté[2,3], Kendrick J. Brown[4,5], Natalia Rybczynski[6,7], John Gosse[8], and Ashley P Ballantyne[1]

[1] College of Forestry and Conservation, University of Montana, Missoula, 59812, USA
[2] Department of Marine Microbiology and Biogeochemistry, NIOZ Royal Netherlands Institute for Sea Research, Den Berg, 1790, Netherlands
[3] Department of Earth Sciences, University of Utrecht, Utrecht, 3508, Netherlands
[4] Natural Resources Canada, Canadian Forest Service, Victoria, V8Z 1M, Canada
[5] Department of Earth, Environmental and Geographic Science, University of British Columbia Okanagan, Kelowna, V1V 1V7, Canada
[6] Department of Palaeobiology, Canadian Museum of Nature, Ottawa, K1P 6P4, Canada
[7] Department of Biology & Department of Earth Sciences, Carleton University, Ottawa, K1S 5B6, Canada
[8] Department of Earth Sciences, Dalhousie University, Halifax, B3H 4R2, Canada
*Authors contributed equally to this work

*Correspondence to*: Tamara Fletcher (tamara.fletcher@umontana.edu)

**Abstract.** The mid-Pliocene is a valuable time interval for understanding the mechanisms that determine equilibrium climate at current atmospheric $CO_2$ concentrations. One intriguing, but not fully understood, feature of the early to mid-Pliocene climate is the amplified arctic temperature response. Current models underestimate the degree of warming in the Pliocene Arctic and validation of proposed feedbacks is limited by scarce terrestrial records of climate and environment, as well as discrepancies in current $CO_2$ proxy reconstructions. Here we reconstruct the $CO_2$, summer temperature and fire regime from a sub-fossil fen-peat deposit on west-central Ellesmere Island, Canada, that has been chronologically constrained using radionuclide dating to 3.9 +1.5/-0.5 Ma.

An empirical transfer function was derived and applied to carbon isotopic measurements of paleo mosses to yield an estimate of Pliocene mean atmospheric $CO_2$ concentrations of $410 \pm 50$ ppm, which are slightly lower than theoretical model predictions of 510 
[revised manuscript text omitted]
 the fractionations of atmospheric CO$_2$ due to diffusion ($a = \sim$ -4.4 ‰) and carboxylation by the enzyme rubisco ($b = \sim$ -27 ‰) are constraints. Thus, isotopic fractionation in C3 plants ($\Delta$ $^{13}$C$_{C3}$) is largely a function of stomatal control of partial pressure of intercellular CO$_2$ – ($p_i$) with respect to the partial pressure of atmospheric CO$_2$ ($p_a$). However, bryophytes lack stomata and thus a mechanism for actively regulating $p_i$, such that isotopic fractionation ($\Delta$ $^{13}$C$_{bryo}$) varies mainly as a function of partial pressure in atmospheric CO$_2$ (i.e. $p_a$). While other environmental factors,

such as humidity, temperature, light availability, and microclimate may also play important roles in isotopic discrimination in bryophytes (Fletcher et al., 2008; Ménot and Burns, 2001; Royles et al., 2014; Skrzypek et al., 2007; Waite and Sack, 2011; White et al., 1994), the first order control on discrimination is the partial pressure of atmospheric $CO_2$ (Fletcher et al., 2008; White et al., 1994). Because atmospheric $CO_2$ is relatively well mixed in the troposphere its mean annual concentration does not differ significantly by location. However, because total atmospheric pressure decreases with atmospheric height ($h$), the partial pressure of atmospheric $CO_2$ must also decrease according to the following exponential function:

$$p_{a(h)} = p_{a(i)}e^{-h/H} \qquad (2)$$

such that the partial pressure of atmospheric $CO_2$ at any given height in the atmosphere ($p_{a(h)}$) can be calculated based on the initial atmospheric partial pressure of atmospheric $CO_2$ ($p_{a(i)}$) and a reference height ($H = 7600$ m), where atmospheric pressure goes to 0.37 Pa (Bonan, 2015). Therefore assuming that carbon isotopic discrimination in bryophytes varies in response to the partial pressure of atmospheric $CO_2$ we can predict from basic physical principles an increase in $\Delta^{13}C_{bryo}$ in response to an increase in $p_{a(h)}$. Furthermore, if the assumptions of this empirical relationship are valid, then this empirical relationship can in theory be used to predict the partial pressure of atmospheric $CO_2$ based on carbon isotopic measurements of bryophytes.

To test this prediction, we compiled data from four studies investigating carbon isotopic variability of different bryophytes, primarily mosses, along elevational transects at different locations. Based on the elevations and locations of moss samples, the atmospheric partial pressure of atmospheric $CO_2$ was estimated from ERA-interim reanalysis data of total atmospheric pressure (Dee et al., 2011) in conjunction with globally averaged atmospheric $CO_2$ concentrations (Global View-$CO_2$, 2013) from the years moss samples were collected. For our analysis we only included measurements of carbon isotopic variability in non-vascular mosses and all isotopic values were normalized to cellulose based on the empirical relationship reported by Ménot and Burns (2001). Carbon isotopic discrimination values for all plant material was calculated as:

$$\Delta^{13}C = (\delta^{13}C_{atm} - \delta^{13}C_{plant})/(1 + \delta^{13}C_{plant}/1000) \qquad (3)$$

where $\delta^{13}C_{plant}$ represents the C isotopic composition of plant cellulose and $\delta^{13}C_{atm}$ represents the mean annual carbon isotopic composition of atmospheric $CO_2$ of the year when samples were collected (Global View-$CO_2$, 2013), or in the case of sub-fossil mosses, when the samples were growing. The response $\Delta^{13}C$ to $pCO_2$ across elevational gradients in modern mosses was then used to calibrate the theoretical model BRYOCARB that has been developed to reconstruct past $CO_2$ levels based on measurements of $^{13}C$ in paleo bryophytes. Thus, our approach provides two independent estimates of Pliocene $CO_2$ concentrations – one empirically derived from our transfer function and the other predicted from the BRYOCARB model calibrated to modern mosses and constrained by our paleoclimate reconstructions.

In order to derive estimates of atmospheric $CO_2$ concentrations during the Pliocene, isotopic composition of source $CO_2$ from the atmosphere (ie. $\delta^{13}C_{atm}$) was estimated during the Pliocene to solve for $\Delta^{13}C$ of mosses (Eq.(3)). This was accomplished by simultaneous measurements of $\delta^{13}C$ in the C3 plant buckbean (*Menyanthes trifoliata* L.) that was identified as a subfossil specimen at the BP site and was also collected from four different sites in the Canadian Boreal Forest. Although it has been demonstrated that $\Delta^{13}C$ of C3 plant material is sensitive to many factors, including mean annual precipitation and altitude (Diefendorf et al. 2010), there is less variability within biomes, so modern buckbean was sampled from within the Canadian Boreal biome that we suspect is very similar to the BP site based upon paleovegetation (Ballantyne et al. 2010, Fletcher et al. 2017). Measurements of $\delta^{13}C$ on modern buckbean were used to constrain estimates of $p_i / p_a$ using modern estimates of $\delta^{13}C_{atm}$ from when the buckbeans were collected. This constrained modern value of $p_i / p_a$ was then applied to our sub-fossil buckbean samples to estimate $\delta^{13}C_{atm}$ during the Pliocene. All plant and moss material were rinsed and placed in a sonicating bath with deionized water to remove any paleosoil from samples. The diagnostic material for mosses was leafy material, whereas buck bean was identified base on seeds. Therefore, to ensure that our isotopic measurements were made on similar compounds, cellulose was extracted from samples according to 
[revised manuscript text omitted]
$_2$ both in empirical observations and theoretical predictions (Fig. 3). However, a much greater change in $\Delta^{13}C_{moss}$ is observed in response to $p_a$ than is predicted from the optimized BRYOCARB simulations. The empirical fit to the observed change in $\Delta^{13}C_{moss}$ in response to $p_a$ is slightly better (RMSE = 1.8 ‰) than the theoretical prediction from the BRYOCARB model (RMSE = 2.1 ‰), but the slopes are quite different, with our empirical slope (0.56 ‰/$p_a$) an order of magnitude greater than the linear approximation of the BRYOCARB slope (0.07 ‰/$p_a$), suggesting that other non-linear processes and not just $p_a$ may be affecting $\delta^{13}C_{moss}$ variability with elevation.

While there does appear to be a global relationship between $p_a$ and $\Delta^{13}C$ of mosses, there are notable differences among sites. Moss $\Delta^{13}C$ values tended to be generally lower in the Swiss Alps (mean = 17.4 ‰) and higher in Hawaii (mean = 20.6 ‰) and the slope of the relationship between $p_a$ and $\Delta^{13}C$ appears to vary across sites with the Andes having the smallest slope and Poland having a much greater slope. We used the BRYOCARB model to test the sensitivity of $\Delta^{13}C$ to other variables that change as a function of elevation (e.g. temperature and $pO_2$). According to our BRYOCARB simulations, with all other variables held constant decreased temperature with increased elevation should slow metabolic rates resulting in an increase in $\Delta^{13}C$ (Supplemental Figure S3), which directly contradicts observations (Fig. 3). Furthermore, the range of mean summer temperature estimates from the Pliocene BP site could only explain ~0.2 ‰ isotopic response in our moss samples. Similarly we evaluated the effect of just changing $pO_2$ in our BRYOCARB simulations and found a decrease in $\Delta^{13}C$ with increasing $pO_2$ that is opposite to the $\Delta^{13}C$ response of mosses to partial pressure across all elevational transects. We also evaluated model performance using a global standard atmospheric sea level pressure of 101.325 kPa, or site-specific atmospheric pressure estimates from ERA-interim reanalysis data. We found that the model using site specific atmospheric pressure estimates performed better at predicting $\Delta^{13}C_{moss}$ (RMSE = 1.096 ‰) than the model using global standard atmospheric sea level pressure (RMSE = 1.216 ‰). Therefore, it appears that partial pressure of atmospheric CO$_2$ is the primary physical mechanism explaining the global relationship between $\Delta^{13}C$ of mosses and elevation and that other factors, such as water availability that may be mediated by different lapse rates (Ménot and Burns, 2001; Royles et al., 2014; Skrzypek et al., 2007; Waite and Sack, 2011), may explain variability among sites. Thus, the optimal model characterizing the observed modern relationship between $\Delta^{13}C_{moss}$ and the $p_a$ was:

$$13C\ \Delta13C_{moss}\ =\ 0.56\ x\ pCO_2\ +\ 1.55 \tag{9}$$

Based on our analysis of cellulose extracted from four different *Menyanthes* L. (i.e. buckbean) plants growing at four different locations in the modern boreal forest, we found $\Delta^{13}C$ of buckbean to be fairly constant 16 ± 0.4 ‰, yielding an estimate of $p_i / p_a$ in modern buckbean of 0.51. Applying this modern of $p_i / p_a$ to our $\delta^{13}C$ measurements from sub-fossil buckbean we obtained estimates of $\delta^{13}C_{atm}$ during the Pliocene of -6.23 ± 0.9 ‰. Using our empirical

transfer function (Eq. 9) in combination with these estimates of $\delta^{13}C_{atm}$, we were able to approximate atmospheric $CO_2$ concentrations over the Pliocene interval captured at the BP site (Fig. 4). We estimated a mean atmospheric $CO_2$ concentration over this interval of $410 \pm 50$ ppm (mean ± transfer error and instrument error) with considerable variability between a minimum atmospheric $CO_2$ concentration of $296$ ppm and a maximum atmospheric $CO_2$ concentration of $480$ ppm. Predicted values of Pliocene $CO_2$ from the BRYOCARB model were slightly higher at $510$ ppm, but the single standard deviation across all estimates was extremely high (967 ppm), suggesting that the BRYOCARB simulations are not significantly different from our empirical model estimates; however, the BRYOCARB model is too sensitive to our range of $\Delta^{13}C_{moss}$ estimates and thus not very precise.

[revised manuscript text omitted]
 transects in the Swiss Alps (black dots; Ménot and Burns, 2001), the Peruvian Andes (blue dots; Royles et al., 2014) ), the mountains of Poland (red dots; Skrzypek et al. 2007), and Hawaii (green dots; Waite and Sack 2011). Partial pressure of atmospheric CO₂ calculated from atmospheric surface pressure reanalysis data (Dee et al., 2011) combined with atmospheric CO₂ observations from year moss samples were collected. All carbon isotopic measurements of mosses have been normalized to cellulose based on published regression of cellulose and whole moss values (Ménot and Burns, 2001) and reported as discrimination (Δ) from atmospheric δ¹³CO₂ (GlobalGlobal View-CO₂, 2013) from the year mosses were collected in units of ‰. Empirical model fit (black line) is plotted with prediction intervals (black dashed) compared with predictions from the BRYOCARB model (Fletcher et al. 2008) with parameters optimized to match observations.**

[Figure]

**Figure 4. Reconstruction of atmospheric CO₂, mean summer temperature, and fire for the Canadian High Arctic during the Pliocene.** Atmospheric $CO_2$ concentrations estimated from carbon isotopic measurements of mosses and plants (red; ± 2 σ). Mean summer temperature reconstructed from a brGDGT based proxy (blue; ± 2 σ) and relative 2010 data point in approximate relative position (purple; ± 2 σ). Charcoal counts reported as the number of fragments per volume (fragments cm⁻³) of peat (Orange ± 2 σ). Green boxes indicate relative depths of pollen sampling. Elevation of the deposit is reported as meters above sea level. (Data: Table S2)

**(A)**

[Figure]

**(B)**

Figure 5. (A) Bar charts showing the relative pollen abundance in each portion of the section (error bars = 95% confidence intervals; MASL- Meters Above Sea Level). (B). Pollen plate of select grains encountered in the BP section: (a) *Pinus*, (b) half a *Picea* grain, (c) *Larix*, (d) *Betula*, (e) *Alnus*, (f) *Salix*, (g) *Myrica*, (h) ericaceous grain, (i) *Epilobium*, and (j) Cyperaceae.  50um scale = (a–c), 75um scale = (d–j).

[Figure]

**Figure 6. A ternary plot illustrating the fractional abundances of the tetra- (Ia-c), penta (IIa-c and II′a-c), and hexamethylated (IIIa-c and III′a-c) brGDGTs. The global soil dataset (open circles; De Jonge et al., 2014), the global peat samples (green circles; Naafs et al., 2017), and lake sediments from East Africa (black circles indicate samples from lakes >20°C, red circles indicate samples from lakes between 10–20°C and orange circles designate samples from lakes <10°C; Russell et al., 2018) are included for comparison with the Beaver Pond sediments (blue circles; this study).**

[Figure]

**Figure 7. (a) Modern geographic distribution of observed occurrences of species common to the Beaver Pond species list, (b) Mean temperature of the warmest quarter (summer average) derived from WorldClim, (c) Mean precipitation of the warmest quarter (summer rain) derived from WorldClim, (d) Count of unique fire pixels detected per day, over 10 years from MODIS 6 Fire Product, normalized by area of the latitude by longitude grid.**

**Supplementary Information**

[Figure]

**Figure S1. Molecular structures of all 15 brGDGTs (I-III). The molecules designated with a prime symbol are referred to as the 6-methyl brGDGTs.**

[Figure]

**Figure S2. A comparison of the (A) area and (B) shape (length to width) of the uppermost samples (130–165) that have a higher mean charcoal concentration, and the lowermost samples (0–125)**

[Figure]

**Figure S3. A)** Changes in moss $\Delta^{13}C$ as a function of temperature. Where the entire range of temperatures represents the lapse rate of moss samples from the Peruvian Andes and the red box represents the mean summer temperature range for the Arctic derived from our tetraether measurements. **B)** Changes in moss $\Delta^{13}C$ only as a function of $pO_2$ showing an increase in $\Delta^{13}C$ with elevation (i.e. reduced $pO_2$).

[revised manuscript text omitted]

---

## Author Comment (AC3) · 15 Oct 2018

Fletcher and colleagues present a Pliocene high arctic record of CO2, temperature, plant species composition, and inferred fire frequency. They then explore how these components may be interconnected. The study summarizes an impressive amount of data. My expertise lies with paleo-CO2 reconstruction and so my review will focus on there.

1. CO2 reconstruction.
-Ben Fletcher developed a process-based model for paleo-CO2 reconstruction based on the δ13C of liverworts. I'm surprised that the authors have not tried to incorporate/modify this method for their own study. It's not even mentioned! Instead, the authors rely on a present-day empirical-based model, which is likely to be inferior to a process-based model. To give just one example, the authors note the problem of growth rates with other paleo-CO2 methods (p. 15, line 10). But growth rate is a key uncertainty with the authors' method, and something that is acknowledged and (partially) addressed in the Fletcher model. This is a key deficiency with the current manuscript.

RE: We are familiar with the BRYOCARB model of Dr. Fletcher and we have incorporated it into our revised analysis. While our empirical model clearly has many errors associated with it, there are also numerous assumptions that must be made to apply a modern physiological model to a paleoenvironment in which many parameter values are hard to constrain. Nonetheless, both our empirical model and the theoretical BRYOCARB model show an increase in $\Delta^{13}C$ moss with increased partial pressure of atmospheric $CO_2$ as would be expected based on physical principles. However, the BRYOCARB is much less sensitive to $CO_2$ over the range of $pCO_2$ than the observed variations in samples among the elevational transects. BRYOCARB does enable us to eliminate possible explanations, such as changes in temperature and $pO_2$ with elevation, that have been proposed to explain the apparent decrease in $\Delta^{13}C$ of mosses with elevation. We are able to optimize variables in the BRYOCARB model to provide a plausible mean estimate of Pliocene $CO_2$ of 510 ppm, but the standard deviation of these estimates is extremely high ± 1080 due to the large change in $pCO_2$ in response to a small change in $\Delta^{13}C$ moss as predicted by the BRYOCARB model. This extremely wide range of $pCO_2$ estimates is still the result when unrealistically large values of $\Delta^{13}C$ (>30‰) are excluded from our Pliocene moss dataset.

Fletcher, B. J., Beerling, D. J., Royer, D. L., and Brentnall, S. J., 2005, Fos- sil brophytes as recorders of ancient CO2 levels: Experimental evidence and a Cretaceous case study: Global Biogeochemical Cycles, v. 19, p. GB3012, doi:3010.1029/2005GB002495.

Fletcher, B. J., Brentnall, S. J., Quick, W. P., and Beerling, D. J., 2006, BRYOCARB: A process-based model of thallose liverwort carbon isotope fractionation in response to CO2, O2, light and temperature: Geochimica et Cosmochimica Acta, v. 70, p. 5676- 5691.

-Using leaf δ13C to reconstruct air δ13C is problematic because many factors—for example water stress—can affect leaf δ13C. The authors are assuming no change in water stress (and other factors that could affect leaf δ13C) between the present-day and Pliocene. Given what is said in section 4.3, this assumption is tenuous.

RE: We acknowledge that this is a limitation to our approach and differences in mean annual precipitation (MAP) may affect our estimates of modern $\Delta\ ^{13}C$. However, we have controlled for this large scale variability in only sampling modern buckbean from the same boreal biome, which is much less variable than the global patterns of $\Delta\ ^{13}C$ in response to global MAP. We also suspect that our fossil BP site was very similar to the modern Boreal forest based on our assessment of paleovegetation. This section has been updated (P7 L6-22) and the relevant Diefendorf citation has been added.

Diefendorf, A. F., Mueller, K. E., Wing, S. L., Koch, P. L., and Freeman, K. H., 2010, Global patterns in leaf 13C discrimination and implications for studies of past and future climate: Proceedings of the National Academy of Sciences, USA, v. 107, p. 5738-5743.

Kohn, M. J., 2010, Carbon isotope compositions of terrestrial C3 plants as indicators of (paleo)ecology and (paleo)climate: Proceedings of the National Academy of Sciences, USA, v. 107, p. 19691-19695.

-The empirical transfer function (Figure 3) maxes out at 360 ppm. The authors use the function to reconstruct ~450 ppm. This is a problem with extrapolation.

-The empirical transfer function is based on a mix of species. Some of the scatter is likely due to "vital effects". This needs to be acknowledged. The best transfer function would be one based on the same species (or genus) as the fossil material.

-The authors have underestimated the uncertainty associated with their paleo-CO2 re-constructions. As best as I can tell, their stated uncertainty (1ïA'ß'= 35 ppm) is the confidence interval from Figure 3 (dashed lines). A confidence interval says how con- fident one is in the regression. But if one wishes to infer the y-axis value from a new single data point (as is being done here), the prediction interval is appropriate. And the prediction interval is wider than the confidence interval. In addition, the authors have not propagated uncertainty associated with the measurement(s) of leaf δ13C at each level; the authors are assuming no error. Beerling et al. (2009) lays out a solid strategy for propagating uncertainty with these kind of empirical functions.

RE: We have revised figure 3 to include the prediction interval and not the confidence interval. We have also included instrumental error with transfer function error in quadrature for our overall estimates of Pliocene $CO_2$ that have been updated in figure 4. We also report the mean, standard deviation and range of estimates in the revised paper.

Beerling, D. J., Fox, A., and Anderson, C. W., 2009, Quantitative uncertainty analyses of ancient atmospheric CO2 estimates from fossil leaves: American Journal of Science, v. 309, p. 775-787.

-The authors deal with the confounding factor of water stress in the Discussion, but this section should move to the Introduction. Otherwise, the informed reader will be wondering why the authors haven't dealt with the issue while they are reading the Intro, Methods, and Results.

RE This is dealt with in the revised results, when we discuss the BRYOCARB simulations.

2. CO2 compilation.

-The B/Ca estimates should be excluded as they are not reliable.

Allen, K. A., and Hönisch, B., 2012, The planktic foraminiferal B/Ca proxy for seawater carbonate chemistry: a critical evaluation: Earth and Planetary Science Letters, v. 345–348, p. 203-211.

-It looks like some Pliocene estimates have been missed. See Foster et al. (2017) for compilation and citations.

Badger, M. P. S., Schmidt, D. N., Mackensen, A., and Pancost, R. D., 2013b, High-resolution alkenone palaeobarometry indicates relatively stable pCO2 during the Pliocene(3.3–2.8âĽ'Ma):PhilosophicalTransactionsoftheRoyalSocietyA,v.371, 20130094.

Bartoli, G., Hönisch, B., and Zeebe, R.E., 2011, Atmospheric CO2 decline during the Pliocene intensification of Northern Hemisphere glaciations: Paleoceanography, v. 26, PA4213, doi:10.1029/2010PA002055.

Foster, G. L., Royer, D. L., and Lunt, D. J., 2017, Future climate forcing potentially without precedent in the last 420 million years: Nature Communications, v. 8, p. 14845, doi:14810.11038/ncomms14845.

Martínez-Botí, M. A., Foster, G. L., Chalk, T. B., Rohling, E. J., Sexton, P. F., Lunt, D. J., Pancost, R. D., Badger, M. P. S., and Schmidt, D. N., 2015, Plio-Pleistocene climate sensitivity evaluated using high-resolution CO2 records: Nature, v. 518, p. 49-54.

Seki, O., Foster, G.L., Schmidt, D.N., Mackensen, A., Kawamura, K., and Pancost, R.D., 2010, Alkenone and boron-based Pliocene pCO2 records: Earth and Planetary Science Letters, v. 292, p. 201-211.

Stap, L. B., de Boer, B., Ziegler, M., Bintanja, R., Lourens, L. J., and van de Wal, R. S. W., 2016, CO2 over the past 5 million years: continuous simulation and new δ11B- based proxy data: Earth and Planetary Science Letters, v. 439, p. 1-10.

Zhang, Y. G., Pagani, M., Liu, Z., Bohaty, S. M., and DeConto, R., 2013, A 40-million- year history of atmospheric CO2: Philosophical Transactions of the Royal Society A, v. 371, 20130096.

RE: Thank you for bringing these more recent Pliocene paleo $CO_2$ reconstructions to our attention.  They have been added to our compilation and Figure 1 has been revised.

3. Temperature component.

-The temperature record feels like a "third wheel" to the CO2 and fire records. In the discussion, other temperature records (often from the same site) are emphasized more than the record generated here.

RE: The temperature record produced from the bacterial tetraethers requires less discussion than the charcoal and $CO_2$ because it is an existing, validated method, its results were clear, and their implication for fire etc (see below) seems straight forward. The mention of previous temperature estimates was to say they are consistent with these records – these have been removed to focus more on the GDGT results. The pollen-record suggesting a possible small-change between the

low and high charcoal components of the sequence was a simple analysis within this study, not the result of a previous study. Climate data from previous studies is only introduced for climate variables that were not measured in this study – Summer precipitation.

4. Link between fire and climate.

-Quite a lot of space in the Discussion is devoted to how fire and climate are inter- connected. And the bulk of this discussion centers on the literature. But, the record generated here shows no obvious link between fire and $CO_2$ or temperature (Figure 4). As a result, there is a logical disconnect. For example, from the Introduction (p. 2, lines 30-31): "We propose that fire in arctic ecosystems may also be an important proximal mechanism for amplifying arctic surface temperatures during the Pliocene."

RE: We have attempted to clarify the importance of the '~13.5 °C as a key threshold for fire' for which the GDGT's provide evidence, through reordering of this section and some rewording of the material.. Without high enough temperatures, the fires would be much more unlikely. However, once this threshold is crossed, differences in moisture and vegetation seem to have a larger impact on fire as suggested by charcoal deposition.

We have clarified at p. 2, lines 30-31 that the potential role of fire in Arctic amplification is part of the motivation for the study rather than a hypothesis we test.

Minor comments:
p. 3, line 15: B/Ca (not "Boron")

RE: Corrected

p. 3, line 19: Foster et al. (2017, Nature Communications) is a more current reference

RE: This reference has been updated.

p. 3, line 24: what do you mean by "Although direct effects may be small"?

RE: Clarified in the text.

p. 4, lines 14-16: "The unit sampled spanned the 1 m remaining of Unit III as per Mitchell et al. (2016). The main sequence examined across the methods used in this study includes material above (Unit IV) and below (Unit II) Unit III, with a total sampled profile of 1.65 m." Parts of these sentences are confusing: 'unit' appears to have a different meaning as 'Unit'; what does '1 m remaining of Unit III' mean? (is most of the originally sampled material gone?); the first sentence implies that all of the data come from Unit III, but the second sentence says that some of Units II and IV are included too.

RE: These sections have been edited to improve clarity

p. 6, line 31: "We also measured δ13C of modern buckbean to constrain our estimates of pi / pa." More context is needed so that the reader can understand this statement. Why do the pi/pa estimates need to be 'constrained', and why do present-day measurements allow you to do this?

RE: This section has been clarified in the revised version (P7 L1-10)

p. 6, line 32: say that cellulose was measured (ditto in line 31).

RE: The suggested changes have been made.

p. 7, line 11: "paleosols"

RE: The suggested change has been made.

p. 7, lines 30-32: Provide a citation for the transfer function. Is the combined error a one-sigma error? Two-sigma? Was quadrature used to calculate the combined error?

RE: See explanation for the comment of referee #1. We don't use this.

p. 8, line 19: "MAT"?

RE: The suggested change has been made.

p. 9, lines 17-18: What is the difference between a maximum probability age and an optimized age?

RE: We have deleted reference to the optimized age as the Bayesian most probable age is now most frequently used.

p. 9, line 22: Unweighted mean age already stated. p. 9, line 23: How was this uncertainty computed?

RE: This section has now been reworded for clarity in response to reviewer 1.

p. 9, lines 28-30: The first and second parts of this sentence are saying the same thing.

RE: This section has been edited.

p. 9, lines 30-31: I don't understand why nonlinearity is expected because Figure 3 plots the log of carbon isotope discrimination (also, see next sentence in the manuscript).

RE: A linear model is now applied in the revised manuscript.

p. 10, line 2: These "other processes" don't need to be nonlinear; the combined additive effect is nonlinear.

RE: A linear model is now applied in the revised manuscript.

p. 10, lines 9-12: It is inappropriate to use the site-specific regression because the associated site-specific information is not available for the Pliocene samples.

RE: This section has been edited.

p. 10, line 11: Is this the same model as in line 1?

RE: This section has been edited as a linear model is now applied in the revised manuscript.

p. 15, line 23: "over the Pliocene". Surely you don't mean the entire Pliocene?

RE: This has now been edited for clarity 'over the deposition of this sediment during the Pliocene'

p. 19, lines 30-31: "fire played an active role in influencing the climate of the Arctic during the Pliocene." That's not what your data suggest (Figure 4).

RE: It is true that the apparent change in fire frequency at our site did not result in a change in a respective change in climate. This section has been edited to better reflect our results – that a change in fire seems to drive a change in environment, and that in the similar modern boreal system this has impacts on the Earth's radiative budget.

Figure 3: Please add the theoretical regression (from equations 2 & 3). Add linear

y-axis tick marks too. In the caption, say that the isotopic discrimination is based on cellulose (or an inferred cellulose value).

RE: This figure has been revised to include theoretical fit from BRYOCARB along with our empirical fit to observations. Scale is now linear based on comments from Smittenberg (above) and y-ticks are included. Caption has been revised stating that all values normalized to cellulose.

Figure 4: unit needed for x-axis.

RE: The suggested change has been made.

Figure S2: Plots need axis labels

RE: Figure S2 has been revised, and now has clear axis labels

Figure S3: Axis and tick mark labels are too pixelated to read

RE: A higher resolution version of this figure is available.

*Authors contributed equally to this work

*Correspondence to*: Tamara Fletcher (tamara.fletcher@umontana.edu)

**Abstract.** The mid-Pliocene is a valuable time interval for understanding the mechanisms that determine equilibrium climate at current atmospheric $CO_2$ concentrations. One intriguing, but not fully understood, feature of the early to mid-Pliocene climate is the amplified arctic temperature response. Current models underestimate the degree of warming in the Pliocene Arctic and validation of proposed feedbacks is limited by scarce terrestrial records of climate and environment, as well as discrepancies in current $CO_2$ proxy reconstructions. Here we reconstruct the $CO_2$, summer temperature and fire regime from a sub-fossil fen-peat deposit on west-central Ellesmere Island, Canada, that has been chronologically constrained using radionuclide dating to 3.9 +1.5/-0.5 Ma. An empirical transfer function was derived and applied to carbon isotopic measurements of paleo mosses to yield an estimate of Pliocene mean atmospheric $CO_2$ concentrations of $410 \pm 50$ ppm, which are slightly lower than theoretical model predictions of 510 
[revised manuscript text omitted]

where the fractionations of atmospheric CO$_2$ due to diffusion ($a = \sim$ -4.4 ‰) and carboxylation by the enzyme rubisco ($b = \sim$ -27 ‰) are constraints. Thus, isotopic fractionation in C3 plants ($\Delta$ $^{13}$C$_{C3}$) is largely a function of stomatal control of partial pressure of intercellular CO$_2$ – ($p_i$) with respect to the partial pressure of atmospheric CO$_2$ ($p_a$). However, bryophytes lack stomata and thus a mechanism for actively regulating $p_i$, such that isotopic fractionation ($\Delta$ $^{13}$C$_{bryo}$) varies mainly as a function of partial pressure in atmospheric CO$_2$ (i.e. $p_a$). While other environmental factors,

such as humidity, temperature, light availability, and microclimate may also play important roles in isotopic discrimination in bryophytes (Fletcher et al., 2008; Ménot and Burns, 2001; Royles et al., 2014; Skrzypek et al., 2007; Waite and Sack, 2011; White et al., 1994), the first order control on discrimination is the partial pressure of atmospheric $CO_2$ (Fletcher et al., 2008; White et al., 1994). Because atmospheric $CO_2$ is relatively well mixed in the troposphere its mean annual concentration does not differ significantly by location. However, because total atmospheric pressure decreases with atmospheric height ($h$), the partial pressure of atmospheric $CO_2$ must also decrease according to the following exponential function:

$$p_{a(h)} = p_{a(i)}e^{-h/H} \qquad (2)$$

such that the partial pressure of atmospheric $CO_2$ at any given height in the atmosphere ($p_{a(h)}$) can be calculated based on the initial atmospheric partial pressure of atmospheric $CO_2$ ($p_{a(i)}$) and a reference height ($H = 7600$ m), where atmospheric pressure goes to 0.37 Pa (Bonan, 2015). Therefore assuming that carbon isotopic discrimination in bryophytes varies in response to the partial pressure of atmospheric $CO_2$ we can predict from basic physical principles an increase in $\Delta^{13}C_{bryo}$ in response to an increase in $p_{a(h)}$. Furthermore, if the assumptions of this empirical relationship are valid, then this empirical relationship can in theory be used to predict the partial pressure of atmospheric $CO_2$ based on carbon isotopic measurements of bryophytes.

To test this prediction, we compiled data from four studies investigating carbon isotopic variability of different bryophytes, primarily mosses, along elevational transects at different locations. Based on the elevations and locations of moss samples, the atmospheric partial pressure of atmospheric $CO_2$ was estimated from ERA-interim reanalysis data of total atmospheric pressure (Dee et al., 2011) in conjunction with globally averaged atmospheric $CO_2$ concentrations (Global View-$CO_2$, 2013) from the years moss samples were collected. For our analysis we only included measurements of carbon isotopic variability in non-vascular mosses and all isotopic values were normalized to cellulose based on the empirical relationship reported by Ménot and Burns (2001). Carbon isotopic discrimination values for all plant material was calculated as:

$$\Delta^{13}C = (\delta^{13}C_{atm} - \delta^{13}C_{plant})/(1 + \delta^{13}C_{plant}/1000) \qquad (3)$$

where $\delta^{13}C_{plant}$ represents the C isotopic composition of plant cellulose and $\delta^{13}C_{atm}$ represents the mean annual carbon isotopic composition of atmospheric $CO_2$ of the year when samples were collected (Global View-$CO_2$, 2013), or in the case of sub-fossil mosses, when the samples were growing. The response $\Delta^{13}C$ to $pCO_2$ across elevational gradients in modern mosses was then used to calibrate the theoretical model BRYOCARB that has been developed to reconstruct past $CO_2$ levels based on measurements of $^{13}C$ in paleo bryophytes. Thus, our approach provides two independent estimates of Pliocene $CO_2$ concentrations – one empirically derived from our transfer function and the other predicted from the BRYOCARB model calibrated to modern mosses and constrained by our paleoclimate reconstructions.

In order to derive estimates of atmospheric $CO_2$ concentrations during the Pliocene, isotopic composition of source $CO_2$ from the atmosphere (ie. $\delta^{13}C_{atm}$) was estimated during the Pliocene to solve for $\Delta^{13}C$ of mosses (Eq.(3)). This was accomplished by simultaneous measurements of $\delta^{13}C$ in the C3 plant buckbean (*Menyanthes trifoliata* L.) that was identified as a subfossil specimen at the BP site and was also collected from four different sites in the Canadian Boreal Forest. Although it has been demonstrated that $\Delta^{13}C$ of C3 plant material is sensitive to many factors, including mean annual precipitation and altitude (Diefendorf et al. 2010), there is less variability within biomes, so modern buckbean was sampled from within the Canadian Boreal biome that we suspect is very similar to the BP site based upon paleovegetation (Ballantyne et al. 2010, Fletcher et al. 2017). Measurements of $\delta^{13}C$ on modern buckbean were used to constrain estimates of $p_i / p_a$ using modern estimates of $\delta^{13}C_{atm}$ from when the buckbeans were collected. This constrained modern value of $p_i / p_a$ was then applied to our sub-fossil buckbean samples to estimate $\delta^{13}C_{atm}$ during the Pliocene. All plant and moss material were rinsed and placed in a sonicating bath with deionized water to remove any paleosoil from samples. The diagnostic material for mosses was leafy material, whereas buck bean was identified base on seeds. Therefore, to ensure that our isotopic measurements were made on similar compounds, cellulose was extracted from samples according to 
[revised manuscript text omitted]
_2$ both in empirical observations and theoretical predictions (Fig. 3). However, a much greater change in $\Delta^{13}C_{moss}$ is observed in response to $p_a$ than is predicted from the optimized BRYOCARB simulations. The empirical fit to the observed change in $\Delta^{13}C_{moss}$ in response to $p_a$ is slightly better (RMSE = 1.8 ‰) than the theoretical prediction from the BRYOCARB model (RMSE = 2.1 ‰), but the slopes are quite different, with our empirical slope (0.56 ‰/$p_a$) an order of magnitude greater than the linear approximation of the BRYOCARB slope (0.07 ‰/$p_a$), suggesting that other non-linear processes and not just $p_a$ may be affecting $\delta^{13}C_{moss}$ variability with elevation.

While there does appear to be a global relationship between $p_a$ and $\Delta^{13}C$ of mosses, there are notable differences among sites. Moss $\Delta^{13}C$ values tended to be generally lower in the Swiss Alps (mean = 17.4 ‰) and higher in Hawaii (mean = 20.6 ‰) and the slope of the relationship between $p_a$ and $\Delta^{13}C$ appears to vary across sites with the Andes having the smallest slope and Poland having a much greater slope. We used the BRYOCARB model to test the sensitivity of $\Delta^{13}C$ to other variables that change as a function of elevation (e.g. temperature and $pO_2$). According to our BRYOCARB simulations, with all other variables held constant decreased temperature with increased elevation should slow metabolic rates resulting in an increase in $\Delta^{13}C$ (Supplemental Figure S3), which directly contradicts observations (Fig. 3). Furthermore, the range of mean summer temperature estimates from the Pliocene BP site could only explain ~0.2 ‰ isotopic response in our moss samples. Similarly we evaluated the effect of just changing $pO_2$ in our BRYOCARB simulations and found a decrease in $\Delta^{13}C$ with increasing $pO_2$ that is opposite to the $\Delta^{13}C$ response of mosses to partial pressure across all elevational transects. We also evaluated model performance using a global standard atmospheric sea level pressure of 101.325 kPa, or site-specific atmospheric pressure estimates from ERA-interim reanalysis data. We found that the model using site specific atmospheric pressure estimates performed better at predicting $\Delta^{13}C_{moss}$ (RMSE = 1.096 ‰) than the model using global standard atmospheric sea level pressure (RMSE = 1.216 ‰). Therefore, it appears that partial pressure of atmospheric $CO_2$ is the primary physical mechanism explaining the global relationship between $\Delta^{13}C$ of mosses and elevation and that other factors, such as water availability that may be mediated by different lapse rates (Ménot and Burns, 2001; Royles et al., 2014; Skrzypek et al., 2007; Waite and Sack, 2011), may explain variability among sites. Thus, the optimal model characterizing the observed modern relationship between $\Delta^{13}C_{moss}$ and the $p_a$ was:

$$13C\ \Delta13C_{moss} = 0.56 \times pCO_2 + 1.55 \tag{9}$$

Based on our analysis of cellulose extracted from four different *Menyanthes* L. (i.e. buckbean) plants growing at four different locations in the modern boreal forest, we found $\Delta^{13}C$ of buckbean to be fairly constant 16 ± 0.4 ‰, yielding an estimate of $p_i / p_a$ in modern buckbean of 0.51. Applying this modern of $p_i / p_a$ to our $\delta^{13}C$ measurements from sub-fossil buckbean we obtained estimates of $\delta^{13}C_{atm}$ during the Pliocene of -6.23 ± 0.9 ‰. Using our empirical

transfer function (Eq. 9) in combination with these estimates of $\delta^{13}C_{atm}$, we were able to approximate atmospheric $CO_2$ concentrations over the Pliocene interval captured at the BP site (Fig. 4). We estimated a mean atmospheric $CO_2$ concentration over this interval of $410 \pm 50$ ppm (mean ± transfer error and instrument error) with considerable variability between a minimum atmospheric $CO_2$ concentration of 296 ppm and a maximum atmospheric $CO_2$ concentration of 480 ppm. Predicted values of Pliocene $CO_2$ from the BRYOCARB model were slightly higher at 510 ppm, but the single standard deviation across all estimates was extremely high (967 ppm), suggesting that the BRYOCARB simulations are not significantly different from our empirical model estimates; however, the BRYOCARB model is too sensitive to our range of $\Delta^{13}C_{moss}$ estimates and thus not very precise.

[revised manuscript text omitted]
 transects in the Swiss Alps (black dots; Ménot and Burns, 2001), the Peruvian Andes (blue dots; Royles et al., 2014) ), the mountains of Poland (red dots; Skrzypek et al. 2007), and Hawaii (green dots; Waite and Sack 2011). Partial pressure of atmospheric CO₂ calculated from atmospheric surface pressure reanalysis data (Dee et al., 2011) combined with atmospheric CO₂ observations from year moss samples were collected. All carbon isotopic measurements of mosses have been normalized to cellulose based on published regression of cellulose and whole moss values (Ménot and Burns, 2001) and reported as discrimination (Δ) from atmospheric δ¹³CO₂ (GlobalGlobal View-CO₂, 2013) from the year mosses were collected in units of ‰. Empirical model fit (black line) is plotted with prediction intervals (black dashed) compared with predictions from the BRYOCARB model (Fletcher et al. 2008) with parameters optimized to match observations.**

[Figure]

**Figure 4. Reconstruction of atmospheric CO₂, mean summer temperature, and fire for the Canadian High Arctic during the Pliocene.** Atmospheric $CO_2$ concentrations estimated from carbon isotopic measurements of mosses and plants (red; ± 2 σ). Mean summer temperature reconstructed from a brGDGT based proxy (blue; ± 2 σ) and relative 2010 data point in approximate relative position (purple; ± 2 σ). Charcoal counts reported as the number of fragments per volume $(fragments\ cm^{-3})$ of peat (Orange ± 2 σ). Green boxes indicate relative depths of pollen sampling. Elevation of the deposit is reported as meters above sea level. (Data: Table S2)

[Figure]

**(A)**

**(B)**

Figure 5. (A) Bar charts showing the relative pollen abundance in each portion of the section (error bars = 95% confidence intervals; MASL- Meters Above Sea Level). (B). Pollen plate of select grains encountered in the BP section: (a) *Pinus*, (b) half a *Picea* grain, (c) *Larix*, (d) *Betula*, (e) *Alnus*, (f) *Salix*, (g) *Myrica*, (h) ericaceous grain, (i) *Epilobium*, and (j) Cyperaceae. 50um scale = (a–c), 75um scale = (d–j).

[Figure]

**Figure 6. A ternary plot illustrating the fractional abundances of the tetra- (Ia-c), penta (IIa-c and II′a-c), and hexamethylated (IIIa-c and III′a-c) brGDGTs. The global soil dataset (open circles; De Jonge et al., 2014), the global peat samples (green circles; Naafs et al., 2017), and lake sediments from East Africa (black circles indicate samples from lakes >20°C, red circles indicate samples from lakes between 10–20°C and orange circles designate samples from lakes <10°C; Russell et al., 2018) are included for comparison with the Beaver Pond sediments (blue circles; this study).**

[Figure]

**Figure 7. (a) Modern geographic distribution of observed occurrences of species common to the Beaver Pond species list, (b) Mean temperature of the warmest quarter (summer average) derived from WorldClim, (c) Mean precipitation of the warmest quarter (summer rain) derived from WorldClim, (d) Count of unique fire pixels detected per day, over 10 years from MODIS 6 Fire Product, normalized by area of the latitude by longitude grid.**

**Supplementary Information**

[Figure]

**Figure S1. Molecular structures of all 15 brGDGTs (I-III). The molecules designated with a prime symbol are referred to as the 6-methyl brGDGTs.**

[Figure]

**Figure S2. A comparison of the (A) area and (B) shape (length to width) of the uppermost samples (130–165) that have a higher mean charcoal concentration, and the lowermost samples (0–125)**

[Figure]

**Figure S3. A)** Changes in moss $\Delta^{13}C$ as a function of temperature. Where the entire range of temperatures represents the lapse rate of moss samples from the Peruvian Andes and the red box represents the mean summer temperature range for the Arctic derived from our tetraether measurements. **B)** Changes in moss $\Delta^{13}C$ only as a function of $pO_2$ showing an increase in $\Delta^{13}C$ with elevation (i.e. reduced $pO_2$).

[revised manuscript text omitted]

---

## Author Comment (AC4) · 16 Oct 2018

Peer Review: Tamara Fletcher et al. "The role of elevated atmospheric CO2 and increased fire in Arctic amplification of temperature during the Early to mid-Pliocene"

My goodness, how important a simple beaver pond has become. But this pond is 4 million years old, at 77 degrees North latitude and may hold important data that can explain discrepancies in paleoclimate reconstructions for a critical period in recent earth history. This is an important paper and show cases the impact of biogeochemistry methods on paleoecology. Unfortunately, I can offer little insight with this part of the paper. My lack of knowledge and inability to read the equations which were in Chinese or special characters accounts for this. This paper is not, however, without it's faults at other levels.

While much is made of the role of fire in amplifying the arctic temperature response to elevated CO2 there is surprisingly little discussion on the topic. Page 19, lines 18- 22, mention Feng et al. 2016 and the direct and indirect effects but these are in the most general terms. Line 32 in the Conclusions is similarly uninformative and therefore unconvincing. We go from great methodological detail to the most general statements in the main focus of the paper.

RE: Also in reply to the other reviewers, we have now edited the paper with the intent to convey that although the impetus for the investigation is the influence of fire and CO2 on arctic amplification of temperature, this study provides necessary data for future studies rather than addressing the issue directly. The need for modelling experiments to address this quantitatively is now made explicit at the end of the conclusions.

Page 4, Site Description leaves much to be desired. Luckily, I had a copy of Mitchell et al. 2016 to provide the details. I certainly couldn't follow what Fletcher et al. were describing. Perhaps there is need for a site diagram to illustrate the stratigraphy, the Units and where and when specific sampling was done. That the BP site has been collected off and on for more than a decade and perhaps into the future and the data assembled is so important means that reproducibility is very important. Therefore, to me, detailed site description and sample locations are critical.

I'm assuming the fen (Beaver Pond) peat is autochthonous even through the over all site is in a fluvial environment. I'm assuming that a till is the surface deposit and that the surface represents the stratigraphic datum and all Unit and sample measurements are from the surface. I'm assuming that Unit III (Mitchell et al.) is the fen peat.

RE: We have updated the description to clarify our sampling location's relationship to the detailed stratigraphic diagrams from Mitchell et al 2016. As per Mitchell, the peat is considered autochthonous, formed in place in a dam or pond section within a fluvial environment. The site is

not excavated from above, thus having a simply measurable depth below a surface, but is a naturally exposed section on a steep hillside. The till layer above is thick, unstable and eroding. The datum for the 2006 sampling was not taken from the glacial till above the site. Precise comparison between sampling years is difficult.

Let me look at page 8, Vegetation and Fire Reconstruction. Page 8, Line 28, sampled at an upper and lower elevation . . . that correspond with changes in charcoal.
Explain "upper and lower elevation" or provide depths. Does this mean that charcoal samples were processed first and the pollen samples selected on those results?

RE: Yes, pollen sampling was conducted after the charcoal was processed and the sample depths were chosen with the observed change in charcoal in mind. Clarification and more detail is now provided in this section.

P. 9, L., 3, plant taxa.

RE: We have made the edit based on this suggestion.

L. 4, is there a better word for "observation"

RE: We have changed this to 'occurrence' as per the GBIF terminology.

L. 23. What is meant by "The age of the Beaver Pond peat is stratigraphically younger . . ." Younger than what? Need diagram.

RE: This paragraph has been substantially revised now to improve clarity based on reviewer comments.

L.25, where does the 104 t0 105 years come from? And how can you be this precise?

RE: This was a typo and the 4 and 5 should have been superscripted. This paragraph has been revised now to improve clarity based on reviewer comments.

P. 10, l. 2 and 3, check pa and p sunscript a Is this a typo or a different measure

RE: This section has undergone significant revision based on other reviewer comments.

P. 11, Figures 6 and 5 are reversed. 6 is referred to before 5.

RE: Thank you for this note. The figure orders have now been addressed.

P.12, L.20, 21 and 25. Do the authors mean "samples" or sections? If "sections" then I'm not sure what they are talking about as one samples sections in the course of field work and samples are processed in subsequent lab work.

RE: We have now corrected our usage of terms here.

Line 29, describes potential Populus pollen. Since this is such an important component of the pollen assemblage, I am surprised that more effort wasn't made to identify the unknown. Possibly SEM analysis, opinion of other experts, even DNA. Populus is afterall capable of inhabiting high latitudes, is an important species for beavers and almost an expected component in a boreal forest environment. Were wood fragments identified?

RE: Multiple experts were consulted on the potential *Populus* pollen and a definitive identification could not be made due to preservation. The resources required for further analysis of these palynomorphs are not available to the team at this time.
No wood fragments were examined as part of this study.

Was a comparison made between macrofossils and pollen taxa? Wouldn't that be interesting and useful?

RE: Macrofossil plant material was not analysed as part of this study. Although it may have been interesting, the sample size required to produce a useful comparison at high resolution was not available.

Why are no pollen sums presented?

RE: The cumulative percentages of each pollen type are presented in figure 6a. Given this limited samples and age-depth model, this was regarded as a suitable way to visualize the data.

Page 13, L. 2, Does 6% Pinus pollen indicate that pine was a component locally? Would the work of Jocelyne Bourgeois, GSC, have any bearing on or aid these interpretations? She analyzed high latitude pollen and demonstrated how pollen and charcoal could be transported long distances.

RE: Although both pollen and charcoal are subject to transport there are reasons to believe the signal is local. *Pinus* of two kinds have been found as macrofossils at the site previously, although the exact stratigraphic relationship between those samples and our pollen is not known. The volume of charcoal is quite remarkable, and the size/shape of charcoal also suggests limited transport.

Page 13, L.28, Such a lack of precision. What does "very close to the BP peat" mean?

RE: The position of sampling in relation to the peat is given in the methods "approximately 4–5 m above and 30 m to the southwest of the peat." Given the rate of deposition of the sediment type and error on the age estimates, this is considered very close in terms of likely age difference.

Page 17, L. 13. Here is another example of field work and site description problems. "It is possible that the Larix-Betula Parkland dominated . . . correspond to the . . . Units II and III." Why aren't they sure? Was there a continuity in field workers over the seasons of field work or was N. R. the only participant carryover? Has the site eroded from 2008 to 2010 to 2012?

RE: The participant in charge of sampling for this material in 2006 was not present in the subsequent years, when the primary stratigraphic effort was undertaken. There is erosion of the site between years, and the exact '00' from 2006 is only approximately known compared to the 2010 samples.

Page 16 to 20. Discussion and Conclusions
What type of data would the presence of fire provide climate modelers to improve their simulations? Albedo? Canopy transpiration-evaporation? Surface texture, snow capture and melt? Fire reoccurrence is an important measure but what about regrowth? Northern B.C. fires can decimate a landscape and within five years it is lush with deciduous regrowth and conifer seedlings.

RE: There are many direct and indirect radiative effects of forest fires that may contribute to arctic amplification. Albedo is one, through changes in the foliage cover from bare earth to regrowth and from one kind to another during succession. Other potentially important influences include, but are not limited to, black carbon deposition on the surface of ice and snow and additional contribution of aerosols – as we are starting to see from modelling experiments still in progress.

If this paper is to be significant and meet the authors claims they must be able to more fully discuss the importance of fire amplification and model conditions.

RE: Further studies modelling the impact of fire are underway, but out of the scope of this study. We have changed the title and edited the introduction to more clearly indicate this study's role in the investigation. As above, the need for modelling experiments to address this quantitatively is now made explicit at the end of the conclusions.

Nearly every pollen sample in the world will contain some charcoal of varying size classes. What then makes this work unique and important?

RE: Although it may seem apparent that where there is boreal forest, there is forest fire, this was not shown until study of these sediments was conducted. The atmospheric conditions for producing lightening were not proven, and although coal seam fires may be an alternative, it seems likely lightening was igniting wildfire in the Pliocene Arctic due to suitable climatic conditions, such as mean summer temperatures consistently above that 13.5°C threshold identified by Young et al 2017. This has implications for Palaeoclimate modelling. In addition, the charcoal found is not a small or trace amount, but highly variable and abundant in the sediments suggesting fire was an important component of the ecosystem. It is only recently that tundra fire has been increasing in the Arctic regions, and this supports that further increases in

wildfire are likely as both temperature rise and vegetation expands northward providing fuels. This is the highest latitude evidence for fire during the Pliocene and provides important data for validation and boundary conditions for future palaeoclimate modelling efforts.

The text will need some careful editing. There are abbreviations that I don't recognize and I enjoy the author's recognition of that "There are numerous assumptions . . ." p. 15, L. 20. No doubt the measured ages, temperatures and precipitations will be refined in the future as methods improve and assumptions become more sound.

RE: Refinement would be ideal and further research in the Canadian Pliocene High Arctic necessary to test the hypotheses presented, however, the estimates here are in agreement with other proxies, providing support for the values presented.

I believe this is a useful, perhaps significant paper but it needs further work in certain key areas that will help to make the case that fire is, was, an important factor in amplification of arctic temperatures.

RE: We thank Professor Schweger for the time invested in a thorough consideration of our manuscript.

Respectfully yours,
Charles Schweger, Professor Emeritus

**Elevated atmospheric $CO_2$ and fire linked to arctic amplification of temperature during the Early to mid-Pliocene**

Tamara Fletcher[1*], Lisa Warden[2*], Jaap S. Sinninghe Damsté[2,3], Kendrick J. Brown[4,5], Natalia Rybczynski[6,7], John Gosse[8], and Ashley P Ballantyne[1]

[1] College of Forestry and Conservation, University of Montana, Missoula, 59812, USA
[2] Department of Marine Microbiology and Biogeochemistry, NIOZ Royal Netherlands Institute for Sea Research, Den Berg, 1790, Netherlands
[3] Department of Earth Sciences, University of Utrecht, Utrecht, 3508, Netherlands
[4] Natural Resources Canada, Canadian Forest Service, Victoria, V8Z 1M, Canada
[5] Department of Earth, Environmental and Geographic Science, University of British Columbia Okanagan, Kelowna, V1V 1V7, Canada
[6] Department of Palaeobiology, Canadian Museum of Nature, Ottawa, K1P 6P4, Canada
[7] Department of Biology & Department of Earth Sciences, Carleton University, Ottawa, K1S 5B6, Canada
[8] Department of Earth Sciences, Dalhousie University, Halifax, B3H 4R2, Canada
*Authors contributed equally to this work

*Correspondence to*: Tamara Fletcher (tamara.fletcher@umontana.edu)

**Abstract.** The mid-Pliocene is a valuable time interval for understanding the mechanisms that determine equilibrium climate at current atmospheric $CO_2$ concentrations. One intriguing, but not fully understood, feature of the early to mid-Pliocene climate is the amplified arctic temperature response. Current models underestimate the degree of warming in the Pliocene Arctic and validation of proposed feedbacks is limited by scarce terrestrial records of climate and environment, as well as discrepancies in current $CO_2$ proxy reconstructions. Here we reconstruct the $CO_2$, summer temperature and fire regime from a sub-fossil fen-peat deposit on west-central Ellesmere Island, Canada, that has been chronologically constrained using radionuclide dating to 3.9 +1.5/-0.5 Ma.

An empirical transfer function was derived and applied to carbon isotopic measurements of paleo mosses to yield an estimate of Pliocene mean atmospheric $CO_2$ concentrations of $410 \pm 50$ ppm, which are slightly lower than theoretical model predictions of 510 
[revised manuscript text omitted]

where the fractionations of atmospheric CO$_2$ due to diffusion ($a = \sim$ -4.4 ‰) and carboxylation by the enzyme rubisco ($b = \sim$ -27 ‰) are constraints. Thus, isotopic fractionation in C3 plants ($\Delta$ $^{13}$C$_{C3}$) is largely a function of stomatal control of partial pressure of intercellular CO$_2$ – ($p_i$) with respect to the partial pressure of atmospheric CO$_2$ ($p_a$). However, bryophytes lack stomata and thus a mechanism for actively regulating $p_i$, such that isotopic fractionation ($\Delta$

$^{13}C_{bryo}$) varies mainly as a function of partial pressure in atmospheric $CO_2$ (i.e. $p_a$). While other environmental factors, such as humidity, temperature, light availability, and microclimate may also play important roles in isotopic discrimination in bryophytes (Fletcher et al., 2008; Ménot and Burns, 2001; Royles et al., 2014; Skrzypek et al., 2007; Waite and Sack, 2011; White et al., 1994), the first order control on discrimination is the partial pressure of atmospheric $CO_2$ (Fletcher et al., 2008; White et al., 1994). Because atmospheric $CO_2$ is relatively well mixed in the troposphere its mean annual concentration does not differ significantly by location. However, because total atmospheric pressure decreases with atmospheric height ($h$), the partial pressure of atmospheric $CO_2$ must also decrease according to the following exponential function:

$$p_{a(h)} = p_{a(i)} e^{-h/H} \qquad (2)$$

such that the partial pressure of atmospheric $CO_2$ at any given height in the atmosphere ($p_{a(h)}$) can be calculated based on the initial atmospheric partial pressure of atmospheric $CO_2$ ($p_{a(i)}$) and a reference height ($H = 7600$ m), where atmospheric pressure goes to 0.37 Pa (Bonan, 2015). Therefore assuming that carbon isotopic discrimination in bryophytes varies in response to the partial pressure of atmospheric $CO_2$ we can predict from basic physical principles an increase in $\Delta\ ^{13}C_{bryo}$ in response to an increase in $p_{a(h)}$. Furthermore, if the assumptions of this empirical relationship are valid, then this empirical relationship can in theory be used to predict the partial pressure of atmospheric $CO_2$ based on carbon isotopic measurements of bryophytes.

To test this prediction, we compiled data from four studies investigating carbon isotopic variability of different bryophytes, primarily mosses, along elevational transects at different locations. Based on the elevations and locations of moss samples, the atmospheric partial pressure of atmospheric $CO_2$ was estimated from ERA-interim reanalysis data of total atmospheric pressure (Dee et al., 2011) in conjunction with globally averaged atmospheric $CO_2$ concentrations (Global View-$CO_2$, 2013) from the years moss samples were collected. For our analysis we only included measurements of carbon isotopic variability in non-vascular mosses and all isotopic values were normalized to cellulose based on the empirical relationship reported by Ménot and Burns (2001). Carbon isotopic discrimination values for all plant material was calculated as:

$$\Delta^{13}C = (\delta^{13}C_{atm} - \delta^{13}C_{plant})/(1 + \delta^{13}C_{plant}/1000) \qquad (3)$$

where $\delta^{13}C_{plant}$ represents the C isotopic composition of plant cellulose and $\delta^{13}C_{atm}$ represents the mean annual carbon isotopic composition of atmospheric $CO_2$ of the year when samples were collected (Global View-$CO_2$, 2013), or in the case of sub-fossil mosses, when the samples were growing. The response $\Delta\ ^{13}C$ to $pCO_2$ across elevational gradients in modern mosses was then used to calibrate the theoretical model BRYOCARB that has been developed to reconstruct past $CO_2$ levels based on measurements of $^{13}C$ in paleo bryophytes. Thus, our approach provides two independent estimates of Pliocene $CO_2$ concentrations – one empirically derived from our transfer function and the other predicted from the BRYOCARB model calibrated to modern mosses and constrained by our paleoclimate reconstructions.

In order to derive estimates of atmospheric $CO_2$ concentrations during the Pliocene, isotopic composition of source $CO_2$ from the atmosphere (ie. $\delta^{13}C_{atm}$) was estimated during the Pliocene to solve for $\Delta^{13}C$ of mosses (Eq.(3)). This was accomplished by simultaneous measurements of $\delta^{13}C$ in the C3 plant buckbean (*Menyanthes trifoliata* L.) that was identified as a subfossil specimen at the BP site and was also collected from four different sites in the Canadian Boreal Forest. Although it has been demonstrated that $\Delta^{13}C$ of C3 plant material is sensitive to many factors, including mean annual precipitation and altitude (Diefendorf et al. 2010), there is less variability within biomes, so modern buckbean was sampled from within the Canadian Boreal biome that we suspect is very similar to the BP site based upon paleovegetation (Ballantyne et al. 2010, Fletcher et al. 2017). Measurements of $\delta^{13}C$ on modern buckbean were used to constrain estimates of $p_i / p_a$ using modern estimates of $\delta^{13}C_{atm}$ from when the buckbeans were collected. This constrained modern value of $p_i / p_a$ was then applied to our sub-fossil buckbean samples to estimate $\delta^{13}C_{atm}$ during the Pliocene. All plant and moss material were rinsed and placed in a sonicating bath with deionized water to remove any paleosoil from samples. The diagnostic material for mosses was leafy material, whereas buck bean was identified base on seeds. Therefore, to ensure that our isotopic measurements were made on similar compounds, cellulose was extracted from samples according to 
[revised manuscript text omitted]
$_2$ both in empirical observations and theoretical predictions (Fig. 3). However, a much greater change in $\Delta^{13}C_{moss}$ is observed in response to $p_a$ than is predicted from the optimized BRYOCARB simulations. The empirical fit to the observed change in $\Delta^{13}C_{moss}$ in response to $p_a$ is slightly better (RMSE = 1.8 ‰) than the theoretical prediction from the BRYOCARB model (RMSE = 2.1 ‰), but the slopes are quite different, with our empirical slope (0.56 ‰/$p_a$) an order of magnitude greater than the linear approximation of the BRYOCARB slope (0.07 ‰/$p_a$), suggesting that other non-linear processes and not just $p_a$ may be affecting $\delta^{13}C_{moss}$ variability with elevation.

While there does appear to be a global relationship between $p_a$ and $\Delta^{13}C$ of mosses, there are notable differences among sites. Moss $\Delta^{13}C$ values tended to be generally lower in the Swiss Alps (mean = 17.4 ‰) and higher in Hawaii (mean = 20.6 ‰) and the slope of the relationship between $p_a$ and $\Delta^{13}C$ appears to vary across sites with the Andes having the smallest slope and Poland having a much greater slope. We used the BRYOCARB model to test the sensitivity of $\Delta^{13}C$ to other variables that change as a function of elevation (e.g. temperature and $p$O$_2$). According to our BRYOCARB simulations, with all other variables held constant decreased temperature with increased elevation should slow metabolic rates resulting in an increase in $\Delta^{13}C$ (Supplemental Figure S3), which directly contradicts observations (Fig. 3). Furthermore, the range of mean summer temperature estimates from the Pliocene BP site could only explain ~0.2 ‰ isotopic response in our moss samples. Similarly we evaluated the effect of just changing $p$O$_2$ in our BRYOCARB simulations and found a decrease in $\Delta^{13}C$ with increasing $p$O$_2$ that is opposite to the $\Delta^{13}C$ response of mosses to partial pressure across all elevational transects. We also evaluated model performance using a global standard atmospheric sea level pressure of 101.325 kPa, or site-specific atmospheric pressure estimates from ERA-interim reanalysis data. We found that the model using site specific atmospheric pressure estimates performed better at predicting $\Delta^{13}C_{moss}$ (RMSE = 1.096 ‰) than the model using global standard atmospheric sea level pressure (RMSE = 1.216 ‰). Therefore, it appears that partial pressure of atmospheric CO$_2$ is the primary physical mechanism explaining the global relationship between $\Delta^{13}C$ of mosses and elevation and that other factors, such as water availability that may be mediated by different lapse rates (Ménot and Burns, 2001; Royles et al., 2014; Skrzypek et al., 2007; Waite and Sack, 2011), may explain variability among sites. Thus, the optimal model characterizing the observed modern relationship between $\Delta^{13}C_{moss}$ and the $p_a$ was:

$$13C\ \Delta13C_{moss} = 0.56 \text{ x } p\text{CO}_2 + 1.55 \tag{9}$$

Based on our analysis of cellulose extracted from four different *Menyanthes* L. (i.e. buckbean) plants growing at four different locations in the modern boreal forest, we found $\Delta^{13}C$ of buckbean to be fairly constant 16 ± 0.4 ‰, yielding an estimate of $p_i$ / $p_a$ in modern buckbean of 0.51. Applying this modern of $p_i$ / $p_a$ to our $\delta^{13}C$ measurements

from sub-fossil buckbean we obtained estimates of $\delta^{13}C_{atm}$ during the Pliocene of -6.23 ± 0.9 ‰. Using our empirical transfer function (Eq. 9) in combination with these estimates of $\delta^{13}C_{atm}$, we were able to approximate atmospheric $CO_2$ concentrations over the Pliocene interval captured at the BP site (Fig. 4). We estimated a mean atmospheric $CO_2$ concentration over this interval of 410 ± 50 ppm (mean ± transfer error and instrument error) with considerable variability between a minimum atmospheric $CO_2$ concentration of 296 ppm and a maximum atmospheric $CO_2$ concentration of 480 ppm. Predicted values of Pliocene $CO_2$ from the BRYOCARB model were slightly higher at 510 ppm, but the single standard deviation across all estimates was extremely high (967 ppm), suggesting that the BRYOCARB simulations are not significantly different from our empirical model estimates; however, the BRYOCARB model is too sensitive to our range of $\Delta^{13}C_{moss}$ estimates and thus not very precise.

[revised manuscript text omitted]
 transects in the Swiss Alps (black dots; Ménot and Burns, 2001), the Peruvian Andes (blue dots; Royles et al., 2014) ), the mountains of Poland (red dots; Skrzypek et al. 2007), and Hawaii (green dots; Waite and Sack 2011). Partial pressure of atmospheric CO₂ calculated from atmospheric surface pressure reanalysis data (Dee et al., 2011) combined with atmospheric CO₂ observations from year moss samples were collected. All carbon isotopic measurements of mosses have been normalized to cellulose based on published regression of cellulose and whole moss values (Ménot and Burns, 2001) and reported as discrimination (Δ) from atmospheric δ¹³CO₂ (GlobalGlobal View-CO₂, 2013) from the year mosses were collected in units of ‰. Empirical model fit (black line) is plotted with prediction intervals (black dashed) compared with predictions from the BRYOCARB model (Fletcher et al. 2008) with parameters optimized to match observations.**

[Figure]

**Figure 4. Reconstruction of atmospheric CO₂, mean summer temperature, and fire for the Canadian High Arctic during the Pliocene.** Atmospheric CO₂ concentrations estimated from carbon isotopic measurements of mosses and plants (red; ± 2 σ). Mean summer temperature reconstructed from a brGDGT based proxy (blue; ± 2 σ) and relative 2010 data point in approximate relative position (purple; ± 2 σ). Charcoal counts reported as the number of fragments per volume (fragments cm⁻³) of peat (Orange ± 2 σ). Green boxes indicate relative depths of pollen sampling. Elevation of the deposit is reported as meters above sea level. (Data: Table S2)

[Figure]

**Figure 5. A ternary plot illustrating the fractional abundances of the tetra- (Ia-c), penta (IIa-c and II′a-c), and hexamethylated (IIIa-c and III′a-c) brGDGTs. The global soil dataset (open circles; De Jonge et al., 2014), the global peat samples (green circles; Naafs et al., 2017), and lake sediments from East Africa (black circles indicate samples from lakes >20°C, red circles indicate samples from lakes between 10–20°C and orange circles designate samples from lakes <10°C; Russell et al., 2018) are included for comparison with the Beaver Pond sediments (blue circles; this study).**

∞∞∞∞∞∞∞∞∞∞∞∞∞∞∞∞∞∞∞∞∞∞∞∞∞Page Break∞∞∞∞∞∞∞∞∞∞∞∞∞∞∞∞∞∞∞∞∞∞∞∞∞∞∞∞∞

**(A)**

[Figure]

**(B)**

**Figure 6. (A) Bar charts showing the relative pollen abundance in each portion of the section (error bars = 95% confidence intervals; MASL- Meters Above Sea Level). (B). Pollen plate of select grains encountered in the BP section: (a) *Pinus*, (b) half a *Picea* grain, (c) *Larix*, (d) *Betula*, (e) *Alnus*, (f) *Salix*, (g) *Myrica*, (h) ericaceous grain, (i) *Epilobium*, and (j) Cyperaceae.  50um scale = (a–c), 75um scale = (d–j).**

[Figure]

[Figure]

Figure 6. A ternary plot illustrating the fractional abundances of the tetra- (Ia-c), penta (IIa-c and II′a-c), and hexamethylated (IIIa-c and III′a-c) brGDGTs. The global soil dataset (open circles; De Jonge et al., 2014), the global peat samples (green circles; Naafs et al., 2017), and lake sediments from East Africa (black circles indicate samples from lakes >20°C, red circles indicate samples from lakes between 10–20°C and orange circles designate samples from lakes <10°C; Russell et al., 2018) are included for comparison with the Beaver Pond sediments (blue circles; this study). ¶
·····································Page Break·····································

**Figure 7. (a) Modern geographic distribution of observed occurrences of species common to the Beaver Pond species list, (b) Mean temperature of the warmest quarter (summer average) derived from WorldClim, (c) Mean precipitation of the warmest quarter (summer rain) derived from WorldClim, (d) Count of unique fire pixels detected per day, over 10 years from MODIS 6 Fire Product, normalized by area of the latitude by longitude grid.**

**Supplementary Information**

[Figure]

**Figure S1. Molecular structures of all 15 brGDGTs (I-III). The molecules designated with a prime symbol are referred to as the 6-methyl brGDGTs.**

[Figure]

**Figure S2. A comparison of the (A) area and (B) shape (length to width) of the uppermost samples (130–165) that have a higher mean charcoal concentration, and the lowermost samples (0–125)**

[Figure]

**Figure S3. A)** Changes in moss $\Delta^{13}C$ as a function of temperature. Where the entire range of temperatures represents the lapse rate of moss samples from the Peruvian Andes and the red box represents the mean summer temperature range for the Arctic derived from our tetraether measurements. **B)** Changes in moss $\Delta^{13}C$ only as a function of $pO_2$ showing an increase in $\Delta^{13}C$ with elevation (i.e. reduced $pO_2$).

[revised manuscript text omitted]

*2010 field season sample

**Table S3. $^{26}$Al/$^{10}$Be burial ages**

| PDF max age (yr) | 1s error +ve (yr) | 1s error -ve (yr) | meanvalue (yr) |
|---|---|---|---|
| 3.62E+06 | 1.50E+06 | 4.78E+05 | 3.58E+06 |
| 3.94E+06 | 3.66E+06 | 4.77E+05 | NA |
| 4.07E+06 | 5.79E+06 | 3.51E+05 | NA |
| 3.95E+06 | 1.49E+06 | 4.31E+05 | 3.95E+06 |

**Table S4. Input for burial modeling**

| Depth cm | Bulk Density g cm$^{-3}$ | Latitude deg | Longitude deg | Surface elevation m | $^{10}$Be conc atoms g$^{-1}$ | $^{10}$Be conc err atoms g$^{-1}$ | $^{26}$Al conc atoms g$^{-1}$ | $^{26}$Al conc err atoms g$^{-1}$ | Eros Rate cm ka$^{-1}$ |
|---|---|---|---|---|---|---|---|---|---|
| 18050 | 2.2 | 78.550 | -82.373 | 333 | 17665 | 402 | 26986 | 7335 | 2.25 |
| 18097 | 2.2 | 78.550 | -82.373 | 333 | 16163 | 376 | 22263 | 7889 | 2.25 |
| 18155 | 2.2 | 78.550 | -82.373 | 333 | 17853 | 387 | 22322 | 10215 | 2.25 |
| 18222 | 2.2 | 78.550 | -82.373 | 333 | 20505 | 598 | 26508 | 7147 | 2.25 |

Notes:

[revised manuscript text omitted]

---

## Author Comment (AC5) · 23 Oct 2018

Please find below the details of the Non-Arboreal Pollen at Beaver Pond as requested by Professor Schweger. Apologies that this was not included in the previous reply.

Salix Myrica Ericaceae Poaceae Asteraceae Rosaceae Amaranthaceae Onagraceae Sacrobatus Lonicera Cyperaceae Typha Menyanthes Sagittaria Triglochin

Regards, Tamara Fletcher (on behalf of the authorship team)
* * *

---

## Referee Report (RR1)

Evidence of fire in the Pliocene Arctic in response to elevated $CO_2$ and temperature

Tamara Fletcher[1*], Lisa Warden[2*], Jaap S. Sinninghe Damsté[2,3], Kendrick J. Brown[4,5], Natalia
Rybczynski[6,7], John Gosse[8], and Ashley P Ballantyne[1]

[1] College of Forestry and Conservation, University of Montana, Missoula, 59812, USA
[2] Department of Marine Microbiology and Biogeochemistry, NIOZ Royal Netherlands Institute for Sea Research, Den
Berg, 1790, Netherlands
[3] Department of Earth Sciences, University of Utrecht, Utrecht, 3508, Netherlands
[4] Natural Resources Canada, Canadian Forest Service, Victoria, V8Z 1M, Canada
[5] Department of Earth, Environmental and Geographic Science, University of British Columbia Okanagan, Kelowna,
V1V 1V7, Canada
[6] Department of Palaeobiology, Canadian Museum of Nature, Ottawa, K1P 6P4, Canada
[7] Department of Biology & Department of Earth Sciences, Carleton University, Ottawa, K1S 5B6, Canada
[8] Department of Earth Sciences, Dalhousie University, Halifax, B3H 4R2, Canada
*Authors contributed equally to this work

*Correspondence to*: Tamara Fletcher (tamara.fletcher@umontana.edu)

**Abstract.** The mid-Pliocene is a valuable time interval for understanding the mechanisms that determine equilibrium climate at current atmospheric $CO_2$ concentrations. One intriguing, but not fully understood, feature of the early to mid-Pliocene climate is the amplified arctic temperature response. Current models underestimate the degree of warming in the Pliocene Arctic and validation of proposed feedbacks is limited by scarce terrestrial records of climate and environment, as well as discrepancies in current $CO_2$ proxy reconstructions. Here we reconstruct the $CO_2$, summer temperature and fire regime from a sub-fossil fen-peat deposit on west-central Ellesmere Island, Canada, that has been chronologically constrained using radionuclide dating to 3.9 +1.5/-0.5 Ma.

An empirical transfer function was derived and applied to carbon isotopic measurements of paleo mosses to yield an estimate of Pliocene mean atmospheric $CO_2$ concentrations of $410 \pm 50$ ppm, which are slightly lower than theoretical model predictions of 510 ppm. The estimate for average mean summer temperature is 15.4±0.8°C using specific bacterial membrane lipids, i.e. branched glycerol dialkyl glycerol tetraethers. Macro-charcoal was present in all samples from this Pliocene section with notably higher charcoal concentration in the upper part of the sequence.

[revised manuscript text omitted]

Although it is generally thought that atmospheric $CO_2$ concentrations of ~ 400 ppm provided the dominant global radiative forcing during the mid-Pliocene, $CO_2$ proxies over the Pliocene do not all agree (Fig. 1). Reconstructions of Pliocene $CO_2$ range between 190 and 440 ppm (Martinez-Boti et al., 2015; Seki et al., 2010). While $CO_2$ estimates from stomata and paleosols tend to be less precise, they are within the range of boron and alkenone derived estimates (Royer, 2006; Foster et al. 2017). Due to this variation in estimates from approximately the same time and variation in $CO_2$ over time, there is no clear value for $CO_2$ concentration in Earth's atmosphere that can be assigned to broad periods 
[revised manuscript text omitted]

where the fractionations of atmospheric CO$_2$ due to diffusion ($a$ = ~ -4.4 ‰) and carboxylation by the enzyme rubisco ($b$ = ~ -27 ‰) are constraints. Thus,  in C3 plants (Δ $^{13}$C$_{C3}$) is largely a function of stomatal control of partial pressure of intercellular $CO_2$ – ($p_i$) with respect to the partial pressure of atmospheric $CO_2$ ($p_a$).

However, bryophytes lack stomata and thus a mechanism for actively regulating $p_i$, such that isotopic fractionation ($\Delta$

$^{13}C_{bryo}$) varies mainly as a function of partial pressure in atmospheric $CO_2$ (i.e. $p_a$). While other environmental factors, such as humidity, temperature, light availability, and microclimate may also play important roles in isotopic discrimination in bryophytes (Fletcher et al., 2008; Ménot and Burns, 2001; Royles et al., 2014; Skrzypek et al., 2007;

Waite and Sack, 2011; White et al., 1994), the first order control on discrimination is the partial pressure of atmospheric $CO_2$ (Fletcher et al., 2008; White et al., 1994). Because atmospheric $CO_2$ is relatively well mixed in the troposphere its mean annual concentration does not differ significantly by location. However, because total atmospheric pressure decreases with atmospheric height ($h$), the partial pressure of atmospheric $CO_2$ must also decrease according to the following exponential function:

$$p_{a(h)} = p_{a(i)}e^{-h/H} \tag{2}$$

such that the partial pressure of atmospheric $CO_2$ at any given height in the atmosphere ($p_{a(h)}$) can be calculated based on the initial atmospheric partial pressure of atmospheric $CO_2$ ($p_{a(i)}$) and a reference height ($H$ = 7600 m), where atmospheric pressure goes to 0.37 Pa (Bonan, 2015). Therefore, assuming that carbon isotopic discrimination in bryophytes varies in response to the partial pressure of atmospheric $CO_2$ we can predict from basic physical principles an increase in $\Delta$ $^{13}C_{bryo}$ in response to an increase in $p_{a(h)}$. Furthermore, if the assumptions of this empirical relationship are valid, then this empirical relationship can in theory be used to predict the partial pressure of atmospheric $CO_2$

based on carbon isotopic measurements of bryophytes.

To test this prediction, we compiled data from four studies investigating carbon isotopic variability of different bryophytes, primarily mosses, along elevational transects at different locations. Based on the elevations and locations of moss samples, the atmospheric partial pressure of atmospheric $CO_2$ was estimated from ERA-interim reanalysis data of total atmospheric pressure (Dee et al., 2011) in conjunction with globally averaged atmospheric $CO_2$

concentrations (Global View-$CO_2$, 2013) from the years moss samples were collected. For our analysis we only included measurements of carbon isotopic variability in non-vascular mosses and all isotopic values were normalized to cellulose based on the empirical relationship reported by Ménot and Burns (2001). Carbon isotopic discrimination values for all plant material was calculated as:

$$\Delta^{13}C = (\delta^{13}C_{atm} - \delta^{13}C_{plant})/(1 + \delta^{13}C_{plant}/1000) \tag{3}$$

where $\delta^{13}C_{plant}$ represents the C isotopic composition of plant cellulose and $\delta^{13}C_{atm}$ represents the mean annual carbon isotopic composition of atmospheric $CO_2$ of the year when samples were collected (Global View-$CO_2$, 2013), or in the case of sub-fossil mosses, when the samples were growing. The response $\Delta$ $^{13}C$ to $pCO_2$ across elevational gradients in modern mosses was then used to calibrate the theoretical model BRYOCARB that has been developed to reconstruct past $CO_2$ levels based on measurements of $^{13}C$ in paleo bryophytes. Thus, our approach provides two independent estimates of Pliocene $CO_2$ concentrations – one empirically derived from our transfer function and the other predicted from the BRYOCARB model calibrated to modern mosses and constrained by our paleoclimate reconstructions.

In order to derive estimates of atmospheric $CO_2$ concentrations during the Pliocene, isotopic composition of source $CO_2$ from the atmosphere (ie, $\delta^{13}C_{atm}$) was estimated during the Pliocene to solve for $\Delta^{13}C$ of mosses (Eq.(3)). This was accomplished by simultaneous measurements of $\delta^{13}C$ in the C3 plant buckbean (*Menyanthes trifoliata* L.) that was identified as a subfossil specimen at the BP site and was also collected from four different sites in the Canadian Boreal Forest. Although it has been demonstrated that $\Delta^{13}C$ of C3 plant material is sensitive to many factors, including mean annual precipitation and altitude (Diefendorf et al. 2010), there is less variability within biomes, so modern buckbean was sampled from within the Canadian Boreal biome that we suspect is very similar to the BP site based upon paleovegetation (Ballantyne et al. 2010, Fletcher et al. 2017). Measurements of $\delta^{13}C$ on modern buckbean were used to constrain estimates of $p_i / p_a$ using modern estimates of $\delta^{13}C_{atm}$ from when the buckbeans were collected. This constrained modern value of $p_i / p_a$ was then applied to our sub-fossil buckbean samples to estimate $\delta^{13}C_{atm}$ during the Pliocene. All plant and moss material were rinsed and placed in a sonicating bath with deionized water to remove any paleosoil from samples. The diagnostic material for mosses was leafy material, whereas buck bean was identified based on seeds. Therefore, to ensure that our isotopic measurements were made on similar compounds, cellulose was extracted from samples according to 
[revised manuscript text omitted]
$_2$ both in empirical observations and theoretical predictions (Fig. 3). However, a much greater change in $\Delta^{13}C_{moss}$ is observed in response to $p_a$ than is predicted from the optimized BRYOCARB simulations. The empirical fit to the observed change in $\Delta^{13}C_{moss}$ in response to $p_a$ is slightly better (RMSE = 1.8 ‰) than the theoretical prediction from the BRYOCARB model (RMSE = 2.1 ‰), but the slopes are quite different, with our empirical slope (0.56

‰/$p_a$) an order of magnitude greater than the linear approximation of the BRYOCARB slope (0.07 ‰/$p_a$), suggesting that other non-linear processes and not just $p_a$ may be affecting $\delta^{13}C_{moss}$ variability with elevation.

While there does appear to be a global relationship between $p_a$ and $\Delta^{13}C$ of mosses, there are notable differences among sites. Moss $\Delta^{13}C$ values tended to be generally lower in the Swiss Alps (mean = 17.4 ‰) and higher in Hawaii (mean = 20.6 ‰) and the slope of the relationship between $p_a$ and $\Delta^{13}C$ appears to vary across sites with the Andes having the smallest slope and Poland having a much greater slope. We used the BRYOCARB model to test the sensitivity of $\Delta^{13}C$ to other variables that change as a function of elevation (e.g. temperature and $pO_2$). According to our BRYOCARB simulations, with all other variables held constant decreased temperature with increased elevation should slow metabolic rates resulting in an increase in $\Delta^{13}C$ (Fig. S3), which directly contradicts observations (Fig.

3). Furthermore, the range of mean summer temperature estimates from the Pliocene BP site could only explain ~0.2

‰ isotopic response in our moss samples. Similarly we evaluated the effect of just changing $pO_2$ in our BRYOCARB

simulations and found a decrease in $\Delta^{13}C$ with increasing $pO_2$ that is opposite to the $\Delta^{13}C$ response of mosses to partial pressure across all elevational transects. We also evaluated model performance using a global standard atmospheric sea level pressure of 101.325 kPa, or site-specific atmospheric pressure estimates from ERA-interim reanalysis data.

We found that the model using site specific atmospheric pressure estimates performed better at predicting $\Delta^{13}C_{moss}$

(RMSE = 1.096 ‰) than the model using global standard atmospheric sea level pressure (RMSE = 1.216 ‰).

Therefore, it appears that partial pressure of atmospheric CO$_2$ is the primary physical mechanism explaining the global relationship between $\Delta^{13}C$ of mosses and elevation and that other factors, such as water availability that may be mediated by different lapse rates (Ménot and Burns, 2001; Royles et al., 2014; Skrzypek et al., 2007; Waite and Sack,

2011), may explain variability among sites. Thus, the optimal model characterizing the observed modern relationship between $\Delta^{13}C_{moss}$ and the $p_a$ was:

$$^{13}C \, \Delta^{13}C_{moss} = 0.56 \times pCO_2 + 1.55 \tag{9}$$

Based on our analysis of cellulose extracted from four different *Menyanthes* L. (i.e. buckbean) plants growing at four different locations in the modern boreal forest, we found $\Delta^{13}C$ of buckbean to be fairly constant 16 ± 0.4 ‰, yielding an estimate of $p_i / p_a$ in modern buckbean of 0.51. Applying this modern of $p_i / p_a$ to our $\delta^{13}C$ measurements from sub-fossil buckbean we obtained estimates of $\delta^{13}C_{atm}$ during the Pliocene of -6.23 ± 0.9 ‰. Using our empirical transfer function (Eq. 9) in combination with these estimates of $\delta^{13}C_{atm}$, we were able to approximate atmospheric

$CO_2$ concentrations over the Pliocene interval captured at the BP site (Fig. 4). We estimated a mean atmospheric $CO_2$

concentration over this interval of 410 ± 50 ppm (mean ± transfer error and instrument error) with considerable variability between a minimum atmospheric $CO_2$ concentration of 296 ppm and a maximum atmospheric $CO_2$

concentration of 480 ppm. Predicted values of Pliocene $CO_2$ from the BRYOCARB model were slightly higher at 510

ppm, but the single standard deviation across all estimates was extremely high (967 ppm), suggesting that the

BRYOCARB simulations are not significantly different from our empirical model estimates; however, the

BRYOCARB model is too sensitive to our range of $\Delta^{13}C_{moss}$ estimates and thus not very precise.

[revised manuscript text omitted]

**in the Swiss Alps (black dots; Ménot and Burns, 2001), the Peruvian Andes (blue dots; Royles et al., 2014) ),**
**the mountains of Poland (red dots; Skrzypek et al. 2007), and Hawaii (green dots; Waite and Sack 2011). Partial**
**pressure of atmospheric CO₂ calculated from atmospheric surface pressure reanalysis data (Dee et al., 2011)**
**combined with atmospheric CO₂ observations from year moss samples were collected. All carbon isotopic**
**measurements of mosses have been normalized to cellulose based on published regression of cellulose and whole**
**moss values (Ménot and Burns, 2001) and reported as discrimination (Δ) from atmospheric δ¹³CO₂**
**(GlobalGlobal View-CO₂, 2013) from the year mosses were collected in units of ‰. Empirical model fit (black**
**line) is plotted with prediction intervals (black dashed) compared with predictions from the BRYOCARB**
**model (blue dashed; Fletcher et al. 2008) with parameters optimized to match observations.**

[Figure]

**Figure 4. Reconstruction of atmospheric CO₂, mean summer temperature, and fire for the Canadian High**
**Arctic during the Pliocene. Atmospheric CO₂ concentrations estimated from carbon isotopic measurements of**
**mosses and plants (red; ± 2 σ). Mean summer temperature reconstructed from a brGDGT based proxy (blue;**
**± 2 σ) and relative 2010 data point in approximate relative position (purple; ± 2 σ). Charcoal counts reported**
**as the number of fragments per volume (fragments cm⁻³) of peat (Orange ± 2 σ). Green boxes indicate relative**
**depths of pollen sampling. Elevation of the deposit is reported as meters above sea level. (Data: Table S3)**

[Figure]

**Figure 5. A ternary plot illustrating the fractional abundances of the tetra- (Ia-c), penta (IIa-c and II′a-c), and**
**hexamethylated (IIIa-c and III′a-c) brGDGTs. The global soil dataset (open circles; De Jonge et al., 2014), the**
**global peat samples (green circles; Naafs et al., 2017), and lake sediments from East Africa (black circles**
**indicate samples from lakes >20°C, red circles indicate samples from lakes between 10–20°C and orange circles**
**designate samples from lakes <10°C; Russell et al., 2018) are included for comparison with the Beaver Pond**
**sediments (blue circles; this study).**

[Figure]

**(A)**

**(B)**

Figure 6. (A) Bar charts showing the relative pollen abundance in each portion of the section (error bars = 95% confidence intervals; MASL- Meters Above Sea Level). (B). Pollen plate of select grains encountered in the BP section: (a) *Pinus*, (b) half a *Picea* grain, (c) *Larix*, (d) *Betula*, (e) *Alnus*, (f) *Salix*, (g) *Myrica*, (h) ericaceous grain, (i) *Epilobium*, and (j) Cyperaceae. 50um scale = (a–c), 75um scale = (d–j).

[Figure]

(a)

(b)

(c)

(d)

**Figure 7. (a) Modern geographic distribution of observed occurrences of species common to the Beaver Pond species list, (b) Mean temperature of the warmest quarter (summer average) derived from WorldClim, (c) Mean precipitation of the warmest quarter (summer rain) derived from WorldClim, (d) Count of unique fire pixels detected per day, over 10 years from MODIS 6 Fire Product, normalized by area of the latitude by longitude grid.**

---

## Author Response (AR2)

Tamara Fletcher
The University of Montana
Campus Drive,
Missoula, MT, 59812

April 2019

Dear Prof. Alberto Reyes,

Please find uploaded the revised version of our manuscript, cp-2018-60, "Evidence for fire in the Pliocene Arctic in response to amplified temperature" for resubmission to *Climate of the Past*.

After discussion we have decided that the we are at an impasse with regard to the utility and application of the BRYOCARB and novel empirical model. As a result, we have removed the $CO_2$ analysis from the revised manuscript while maintaining discussion of our results within the context of the broader $CO_2$ record for the Pliocene.

Other changes to the manuscript in response to reviewers are detailed below, including clarifying and increasing the explicit discussion of feedbacks in the discussion in response to Dana Royer's suggestions.

We thank the reviewers and yourself for the time invested in this manuscript, and I hope the changes make this manuscript suitable for publication in *Climate of the Past*.

Sincerely,

Tamara Fletcher (On behalf of the authorship team)

**Referee #1: Rienk Smittenberg**

Although the paper is improved compared to the first submission, I still have major problems with their CO2 reconstruction. I would agree with their very general observation that higher pCO2 lead to greater isotope fractionation, however their empirical model has very high - but not well acknowledged - uncertainties rendering it not very useful for a quantitative paleoCO2 reconstruction

Their basic isotope data are missing from the main text and their calculations are still not fully clear.

They introduce the Bryocarb model but do not give any details on it.

RE: We seem to be at an impasse due to differing opinions on the utility of BRYOCARB and the empirical model as devised for this study. As such, the $CO_2$ component of this paper has been removed.

The general writing style is still not very good in part, and not built up logically here and there, this should be given a careful look (again).

I have uploaded an annotated pdf with more detailed comments

Comments copied out of the PDF:

Line 138: Strike out ($\Delta$ 13 C)
RE: The $CO_2$ component of this paper has been removed.

Line 141: Farquhar proposed to use Big delta 13C as a term for isotope fractionation effect (or discrimination) and this is still common in the ecosystem isotope literature. However there is a broader consensus of using epsilon for this, reserving Big Delta to express simply the difference between two pools/species. The isotope fractionation factor, again, is expressed by alpha. At the moment the various styles are mixed in the text and this is confusing. The authors need to be consistent in their isotope language.
RE: The $CO_2$ component of this paper has been removed.

Line 144: Strike out
RE: The $CO_2$ component of this paper has been removed.

Line 145 strike out –
RE: The $CO_2$ component of this paper has been removed.

Line 147: Strike out
RE: The $CO_2$ component of this paper has been removed.

Line 157: What would be the effect of lowering pO2 levels, reducing the inhibition of photosynthesis due to photorespiration?
See Dai, Z., Ku, M.S.B. & Edwards, G.E. Planta (1996) 198: 563.
https://doi.org/10.1007/BF00262643
RE: The $CO_2$ component of this paper has been removed.

Line 176: Note there is a 0.2 permil difference between growing season summer CO2 and mean mean annual.
RE: The $CO_2$ component of this paper has been removed.

Line 178: explain what 'sub-fossil' entails here.

RE: The $CO_2$ component of this paper has been removed.

Line 179: reference to BRYOCARB model missing, and it is totally unclear what this model does and why it is used.
RE: The $CO_2$ component of this paper has been removed.

Line 181: At this point in the text it is fairly unclear what this transfer function entails, one would expect some equation.
RE: The $CO_2$ component of this paper has been removed.

Line 182: If the Bryocarb model is calibrated with own data then how can it be independent from that?
RE: The $CO_2$ component of this paper has been removed.

Line 185: sentence is oddly constructed and hard to understand, and I doubt you went back in time to the Pliocene
RE: The $CO_2$ component of this paper has been removed.

Line 186: simultaneous with what
RE: The $CO_2$ component of this paper has been removed.

Line 188: Strike out  and
RE: The $CO_2$ component of this paper has been removed.

Line 189: There is a quite a large spread in isotope discrimination among different Tunda types, from 14 to 20 permil, with a very high sensitivity to mean annual temperature, see the figure from Buchman&Kaplan (2001) in Pataki et al (2003) GlobBiogeochemCycles17-1022
RE: The $CO_2$ component of this paper has been removed.

Line 189: if it is sensitive to altitude, does that not undermine the assumptions used to make the transfer function?
RE: The $CO_2$ component of this paper has been removed.

Line 192: why not take the ERA interim data?
RE: The $CO_2$ component of this paper has been removed.

Line 192: I assume one gets one (average) estimate of p(i)/p(a) because we only have one atm. 13C?
RE: The $CO_2$ component of this paper has been removed.

Line 206: MAAT?
RE: The suggested change has been made.

Line 221: Strike out
RE: The wording here has been changed.

Line 227: Strike out
RE: Change made

Line 241: Calculating MST comes out of the blue in the text, needs to be introduced a little earlier
RE: This has been introduced at the end of section 2.3

Line 253: It would be useful to mention the RMSE's of these calibrations, i.e. the uncertainty of the proxy (let alone the uncertainty in the measurements)
RE: This is now provided in equations 3, 4 and 5.

Line 294: There are three stippled lines in Fig 3 one being the Bryocarb relation, this is confusing.
RE: The $CO_2$ component of this paper has been removed.

Line 302: In my opinion this large spread of sensitivity is highly problematic. On top, they come from hugely different ecosystems with undoubtedly different types of mosses. Essentially the authors have (re)produced four estimates of the sensitivity (S) of fractionation with elevation, and it clearly shows that the various factors like humidity, temperature, but also possibly canopy effect, wind, etc, play into the game. In my opinion it is not warranted to pool all these results and come with one S, instead they should combine the four estimates of S which then has an uncertainty associated with it (also including the uncertainties of the individual S). That said, they come with a prediction interval in Fig. 3 that does have a spread of 7 permil for any given pCO2. In other words, one needs a 7 permil difference in 13C between fossil mosses arrive at the statement they grew under significantly different pCO2 levels.
RE: The $CO_2$ component of this paper has been removed.

Line 307: But we are discussing the modern calibration here?
RE: The $CO_2$ component of this paper has been removed.

Line 320: An estimate of the error of the slope (and intercept) is missing
RE: The $CO_2$ component of this paper has been removed.

Line 320: 13C x DELTA 13Cmoss ?
RE: The $CO_2$ component of this paper has been removed.

Line 323: add table
RE: The $CO_2$ component of this paper has been removed.

Line 326: and measured fossil moss 13C values. These values should be shown in the paper!!
RE: The $CO_2$ component of this paper has been removed.

Line 328: Not clear at all where this 50ppm error comes from. It is different than the range of 296 - 480?
RE: The $CO_2$ component of this paper has been removed.

Line 328: what is the transfer error?
RE: The $CO_2$ component of this paper has been removed.

Line 333: That it is not different already shows from figure 3 where the Bryocarb solution falls within the 'empirical' range. The latter model is thus equally imprecise. Another problem is the calibration range, which goes to 36 Pa (approx 360ppm) thus anything beyond that is an extrapolation - however there is no indication why the relation should be linear.
RE: The $CO_2$ component of this paper has been removed.

Lines 336-341: Anyone not intricately familiar with the brGDGT literature will be totally confused by these long sentences.
RE: Edits have been made to improve clarity.

Line 360: I'd say the quantification becomes different, not the abundance itself.
RE: This change has been made.

Line 376: If the RMSE of that calibration is +/- 2.5', how can a reconstruction be more precise?
RE: This is the standard deviation, not the RMSE. The text has been edited to reflect this.

Line 450: what is a 'slope of less isotopic discrimination'?
RE: The $CO_2$ component of this paper has been removed.

Line 460: and slopes
RE: The $CO_2$ component of this paper has been removed.

Line 469: The above is a good discussion. Concluding there are uncertainties in the approach. The next step is to quantify that uncertainty.
RE: The $CO_2$ component of this paper has been removed.

Line 475: Strike out
RE: The $CO_2$ component of this paper has been removed.

Line 480: 0.17 mg/C means??
RE: The $CO_2$ component of this paper has been removed.

Line 482: which one is that? the one of -20.9 permil?
RE: The $CO_2$ component of this paper has been removed.

Line 483: Abrupt transition about some other proxies,
RE: The $CO_2$ component of this paper has been removed.

Line 493: And then suddenly back to own estimates
RE: The $CO_2$ component of this paper has been removed.

Line 449: But that is only really because the ocean carbonate system cannot keep up at the moment exchanging and buffering the light C from fossil fuels. To sustain a very low 13CO2 over a long time scale the geological C cycle needs to look very different. Is there any carbonate 13C evidence for low 13CO2?
RE: The $CO_2$ component of this paper has been removed.

Line 555: Many grammatical errors in this paragraph
RE: The authors have made changes that we hope improve this paragraph

Line 582: Posited by whom?

RE: Change made. We posit that.

Line 602: where do these numbers come from?
RE: The exploratory CRACLE analysis described from 597. We now specify which analysis at that point in the text.

Line 649: Importantly, this is just one site and may not be representative for the entire Pliocene Arctic.
RE: Changes to this section of the text now highlight this point and the need for additional palaeofire studies at other sites in the CAA.

Line 654: add uncertainty
RE: This change has been made.

**Referee #2: Dana Royer,**

I'll start my review with two core concerns:

1) There is a fundamental disconnect in the manuscript. The Abstract and Introduction set up as a central tenet the link between fire frequency and climate amplification in the Arctic:

Abstract: "One intriguing, but not fully understood, feature of the early to mid-Pliocene climate is the amplified arctic temperature response. Current models underestimate the degree of warming in the Pliocene Arctic and validation of proposed feedbacks is limited by scarce terrestrial records of climate and environment, as well as discrepancies in current CO2 proxy reconstructions. Here we reconstruct the CO2, summer temperature and fire regime from a sub-fossil fen-peat deposit";

Introduction: "We propose that fire in arctic ecosystems may also be an important mechanism for amplifying arctic surface temperatures during the Pliocene, and so seek to understand its characteristics through quantification from the sediment record".

But this theme is not returned to; not in the Abstract, and not in the Discussion. This leaves the reader unsatisfied. The authors do not even state whether temperature amplification exists for their site (beyond what is predicted from Pliocene global climate models), despite having the (summer) temperatures and CO2 concentrations to do so. That would be step 1.

Let's assume that an exaggerated amplification is present (relative to GCMs). The authors have strong evidence for wildfire. Could wildfire amplify the temperature response to an increase in greenhouse gas forcing (relative to the feedbacks present in GCMs currently used for the Pliocene)? Again, the authors do not lay out these arguments.

An alternative approach would be to present the CO2, temperature, and fire data, and leave it at that, with only some minor comments about climate feedbacks. That is essentially how the manuscript is currently written, if one were to remove the above-mentioned sections in the Abstract and Introduction. That would be a fine paper.

RE: The authors consider that this theme was returned to in the discussion, both implicitly through discussion of the feedbacks between fire and temperature, fire and climate, vegetation and climate, and vegetation and fire, and explicitly 970–976 (current markup manuscript line numbers). This section references the preliminary work conducted on wildfire as a feedback due to its "complex direct impacts on the surface radiative budget and direct and indirect effects on the top of the atmosphere radiative budget (Feng et al., 2016)."

The conclusion linked the interactions between climate, CO2, vegetation and fire. It also explicitly states the need for modelling experiments to "quantitatively investigate the effects [on climate] of the kind of fire regime presented here".

To make the link to feedbacks clearer, have now changed the title of the manuscript, the discussion subheading, added short sections within the discussion that highlight the nature of these relationships as feedbacks, and added more details of the kinds of direct impacts we might expect in the final paragraph of the discussion. We have also devised a new figure that demonstrates the feedbacks between fire, vegetation and temperature in this ecosystem. This aspect has also been de-emphasised in the introduction through the removal of some background material.

2) Given the first set of reviews, I'm surprised that the authors continue to emphasize their empirical CO2 model. The fact that the BRYOCARB slope is shallower than the empirical one (Figure 3) should concern them. As the authors mention in the main text, there's something funky going on with the Poland data. Those data steepen the empirical slope. As the authors also mention in the main text, the Andes data, which span the most elevation and perhaps have the least variability in other environmental factors (like moisture), show a shallower slope that looks close to the BRYOCARB slope. So why emphasize the empirical equation?? Especially because it requires extrapolation beyond the calibration data (which the authors do not acknowledge).

The errors with BRYOCARB are larger (= worse precision) because BRYOCARB more fully takes into account the various possible confounding factors. But the BRYOCARB estimates should be more reliable than the empirical ones (= more accurate), especially when applied to fossil settings, because they are underpinned by universal principles, not a series of regional, present-day empirical measurements.

Whether CO2 was 400 ppm or 500 ppm doesn't make much of a difference for the authors' story. The conceptual background of the BRYOCARB model, and the decisions for the inputs used in the model, need to be stated, though.

RE: As above, we seem to be at an impasse due to differing opinions on the utility of BRYOCARB and the empirical model as devised for this study. As such, the $CO_2$ component of this paper has been removed.

More detailed comments:

Lines 24, 25: Need to say what the uncertainties represent (one-sigma, 95% confidence, etc.).
RE: The CO2 results have been removed and the information for temperature results has now been added.

Line 25: The reader won't know what the "theoretical model" is. Some context is needed. Also, is 410 ppm "slightly lower" than 510 ppm (~20% difference)?

RE: The $CO_2$ component of this paper has been removed.

Lines 28-29: "…promoting taxa increase in abundance." I don't know what this means.
RE: We have added a dash in fire-promoting to clarify that this is an adjective, and also specified examples, *Pinus* and *Picea*.

Line 50: The feedbacks are "engaged", their effects just haven't fully manifested.
RE: The text has been changed as requested.

Line 55: Why is this important?
RE: Now added "for comparability to the modern climate system"

Line 65: What does "it" refer to?
RE: This has been reworded for clarity

Lines 73-74: Would anyone claim that there is a single CO2 value for the entirety of the Pliocene?? Seems like a straw-man.
RE: They may contend that the addition of a $CO_2$ estimate from the same site as temperature estimates are taken is not a useful addition because we have many records of $CO_2$ through the Pliocene, from many proxies already. This points out the issues with using an existing value unless you have very precise age control. This section has been reworked.

Line 145: Remove dash (it looks like a minus sign).
RE: The $CO_2$ component of this paper has been removed.

Line 146: Why say "However" here? It's not needed.
RE: The $CO_2$ component of this paper has been removed.

Line 170: Saying "non-vascular mosses" implies that vascular mosses exist.
RE: The $CO_2$ component of this paper has been removed.

Line 179: The mention of BRYOCARB here comes as a surprise because: 1) it is the first mention of the model and acronym; more context is needed; and 2) the theoretical model has not been properly introduced; equation (1) is not sufficient. Somewhere in the manuscript (or supplement), the choice of inputs needs to be stated and defended. As an aside, Kowalczyk and others recently published an R version of BRYOCARB, which may be more user-friendly (see their supplement):
https://agupubs.onlinelibrary.wiley.com/doi/abs/10.1029/2018PA003356
RE: The $CO_2$ component of this paper has been removed.

Line 196: Was cellulose d13C (not bulk carbon) measured for the extant buckbean? This is not clear. Also, what organs were measured in the extant buckbean? Seeds would make for the best comparison with the fossil seeds.
RE: The $CO_2$ component of this paper has been removed.

Line 299: Why must these processes be nonlinear?
RE: The $CO_2$ component of this paper has been removed.

Lines 328, 377, 388: What are these errors? One-sigma?

RE: The CO2 results have been removed and the information for temperature results has now been added.

Line 331: The uncertainty with the BRYOCARB CO2 estimate should be asymmetric. Have you computed it correctly?
RE: The $CO_2$ component of this paper has been removed.

Lines 483-492: This paragraph doesn't seem necessary. There's no need to criticize other methods here.
RE: The $CO_2$ component of this paper has been removed.

Lines 486-487: The residence time shouldn't matter (other than needing to constrain the boron isotopic composition of sea water). What matters is the relative proportion of the two stable boron isotopes that is incorporated into carbonate minerals.
RE: The $CO_2$ component of this paper has been removed.

Lines 498: What is the value based on forams?
RE: The $CO_2$ component of this paper has been removed.

Line 616: "excepted"?
RE: This correction was made.

Line 654: Don't give new information in the Conclusion (Eureka present-day summer temperature).
RE: This information is now introduced earlier in the discussion.

Figure 4: This would be easier to interpret if it were rotated 90 degrees clockwise, so that the vertical axis is age.
RE: This change, along with the deletion of the $CO_2$ estimates, has been made.

[revised manuscript text omitted]

¶
where the fractionations of atmospheric CO₂ due to diffusion (*a* = ~ -4.4 ‰) and carboxylation by the enzyme rubisco (*b* = ~ -27 ‰) are constraints. Thus, isotopic fractionation in C3 plants (Δ ¹³C₍C3₎) is largely a function of stomatal control of partial pressure of intercellular CO₂ – (*p_i*) with respect to the partial pressure of atmospheric CO₂ (*p_a*). However, bryophytes lack stomata and thus a mechanism for actively regulating *p_i*, such that isotopic fractionation (Δ ¹³C_bryo) varies mainly as a function of partial pressure in atmospheric CO₂ (i.e. *p_a*). While other environmental factors, such as humidity, temperature, light availability, and microclimate may also play important roles in isotopic discrimination in bryophytes (Fletcher et al., 2008; Ménot and Burns, 2001; Royles et al., 2014; Skrzypek et al., 2007; Waite and Sack, 2011; White et al., 1994), the first order control on discrimination is the partial pressure of atmospheric CO₂ (Fletcher et al., 2008; White et al., 1994). Because atmospheric CO₂ is relatively well mixed in the troposphere its mean annual concentration does not differ significantly by location. However, because total atmospheric pressure decreases with atmospheric height (*h*), the partial pressure of atmospheric CO₂ must also decrease according to the following exponential function: ¶
¶
$p_{a(h)} = p_{a(i)}e^{-h/H}$ ⟶⟶⟶⟶⟶⟶⟶(2)¶
¶
such that the partial pressure of atmospheric CO₂ at any given height in the atmosphere (*p_a(h)*) can be calculated based on the initial atmospheric partial pressure of atmospheric CO₂ (*p_a(i)*) and a reference height (*H* = 7600 m), where atmospheric pressure goes to 0.37 Pa (Bonan, 2015). Therefore, assuming that carbon isotopic discrimination in bryophytes varies in response to the partial pressure of atmospheric CO₂ we can predict from basic physical principles an increase in Δ ¹³C_bryo in response to an increase in *p_a(h)*. 
[revised manuscript text omitted]
₂ both in empirical observations and theoretical predictions (Fig. 3). However, a much greater change in Δ¹³C_moss is observed in response to *p_a* than is predicted from the optimized BRYOCARB simulations. The empirical fit to the observed change in Δ¹³C_moss in response to *p_a* is slightly better (RMSE = 1.8 ‰) than the theoretical prediction from the BRYOCARB model (RMSE = 2.1 ‰), but the slopes are quite different, with our empirical slope (0.56 ‰/*p_a*) an order of magnitude greater than the linear approximation of the BRYOCARB slope (0.07 ‰/*p_a*), suggesting that other non-linear processes and not just *p_a* may be affecting δ¹³C_moss variability with elevation. ¶
─While there does appear to be a global relationship between *p_a* and Δ¹³C of mosses, there are notable differences among sites. Moss Δ¹³C values tended to be generally lower in the Swiss Alps (mean = 17.4 ‰) and higher in Hawaii (mean = 20.6 ‰) and the slope of the relationship between *p_a* and Δ¹³C appears to vary across sites with the Andes having the smallest slope and Poland having a much greater slope. We used the BRYOCARB model to test the sensitivity of Δ¹³C to other variables that change as a function of elevation (e.g. temperature and *p*O₂). According to our BRYOCARB simulations, with all other variables held constant decreased temperature with increased elevation should slow metabolic rates resulting in an increase in Δ¹³C (Fig. S3), which directly contradicts observations (Fig. 3). Furthermore, the range of mean summer temperature estimates from the Pliocene BP site could only explain ~0.2 ‰ isotopic response in our moss samples. Similarly we evaluated the effect of just changing *p*O₂ in our BRYOCARB simulations and found a decrease in Δ¹³C with increasing *p*O₂ that is opposite to the Δ¹³C response of mosses to partial pressure across all elevational transects. We also evaluated model performance using a global standard atmospheric sea level pressure of 101.325 kPa, or site-specific atmospheric pressure estimates from ERA-interim reanalysis data. We found that the model using site specific atmospheric pressure estimates performed better at predicting Δ¹³C_moss (RMSE = 1.096 ‰) than the model using global standard atmospheric sea level pressure (RMSE = 1.216 ‰). Therefore, it appears that partial pressure of atmospheric CO₂ is the primary physical mechanism explaining Δ¹³C of mosses and elevation and that other factors, such as water availability that may be mediated by different lapse rates (
[revised manuscript text omitted]

[Figure]

Global soil data set (De Jonge et al., 2014)
Peat data set (Naafs et al., 2017)
Lake sediments <10°C (Russell et al., 2018)
Lake sediments 10-20°C (Russell et al., 2018)
Lake sediments >20°C (Russell et al., 2018)
Beaver Pond sediments (this study)

pentamethylated brGDGTs hexamethylated brGDGTs tetramethylated brGDGTs

**Figure 3, A ternary plot illustrating the fractional abundances of the tetra- (Ia-c), penta (IIa-c and II′a-c), and**
**hexamethylated (IIIa-c and III′a-c) brGDGTs. The global soil dataset (open circles; De Jonge et al., 2014), the**
**global peat samples (green circles; Naafs et al., 2017), and lake sediments from East Africa (black circles**
**indicate samples from lakes >20°C, red circles indicate samples from lakes between 10–20°C and orange circles**
**designate samples from lakes <10°C; Russell et al., 2018) are included for comparison with the Beaver Pond**
**sediments (blue circles; this study).**

[Figure]

Figure 3. Sensitivity of carbon isotopic discrimination to the partial pressure of atmospheric $CO_2$ in mosses sampled from different elevational transects. Moss carbon isotope data collected from an elevational transects in the Swiss Alps (black dots; Ménot and Burns, 2001), the Peruvian Andes (blue dots; Royles et al., 2014) ), the mountains of Poland (red dots; Skrzypek et al. 2007), and Hawaii (green dots; Waite and Sack 2011). Partial pressure of atmospheric $CO_2$ calculated from atmospheric surface pressure reanalysis data (Dee et al., 2011) combined with atmospheric $CO_2$ observations from year moss samples were collected. All carbon isotopic measurements of mosses have been normalized to cellulose based on published regression of cellulose and whole moss values (Ménot and Burns, 2001) and reported as discrimination (Δ) from atmospheric $\delta^{13}CO_2$ (GlobalGlobal View-$CO_2$, 2013) from the year mosses were collected in units of ‰. Empirical model fit (black line) is plotted with prediction intervals (black dashed) compared with predictions from the BRYOCARB model (blue dashed; Fletcher et al. 2008) with parameters optimized to match observations.

Page Break

... [4]

[Figure]

Figure 4. Reconstruction of mean summer temperature and fire for the Canadian High Arctic during the Pliocene. Mean summer air temperature reconstructed from a brGDGT based proxy (blue; ± 2 σ) and relative 2010 data point in approximate relative position (purple; ± 2 σ). Charcoal counts reported as the number of fragments per volume (fragments cm$^{-3}$) of peat (Orange ± 2 σ). Green boxes indicate relative depths of pollen sampling. Elevation of the deposit is reported as meters above sea level. (Data: Table S3)

**(A)**

[Figure]

**(B)**

**Figure 5. (A) Bar charts showing the relative pollen abundance in each portion of the section (error bars =**
**95% confidence intervals; MASL- Meters Above Sea Level). (B). Pollen plate of select grains encountered in**
**the BP section: (a)** *Pinus*, **(b) half a** *Picea* **grain, (c)** *Larix*, **(d)** *Betula*, **(e)** *Alnus*, **(f)** *Salix*, **(g)** *Myrica*, **(h)**
**ericaceous grain, (i)** *Epilobium*, **and (j) Cyperaceae.  50um scale = (a–c), 75um scale = (d–j).**

[Figure]

**Figure 6: Examples of the feedbacks between temperature, vegetation and wildfire at the Beaver Pond site.**

[Figure]

**Figure 7. (a) Modern geographic distribution of observed occurrences of species common to the Beaver Pond species list, (b) Mean temperature of the warmest quarter (summer average) derived from WorldClim, (c) Mean precipitation of the warmest quarter (summer rain) derived from WorldClim, (d) Count of unique fire pixels detected per day, over 10 years from MODIS 6 Fire Product, normalized by area of the latitude by longitude grid.**

---

## Author Response (AR3)

Tamara Fletcher
Institute of Applied Ecology
Chinese Academy of Sciences
Wenhua Rd,
Shenyang, Liaoning, 110164

May 2019

Dear Prof. Reyes,

Please find below our point-by-point reply to the suggested revisions to our manuscript, cp-2018-60, "Evidence for fire in the Pliocene Arctic in response to amplified temperature" for publication in *Climate of the Past*.

The substantive changes are early in the section 4.2, where we have developed the section on fire as a feedback to climate and simplified the section on climate as a feedback to fire, following your suggestions.

Thank you for the time you have invested in this manuscript, and I hope the changes make this manuscript suitable for publication in Climate of the Past.

Sincerely,

Tamara Fletcher (On behalf of the authorship team)

**Editor Decision: Publish subject to minor revisions (review by editor)** (29 Apr 2019)
by Alberto Reyes
Comments to the Author:
Dear Dr. Fletcher,

Thank you for the submission of a revised version of your manuscript on fire in the Pliocene forests of Ellesmere Island. Your decision to remove the CO2 reconstruction from the manuscript removes many of the contentious points that were raised in review, and at this point I'm keen to move forward with publishing your manuscript subject to some minor revision. I'm still concerned that Dana Royer's earlier concerns about the feedback angle in the manuscript haven't been fully dealt with, but I think this can be addressed with minor revision. I also have some other suggestions for minor revision, mostly for clarity, which are detailed below.

line 31: "much greater than present…."
RE: This sentence has been edited.

Lines 65-67: Please check for more recent relevant literature on efforts to model warm intervals such as the Pliocene – a good starting point is the PlioMIP2 website. Depending on your assessment of newer literature, you might want to revise the next two sentences too.
RE: This section has been updated with results from PlioMIP2 published since submitting this manuscript.

Line 157: "charcoal counts" seems more appropriate?
RE: This has been changed to specify counts and measurements.

Line 204: "it has been"… "it" could refer to several things in this context – please clarify.
RE: This sentence has been edited for clarity.

Section 2.3: Please provide a brief explanation of how uncertainty is treated (both with reconstruction uncertainty and propagated analytical uncertainty). Ref. 1 picked up on this point too.
RE: The RMSE are reported in the equations (3, 4 and 5), and the derivation of these is found in the citations given (Pearson et al. 2011; Russel et al. 2018). The analytical uncertainty is now reported at the end of 2.3

Line 433: " 1 cm3"
RE: This change has been made.

Line 444: "…grid cells, was counted…"
RE: This has been changed. "...was tabulated".

Lines 650 and 651: I assume the quoted MAAT and MST are mean +/- stdev for n stratigraphic intervals? Please clarify, and provide the n.

RE: The n=34 is now reported at 296 of the mark-up version, and average is changed to mean for specificity. This n-value matches description of the sample layers used for each analysis given in section 2.1 *" temperature estimates from specific bacterial membrane lipids were taken from 22 of the sample layers collected in 2006 and an additional 12 samples collected in 2010"*.

Line 678: "definitively" not "definitely"
RE: This change has been made.

Line 711: "have" not "has"
RE: This change has been made.

Line 728: seems like a good spot to refer to Fig S3
RE: This has been added.

Line 899: "….suggests mean reconstructed summer temperatures were…"
RE: This change has been made.

Lines 900-907: This section is pretty awkward. The caveat about comparing Pliocene-modern differences in MST vs MAT is a point well taken, the execution is clunky and needs careful wordsmithing. Similarly, the last sentence is a good point but needs some massaging.
RE: I have substantially edited and simplified this section.

Line 948: unclear what is meant by "causatively here"
RE: This has been reworded for clarity.

Line 962: "…low concentration of charcoal in this stratigraphic interval…"
RE: This change has been made.

Line 967-970: use commas to separate clauses in this sentence
RE: This change has been made.

Line 997: do you mean Castor? Or the extinct taxon Dipoides?
RE: This change has been made.

Lines 1010-1012: minor editing to clarify what the many "it" occurrences refer to.
RE: This change has been made.

Lines 1027-1029: This is really just a snapshot in time, so I suggest being a little more equivocal: "Thus, while vegetation and fire regimes seemingly changed through time….." and "…have no apparent trend, within analytical and reconstruction uncertainty."
RE: This change has been made.

Line 1033: "…change in vegetation community"
RE: This change has been made.

Line 1034: what is a fine fuel load? Do you mean fire fuel load?
RE: This is now specified.

Line 1044-1050: This bit on fire and feedback comes as a bit of a surprise at this point in the manuscript. It's also pretty underdeveloped. Is there not more literature to support the contention of a potential feedback to climate warming in a forested High Arctic?
RE: This section now follows paragraph two of the discussion section 4.2. It has been revised to include examples from the literature on modern modelling and observational work related to fire's impact on climate in the high northern latitudes. Most Pliocene climate models have not explicitly evaluated fire, and so coverage of this is necessarily brief.

Line 1053: Careful with the snapshot nature of this study when referring to "the Pliocene"
RE: Specifics have now been added.

Line 1057: suggest omitting "as a feedback" here
RE: This has now been deleted.

[revised manuscript text omitted]